# Optimal Best-Arm Identification Methods for Tail-Risk Measures

**Shubhada Agrawal**
TIFR, Mumbai, India
shubhadaiitd@gmail.com

**Wouter M. Koolen**
CWI, Amsterdam
wmkoolen@cwi.nl

**Sandeep Juneja**
TIFR, Mumbai, India
juneja@tifr.res.in

## Abstract

Conditional value-at-risk (CVaR) and value-at-risk (VaR) are popular tail-risk measures in finance and insurance industries as well as in highly reliable, safety-critical uncertain environments where often the underlying probability distributions are heavy-tailed. We use the multi-armed bandit best-arm identification framework and consider the problem of identifying the arm from amongst finitely many that has the smallest CVaR, VaR, or weighted sum of CVaR and mean. The latter captures the risk-return trade-off common in finance. Our main contribution is an optimal $\delta$-correct algorithm that acts on general arms, including heavy-tailed distributions, and matches the lower bound on the expected number of samples needed, asymptotically (as $\delta$ approaches $0$). The algorithm requires solving a non-convex optimization problem in the space of probability measures, that requires delicate analysis. En-route, we develop new non-asymptotic, anytime-valid, empirical-likelihood-based concentration inequalities for tail-risk measures.

## 1  Introduction

Tail risk is a common term used to quantify losses occurring due to rare events, and has been an important topic in finance, insurance and other safety critical uncertain environments. [44] first formalized the problem of identifying optimal investment in financial assets as a multi-criteria optimization problem of maximizing the average return, while minimizing the risk (measured via variance). Since then, several other risk measures have been considered. Lately, risk-measures based on tails of the distribution, like the conditional value-at-risk (CVaR) and value-at-risk (VaR), have gained popularity in financial regulations and risk management (see, [48, 47]), where the underlying probability distributions are mostly heavy tailed (i.e. having infinite moment generating function for all $\theta > 0$). Informally, for a probability measure $\eta$, VaR at level $\pi \in (0, 1)$ is the $\pi^{th}$ quantile for $\eta$, i.e., the outcome below which there is exactly $\pi$ mass. CVaR at level $\pi$ is the conditional expectation of $\eta$, conditioned on values beyond the VaR at level $\pi$. See Section 2 for precise definitions, and [50, 46] for applications of these risk measures in finance and optimization. As opposed to VaR, CVaR is a coherent risk-measure, and is a preferable metric (see, [5] for precise definition and properties of coherence). Outside finance, these tail-risk measures are being used to control risk in operations management, for example, in inventory management [4], supply chain management [51], etc. Recently, coherent risk measures, especially CVaR, have also been used in connection with fairness in machine learning [58].

The importance of these risk measures in the sequential decision making set-up has well been acknowledged (see, [49, 42]). Typically in the stochastic multi-armed bandit (MAB) literature, the quality of an arm is measured using its mean. Tight asymptotic and finite time guarantees exist for different MAB problems with performance measured by the mean (see, [27, 36, 16, 13, 1, 7, 53]). Also, see [12] for a survey of the variants of stochastic MAB problems. However, maximizing the average reward is not always the primary desirable objective. In clinical trials, for example, the treatment that is good on average might result in adverse outcomes for some patients. In finance, one

is typically interested in balancing the mean return with the risk of extreme losses. Risk sensitivity has been well studied in the online learning setting, where in each round, the player sees reward from every arm (see, [24, 56]). However, there is very limited work which incorporates these risk-measures into the MAB framework.

In this paper, we provide a systematic approach for identifying the distribution (or arm) from a given finite set of distributions (or arms) with minimum tail-risk (as measured by CVaR or VaR, or by a conic combination of mean and CVaR, which we will henceforth refer to as the "mean-CVaR" objective). Adopting the best-arm identification (BAI) framework of the stochastic MAB problem, we consider algorithms that generate samples from the given arms, and are $\delta$-correct, i.e., identify the correct answer (arm with minimum VaR, CVaR or mean-CVaR) with probability at least $1 - \delta$, for some pre-specified confidence level $\delta$. While ensuring $\delta$-correctness, the aim is to minimize the number of samples needed by the algorithm before its termination. This is the typical fixed-confidence setting of the BAI MAB problem (see, [37, 2]). Variants of this problem have been widely studied in the literature, where the best-arm is the one with maximum mean (see, [43, 25, 6, 14, 26, 33, 27, 34]).

A relaxation of the pure exploration setting described above is the $(\epsilon, \delta)$-PAC setting, where the aim is to output an $\epsilon$-optimal arm (for an appropriate notion of $\epsilon$-optimality), with probability at least $1 - \delta$, while minimizing the number of samples generated. [59, 18, 32] consider the pure exploration problem of identifying the arm with minimum risk in the $(\epsilon, \delta)$-PAC setting. While [59] consider both VaR and CVaR as measures of risk, [18, 32] focus on the VaR-problem. Recently, [39] and [35] have studied the BAI MAB problem with CVaR and mean-CVaR objectives, respectively, in the closely related "fixed-budget" framework, in which the total number of samples the algorithm is allowed to take is fixed, and the aim of the algorithm is to minimize the error-probability.

## 1.1 Outline of the approach and main assumption

As a warm-up, we first solve our minimum tail risk identification problems in the simple commonplace setting of arm-distributions belonging to a canonical single parameter exponential family (SPEF) of distributions. Each distribution in this family is uniquely identified with its parameter. We show that both CVaR and VaR are monotonic functions of this parameter, as is the mean. Hence, finding the best-(CVaR/VaR/mean-CVaR) arm reduces to finding the arm with the minimum mean.

Since risk-sensitive objectives are particularly important when there is a non-trivial probability of occurrence of extreme outcomes, it is important to consider arm-distributions beyond canonical SPEF, for which the above-mentioned equivalence breaks. We solve the VaR problem for arbitrary arm distributions.

In contrast the CVaR problem is unlearnable in full generality: on the class of all arm distributions, any $\delta$-correct algorithm requires an infinite number of samples in expectation to identify the best arm amongst any finite collection of arms (Remark 3.1). To avoid this, we impose a mild and standard raw $(1 + \epsilon)$-moment restriction on the arm-distributions. Let $\mathcal{P}(\Re)$ denote the collection of all the probability distributions on the reals $\Re$, and let $B$ and $\epsilon$ be positive constants. For risk measure CVaR and for the mean-CVaR objective, we restrict the class of allowed arm distributions to

$$\mathcal{L} = \left\{ \eta \in \mathcal{P}(\Re) : \mathbb{E}_\eta \left( |X|^{1+\epsilon} \right) \leq B \right\}.$$

We discuss the choice of parameters in Section 1.3 below. For each tail measure, we prove information-theoretic lower bounds on the sample complexity of any $\delta$-correct algorithm, and use these to develop a $\delta$-correct algorithm whose sample complexity exactly matches the lower bound as $\delta \to 0$, for CVaR, mean-CVaR, and VaR problems. The mean-CVaR problem is conceptually and technically similar to the CVaR problem. Hence, for simplicity of presentation, we primarily focus on the CVaR setting in the main text and give details of the mean-CVaR setting in Appendix I. We also spell out the somewhat analogous analysis for the VaR setting towards the end (Section 4.2), with details deferred Appendix to H.

## 1.2 Technical contributions

As is well known in the BAI MAB literature, the lower bound problem takes the underlying arm distributions as inputs and solves for optimal weights that determine the proportion of samples that should ideally be allocated to each arm. The proposed algorithm uses a plugin strategy that at each

sequential stage, modulo mild forced exploration, uses the generated empirical distributions as a proxy for the true distributions and arrives at weights that guide the sequential sampling strategy.

In order to highlight the technical challenges arising in our non-parametric case, we will need to introduce two functionals next that are central to our lower bounds, algorithms, confidence intervals etc.

**Information distance for CVaR problem:** Given $\eta_1, \eta_2$ in $\mathcal{P}(\Re)$, let $\mathrm{KL}(\eta_1, \eta_2)$ denote the KL-divergence between them, i.e., $\mathrm{KL}(\eta_1, \eta_2) := \int \log \frac{d\eta_1}{d\eta_2}(y) d\eta_1(y)$. Furthermore, for the probability measure $\eta$ let $c_\pi(\eta)$ denote its CVaR at the given confidence level $\pi \in (0, 1)$ (see Section 2 for the exact definition). Then, given $\eta \in \mathcal{P}(\Re)$ and $x \in \Re$, we define functionals $\mathrm{KL}_{\inf}^{\mathrm{U}} : \mathcal{P}(\Re) \times \Re \longrightarrow \Re^+$, and $\mathrm{KL}_{\inf}^{\mathrm{L}} : \mathcal{P}(\Re) \times \Re \longrightarrow \Re^+$, where $\Re^+$ denotes the non-negative reals, as

$$\mathrm{KL}_{\inf}^{\mathrm{U}}(\eta, x) := \min_{\kappa \in \mathcal{L}:\, c_\pi(\kappa) \geq x} \mathrm{KL}(\eta, \kappa) \quad \text{and} \quad \mathrm{KL}_{\inf}^{\mathrm{L}}(\eta, x) := \min_{\kappa \in \mathcal{L}:\, c_\pi(\kappa) \leq x} \mathrm{KL}(\eta, \kappa). \quad (1)$$

See [2, 30, 15] for related quantities. These projection functionals appear in the lower bound (Section 3), and are central to our plugin algorithm.

Unlike their analogues in the mean case, $\mathrm{KL}_{\inf}^{\mathrm{U}}$ and $\mathrm{KL}_{\inf}^{\mathrm{L}}$ in (1) are not symmetric, and need to be studied separately. In particular, $\mathrm{KL}_{\inf}^{\mathrm{U}}$ is a convex optimization problem, while $\mathrm{KL}_{\inf}^{\mathrm{L}}$ is not. This is because $c_\pi(\cdot)$ is a concave function, whence, the CVaR constraint in the $\mathrm{KL}_{\inf}^{\mathrm{L}}$ problem in (1) renders the feasible region non-convex (see Section 2). CVaR can be expressed as the optimal value of a minimization problem. This helped in re-expressing $\mathrm{KL}_{\inf}^{\mathrm{L}}$ as minimization over 2 variables, fixing one of which resulted in convex optimization over the other (see Section 3).

For proving $\delta$-correctness, we develop a new concentration inequality for weighted sums of these functionals (Proposition 4.2). Dual representations of these suggest natural candidates for super-martingales, whose mixtures help us in proving the concentration result. Similar inequalities were developed in [38, 20, 54] in different settings. See [40, Chapter 20] for an overview of the method of mixtures. We also propose $\mathrm{KL}_{\inf}^{\mathrm{U}}$- and $\mathrm{KL}_{\inf}^{\mathrm{L}}$-based tight anytime-valid confidence intervals for CVaR for heavy-tailed distributions, and show that classical confidence intervals derived using popular truncation-based estimators can be recovered using our method, with only a minor overhead (see Section 4.3).

Since distributions in $\mathcal{L}$ are not characterized by parameters, we work in the space of probability measures instead of in the Euclidean space. A key and non-trivial requirement for the proof of asymptotic optimality of the algorithm is the joint continuity of $\mathrm{KL}_{\inf}^{\mathrm{L}}$ and $\mathrm{KL}_{\inf}^{\mathrm{U}}$ in a well-chosen metric, which should generate a topology that is sufficiently fine to ensure this continuity, but coarse to ensure fast convergence of the empirical distributions to the true-arm distributions. We endow $\mathcal{P}(\Re)$ with the topology of weak convergence, or equivalently, with the Lévy metric (see Section 2 for definitions). Another nuance in our analysis is that the empirical distributions may not lie in $\mathcal{L}$. This is handled by projecting these on to $\mathcal{L}$ under a suitable metric.

Our proposed algorithm is a plugin strategy that involves solving the lower bound problem using the empirical distributions as a proxy for the actual arm distributions. This can be computationally demanding especially as the underlying samples in the empirical distribution become large. To ease the numerical burden we propose modifications that require solving the lower bound only order $\log(n)$ many times till stage $n$ of the algorithm (where $n$ samples are generated). This modification substantially reduces the computation burden. We show that it is optimal up to a constant (Appendix K).

**VaR problem:** Our algorithm for CVaR, with $\mathrm{KL}_{\inf}^{\mathrm{L}}$ and $\mathrm{KL}_{\inf}^{\mathrm{U}}$ replaced by the corresponding functionals with the VaR constraints instead, is asymptotically optimal for this problem in complete generality (Section 4.2). Here, $\mathrm{KL}_{\inf}^{\mathrm{U}}$ and $\mathrm{KL}_{\inf}^{\mathrm{L}}$ have closed form representations. However, they are no longer jointly-continuous in the Lévy metric, which introduces new technical challenges in the analysis of the algorithm.

### 1.3 Regarding the choice of $\epsilon$ and $B$ in our assumption

Firstly, BAI problems are important in simulation where the best model may need to be identified amongst many intricate models in terms of a performance measure such as CVaR or VaR, using minimal computational effort (see, [31]). Input distributions in simulation are known and may often

involve heavy tails. In some cases, by the use of Lyapunov-function-based techniques, bounds on moments of output random variables, $B$, can be determined. (see, e.g., [28] and references therein). Secondly, consider rewards (returns) from a number of hedge funds. Each time some amount of money is invested into a fund, a random return may be revealed from that fund but not from others. To assume that these returns come from a class of parametric distributions or have known bounded support can be a substantially inaccurate simplification. Typically, from historical analysis, it is known that the distribution of securities have a particular tail index, say, $(1 + \epsilon)$. For stock returns, extensive research suggests that $(1 + \epsilon) \in [2, 5]$. For daily exchange rates and income and wealth distributions we may have $(1 + \epsilon) \in (1, 2]$. Extreme value theory, under reasonable dependence structure amongst underlying securities, shows that a portfolio (a weighted sum) will also have the same tail index of $(1 + \epsilon)$ (see, [19]). So the key approximation needed is in arriving at $B$. It is easy to arrive at distributions $\eta$ and $\kappa$ whose $(1 + \epsilon)^{th}$ moments are arbitrarily far while the KL distance between them is arbitrarily close to zero. This makes it difficult to infer $B$ from a given sample of data without further restriction on the two distributions. One may take a pragmatic view and approximate $B$ by estimating the $(1 + \epsilon)^{th}$ moment from observed samples and padding it up with a reasonably large factor. A further set of distributional assumptions would be needed to justify the above procedure to arrive at $B$. Again, verifying those assumptions will entail similar problems. In practice, one may live with the above approximation even though in rare settings it may be inaccurate and lead to sub-optimal allocations in our algorithm. One accepts this risk as one often accepts the assumption that the distributions of the random samples from each arm are time stationary or are independent, even though these may only be approximately correct.

## 2  Background

For $K \geq 2$, let $\mathcal{M} = \mathcal{L}^K$ denote the collection of all $K$-vectors of distributions, $\nu = (\nu_1, \ldots, \nu_K)$, such that for all $i$, $\nu_i$ belongs to $\mathcal{L}$. Let $\mu \in \mathcal{M}$ be the given bandit problem, and $\pi \in (0, 1)$ denote the fixed confidence level. For $\eta \in \mathcal{P}(\Re)$, let $F_\eta(y) = \eta((-\infty, y])$ denote the CDF function for $\eta$, and let $m(\eta)$ denote mean of measure $\eta$.

**VaR, CVaR:** With the above notation, VaR at level $\pi$ for the distribution $\eta$, denoted as $x_\pi(\eta)$, equals $\min\{z \in \Re : F_\eta(z) \geq \pi\}$. Since $F_\eta(\cdot)$ is a non-decreasing and right-continuous function, the minimum in the expression of VaR is always attained. Define CVaR at level $\pi$, $c_\pi(\eta)$, as

$$c_\pi(\eta) = \frac{F_\eta(x_\pi(\eta)) - \pi}{1 - \pi} x_\pi(\eta) + \frac{1}{1 - \pi} \int_{x_\pi(\eta)}^{\infty} y \, dF_\eta(y).$$

If $\eta$ has a density in a neighbourhood around $x_\pi$, then $c_\pi(\eta) = \mathbb{E}_\eta(X | X \geq x_\pi(\eta))$, i.e., it measures the average loss conditioned on the event that losses are larger than the VaR.

In the figure above, the total shaded area (green and blue regions, together) divided by $1 - \pi$ denotes the CVaR of the measure whose CDF function is displayed in red. To see this, observe that the first term in the expression above, scaled by $(1 - \pi)$, equals the blue region. The integral in the second term when simplified using integration by parts can be seen to equal the green region. There are alternative formulations of CVaR, which we state without proofs.

$$c_\pi(\eta) = \frac{1}{1 - \pi} \int_{p \in [\pi, 1]} x_p(\eta) \, dp = \min_{x_0 \in \Re} \left\{ x_0 + \frac{1}{1 - \pi} \mathbb{E}_\eta((X - x_0)_+) \right\} \tag{2}$$

$$= \max_{v \in M^+(\Re)} \frac{1}{1 - \pi} \int_\Re y \, dv(y) \quad \text{s.t.} \quad \forall y, \ dv(y) \leq d\eta(y) \ \text{and} \ \int_\Re dv(y) = 1 - \pi, \tag{3}$$

where $(x)_+$ denotes $\max\{0, x\}$ and $M^+(\Re)$ denotes collection of all non-negative measures on $\Re$.

From (2), since $c_\pi(\eta)$ is a minimum of linear functions of $\eta$, it is a concave function of $\eta$. Thus, the $\mathrm{KL}_{\mathrm{inf}}^{\mathrm{U}}$ problem in (1) is a convex optimization problem, while the $\mathrm{KL}_{\mathrm{inf}}^{\mathrm{L}}$ problem is not, since the $c_\pi(\cdot)$ constraint makes the feasible region non-convex. See, [50] for a comprehensive tutorial on the two tail-risk measures, and their properties.

**Parametric case:**  Using the definition of VaR, it can be argued that $x_\pi(\eta_\theta)$ is a monotonically increasing function of $\theta$ when $\eta_\theta$ belongs to a canonical SPEF with parameter $\theta$, as is the mean. The

first formulation in 2 then gives that $c_\pi(\eta_\theta)$ is also monotonically increasing. Thus, the problem of identifying the best-(CVaR/VaR/mean-CVaR) arm is equivalent to identifying that with minimum mean. See Appendix A for details.

However, the ranking in mean and in CVaR can be very different in general. To see this, fix $\pi = 0.8$, and consider a 3-armed bandit instance, $\nu$, with $\nu_1 = 0.8\delta_0 + 0.2\delta_1$, $\nu_2 = 0.8\delta_0 + 0.2\delta_{0.5}$, and $\nu_3 = 0.8\delta_{-0.5} + 0.2\delta_2$. Clearly, $m(\nu_1) > m(\nu_2) > m(\nu_3)$, yet $c_\pi(\nu_2) < c_\pi(\nu_1) < c_\pi(\nu_3)$.

**General case:** For $\eta$ in class $\mathcal{L}$, the moment-constraint limits the minimum and maximum possible values of VaR and CVaR, as discussed in the following lemma (proof in Appendix B).

**Lemma 2.1.** *For $\eta \in \mathcal{L}$, $c_\pi(\eta) \in D$ and $x_\pi(\eta) \in C$, where*

$$D \triangleq \left[-B^{\frac{1}{1+\epsilon}}, \left(\frac{B}{1-\pi}\right)^{\frac{1}{1+\epsilon}}\right] \quad and \quad C \triangleq \left[-\left(\frac{B}{\pi}\right)^{\frac{1}{1+\epsilon}}, \left(\frac{B}{1-\pi}\right)^{\frac{1}{1+\epsilon}}\right].$$

**Topology of weak convergence and the Lévy metric:** Let $\phi$ be a bounded and continuous function on $\Re$, $\delta > 0$, and $x \in \Re$. Consider the topology on $\mathcal{P}(\Re)$, generated by the base sets of the form $\mathcal{U}(\phi, x, \delta) = \left\{\eta \in \mathcal{P}(\Re) : |\int_\Re \phi(y)d\eta(y) - x| < \delta\right\}$. Weak convergence of a sequence $\kappa_n$ to $\kappa$, denoted as $\kappa_n \overset{D}{\Longrightarrow} \kappa$, is convergence in this topology [see 23, Section D.2]. It is equivalent to convergence in the Lévy metric on $\mathcal{P}(\Re)$, (denoted by $d_L$), defined next (see, [9, Theorem 6.8], [23, Theorem D.8]). For $\eta, \kappa \in \mathcal{P}(\Re)$, $d_L(\eta, \kappa)$ equals $\inf\left\{\delta > 0 : F_\eta(x-\delta) - \delta \leq F_\kappa(x) \leq F_\eta(x+\delta) + \delta, \forall x \in \Re\right\}$. Additionally, the metric space $(\mathcal{P}(\Re), d_L)$ is complete and separable.

## 3 Lower bound

We consider $\delta$-correct algorithms for identifying the arm with minimum CVaR, acting on bandit problems in $\mathcal{M}$. While ensuring $\delta$-correctness, the aim is to minimize the sample complexity, i.e., expected number of samples generated by the algorithm before it terminates. As is well known, the $\delta$-correctness property imposes a lower bound on the sample complexity of such algorithms.

Let $\mu \in \mathcal{M}$ denote the given bandit problem. Henceforth, for ease of notation, we assume without loss of generality that the best-CVaR arm in $\mu$ is arm 1. Let $\Sigma_K$ denote the probability simplex in $\Re^K$, $\mathcal{A}_j$ denote the collection of all bandit problems in $\mathcal{M}$ which have arm $j$ as the best-CVaR arm, $\tau_\delta$ be the stopping time for the $\delta$-correct algorithm, $N_a(\tau)$ denote the number of times arm $a$ has been sampled by the algorithm, and for a set $S$, let $S^o$ denote its interior. It is easy to deduce using standard arguments (see, e.g., [40, Theorem 33.5]) that for a $\delta$-correct algorithm acting on $\mu \in \mathcal{A}_1$,

$$\mathbb{E}(\tau_\delta) \geq V(\mu)^{-1} \log \frac{1}{4\delta} \quad \text{where} \quad V(\mu) = \sup_{t \in \Sigma_K} \inf_{\nu \in \mathcal{A}_1^c} \sum_{a=1}^K t_a \text{KL}(\mu_a, \nu_a), \text{ and } \mathcal{A}_j^c = \mathcal{M} \setminus \mathcal{A}_j. \quad (4)$$

**Lemma 3.1.** *For $\mu \in \mathcal{A}_1$, the inner minimization problem in $V(\mu)$ equals*

$$\min_{j \neq 1} \inf_{x \leq y} \left\{t_1 \text{KL}_{\text{inf}}^{\text{U}}(\mu_1, y) + t_j \text{KL}_{\text{inf}}^{\text{L}}(\mu_j, x)\right\},$$

*and hence*

$$V(\mu) = \sup_{t \in \Sigma_K} \min_{j \neq 1} \inf_{x \leq y} \left\{t_1 \text{KL}_{\text{inf}}^{\text{U}}(\mu_1, y) + t_j \text{KL}_{\text{inf}}^{\text{L}}(\mu_j, x)\right\}. \quad (5)$$

Recall from (1) that the expressions in (4) and (5) above differ from those in the best-mean arm setting in that the functionals $\text{KL}_{\text{inf}}^{\text{L}}$ and $\text{KL}_{\text{inf}}^{\text{U}}$ here are defined instead with the CVaR constraints.

**Remark 3.1.** Without any restriction on arm distributions for the CVaR-problem, for $y \in \Re$ and $\eta \in \mathcal{L}$, $\text{KL}_{\text{inf}}^{\text{U}}(\eta, y) = 0$. This is essentially because $\eta$ can be perturbed in KL only slightly by shifting an arbitrarily small mass from the lower tail to the extreme right, so that the CVaR constraint is satisfied. Thus, without any restrictions, $V(\mu) = 0$ (see, [2, Lemma 1, Theorem 3] for similar results in selecting the arm with the largest mean setting). However, we later solve the VaR-problem without such assumptions, i.e., arm distributions are allowed to be arbitrary probability measures in $\Re$. The lower bound for the VaR problem is as in (4), with $\text{KL}_{\text{inf}}^{\text{L}}$ and $\text{KL}_{\text{inf}}^{\text{U}}$ in the representation in (5) defined with VaR constraints, instead.

A proof of Lemma 3.1 can be found in Appendix C.1. Let $t^* : \mathcal{M} \to 2^{\Sigma_K}$. In particular, for $\nu \in \mathcal{M}$, let $t^*(\nu)$ denote the set of maximizers in the $V(\nu)$ optimization problem in (5). A key nuance of our algorithm and the related analysis is that the vector of empirical distributions may not belong to the class $\mathcal{M}$. The algorithm first projects the empirical distribution to class $\mathcal{L}$, then solves for the optimal $t^*$ in (5) for the projected distributions, and samples the arms in proportion to the computed $t^*$ (Section 4). For appropriate choices of the projection maps, the following lemmas guarantee that as the empirical distributions converge to the actual arm-distributions (in the weak topology), the $t^*$ computed by the algorithm converge to the optimal weights corresponding to $\mu$.

**Lemma 3.2.** *$\mathcal{L}$ is a compact set in the topology of weak convergence and the (Skorokhod transforms of) its members form a uniformly integrable collection of random variables. When restricted to $\mathcal{L} \times D^o$, $\mathrm{KL}_{\mathrm{inf}}^{\mathrm{L}}$ and $\mathrm{KL}_{\mathrm{inf}}^{\mathrm{U}}$ are both jointly continuous functions of the arguments. Moreover, for fixed $x$, $\mathrm{KL}_{\mathrm{inf}}^{\mathrm{U}}(\nu, x)$ is a convex function of $\nu$.*

**Definition (Upper hemicontinuity)** A set-valued function $\Gamma : S \to T$ is upper hemicontinuous at $s \in S$ if for any open neighbourhood $V$ of $\Gamma(s)$ there exists a neighbourhood $U$ of $s$ such that for all $x \in U$, $\Gamma(x)$ is a subset of $V$.

**Lemma 3.3.** *$t^*$ is an upper-hemicontinuous correspondence. For $\nu \in \mathcal{M}^o$, $t^*(\nu)$ is a convex set.*

In Lemma 3.2, we restrict to the interior of $D$ as $\mathrm{KL}_{\mathrm{inf}}^{\mathrm{U}}(\cdot, B^{\frac{1}{1+\epsilon}}(1-\pi)^{\frac{-1}{1+\epsilon}})$ and $\mathrm{KL}_{\mathrm{inf}}^{\mathrm{L}}(\cdot, -B^{\frac{1}{1+\epsilon}})$ are not continuous (see, Remark C.2). In Lemma 3.3, we only need to eliminate distributions with these extreme CVaRs (there are only two such distributions. See, Remark C.1). Lemma 3.3 and Theorem 4.1 (optimality and $\delta$-correctness of the proposed algorithm) hold for distributions with CVaR in $D^o$. For ease of notation, we restrict $\mu$ to lie in the interior of $\mathcal{M}$.

The proofs of the above two lemmas are technically challenging and involve nuanced analysis. Detailed steps are given in Appendix C.2 and C.3. We first prove joint lower- and upper-semicontinuity of the KL-projection functionals separately. These rely on various properties of the weak convergence of probability measures in $\mathcal{L}$, the dual representations for $\mathrm{KL}_{\mathrm{inf}}^{\mathrm{L}}$ and $\mathrm{KL}_{\mathrm{inf}}^{\mathrm{U}}$ (see Theorem 3.4), properties of CVaR for probability measures in $\mathcal{L}$, and the classical Berge's theorem (see, [52]) for continuity of the optimal value and the set of optimizers for a parametric optimization problem. We then use these to prove the continuity in Lemma 3.3. Convexity follows since $t^*$ is the set of maximizers of a concave function over a convex, compact set.

**Understanding the lower bound:** Our proposed algorithm requires repeated evaluations of the lower bound in (4) at its estimates of $\mu$. To facilitate this, we now provide more tractable characterizations of the two KL-projection functionals, and in particular, of (5). We also discuss the statistical and computational implications of these alternative characterizations.

For $\eta \in \mathcal{P}(\Re)$, let $\mathrm{Supp}(\eta)$ denote the collection of points in the support of measure $\eta$. For $v \in D^o$, $x_0 \in C$, $\boldsymbol{\lambda} \in \Re^3$, $\boldsymbol{\gamma} \in \Re^2$, and $X \in \Re$, set

$$g^U(X, \boldsymbol{\lambda}, v) = 1 + \lambda_1 v - \lambda_2(1-\pi) + \lambda_3(|X|^{1+\epsilon} - B) - (\lambda_1 X (1-\pi)^{-1} - \lambda_2)_+,$$

and

$$g^L(X, \boldsymbol{\gamma}, v, x_0) = 1 - \gamma_1(v - x_0 - (X - x_0)_+ (1-\pi)^{-1}) - \gamma_2(B - |X|^{1+\epsilon}).$$

Furthermore, define $\hat{S}(v) = \left\{ \lambda_1 \geq 0, \lambda_2 \in \Re, \lambda_3 \geq 0 : \forall x \in \Re, \ g^U(x, \boldsymbol{\lambda}, v) \geq 0 \right\}$, and $\mathcal{R}_2(x_0, v)$ to be $\left\{ \gamma_1 \geq 0, \gamma_2 \geq 0 : \forall y \in \Re, \ g^L(y, \boldsymbol{\gamma}, v, x_0) \geq 0 \right\}$. Notice that these are convex sets.

As shown in Theorem 3.4 below, $g^U(y, \cdot, v)$ and $g^L(y, \cdot, v, x_0)$ are related to the dual formulations of $\mathrm{KL}_{\mathrm{inf}}^{\mathrm{U}}$ and $\mathrm{KL}_{\mathrm{inf}}^{\mathrm{L}}$, respectively, and the parameters $\boldsymbol{\lambda}$ and $\boldsymbol{\gamma}$ are the corresponding dual variables. The sets $\hat{S}$ and $\mathcal{R}_2$ correspond to the feasible values of these dual variables.

**Theorem 3.4.** *For $\eta \in \mathcal{P}(\Re)$ and $v \in D^o$,*

*(a)*

$$\mathrm{KL}_{\mathrm{inf}}^{\mathrm{U}}(\eta, v) = \max_{\boldsymbol{\lambda} \in \hat{S}(v)} \mathbb{E}_\eta \left( \log \left( g^U(X, \boldsymbol{\lambda}, v) \right) \right).$$

*The maximum in this expression is attained at a unique point $\lambda^* \in \hat{S}(v)$. The unique probability measure $\kappa^* \in \mathcal{L}$ that achieves infimum in the primal problem satisfies*

$$\frac{d\kappa^*}{d\eta}(y) = \frac{1}{g^U(y, \boldsymbol{\lambda^*}, v)}, \quad \text{for } y \in \mathrm{Supp}(\eta).$$

*Moreover, it has mass on at most 2 points outside* $\mathrm{Supp}(\eta)$. *Furthermore, for* $y' \in \{\mathrm{Supp}(\kappa^*) \setminus \mathrm{Supp}(\eta)\}$, $g^U(y', \boldsymbol{\lambda^*}, v) = 0$.

*(b)*

$$\mathrm{KL}^L_{\inf}(\eta, v) = \min_{x_0 \in [-\left(\frac{B}{\pi}\right)^{\frac{1}{1+\epsilon}}, v]} \max_{\boldsymbol{\gamma} \in \mathcal{R}_2(x_0, v)} \mathbb{E}_\eta \left( \log \left( g^L(X, \boldsymbol{\gamma}, v, x_0) \right) \right).$$

*For a fixed* $x_0$, *the maximum in the inner problem is attained at a unique* $\boldsymbol{\gamma}^*$ *in* $\mathcal{R}_2(x_0, v)$. *The unique probability measure* $\kappa^* \in \mathcal{L}$ *achieving infimum in the primal problem satisfies*

$$\frac{d\kappa^*}{d\eta}(y) = \frac{1}{g^L(y, \boldsymbol{\gamma^*}, v, x_0)}, \quad \text{for } y \in \mathrm{Supp}(\eta).$$

*Moreover, size of the set* $\{\mathrm{Supp}(\kappa^*) \setminus \mathrm{Supp}(\eta)\}$ *is at most 1, and for* $y' \in \{\mathrm{Supp}(\kappa^*) \setminus \mathrm{Supp}(\eta)\}$, $g^L(y'\boldsymbol{\gamma^*}, v, x_0) = 0$.

These dual formulations help in reformulating the lower bound optimization problem in (5) as optimization over reals. A computationally more efficient approach for this is to consider the joint dual of the inner optimization problem in (5).

For $\eta_1, \eta_2 \in \mathcal{P}(\Re)$, and non-negative weights $\alpha_1, \alpha_2$, let

$$Z = \inf_{x \leq y} \left\{ \alpha_1 \mathrm{KL}^U_{\inf}(\eta_1, y) + \alpha_2 \mathrm{KL}^L_{\inf}(\eta_2, x) \right\}. \tag{6}$$

For $y \in \Re$, $\boldsymbol{\lambda} \in \Re^2$, $\boldsymbol{\rho} \in \Re^2$, and $\boldsymbol{\gamma}_2 \in \Re$, let

$$h^L(y, \boldsymbol{\lambda}, \boldsymbol{\gamma}, \boldsymbol{\rho}, x_0) = 1 - \lambda_1 + \gamma_2(|y|^{1+\epsilon} - B) + \rho_1(x_0 + (y - x_0)_+(1 - \pi)^{-1}),$$

and

$$h^U(y, \boldsymbol{\lambda}, \boldsymbol{\gamma}, \boldsymbol{\rho}) = 1 + \lambda_1 + \lambda_2(|y|^{1+\epsilon} - B) - (\rho_2 + (\rho_1 y - \rho_2)_+(1 - \pi)^{-1}).$$

For $x_0 \in C$, define the convex region $\mathcal{D}_{x_0}$ to be collection of $\lambda_1 \in \Re$, $\rho_2 \in \Re$, $\lambda_2 \geq 0$, $\gamma_2 \geq 0$, and $\rho_1 \geq 0$, such that for all $y \in \Re$, $h^L(y, \boldsymbol{\lambda}, \boldsymbol{\gamma}, \boldsymbol{\rho}, x_0) \geq 0$ and $h^U(y, \boldsymbol{\lambda}, \boldsymbol{\gamma}, \boldsymbol{\rho}) \geq 0$. As we show next, these quantities are related to the dual formulation of (6).

**Proposition 3.5.** *For* $\eta_1, \eta_2 \in \mathcal{P}(\Re)$ *and weights* $\alpha_1, \alpha_2 \in [0, 1]$, $Z$ *equals*

$$\min_{x_0 \in C} \max_{(\boldsymbol{\lambda}, \boldsymbol{\gamma}, \boldsymbol{\rho}) \in \mathcal{D}_{x_0}} \alpha_1 \mathbb{E}_{\eta_1} \left( \log \left( h^U(X, \boldsymbol{\lambda}, \boldsymbol{\gamma}, \boldsymbol{\rho}) \right) \right) + \alpha_2 \mathbb{E}_{\eta_2} \left( \log \left( h^L(X, \boldsymbol{\lambda}, \boldsymbol{\gamma}, \boldsymbol{\rho}, x_0) \right) \right)$$
$$- \alpha_1 \log \alpha_1 - \alpha_2 \log \alpha_2 + (\alpha_1 + \alpha_2) \log (\alpha_1 + \alpha_2) - (\alpha_1 + \alpha_2) \log 2.$$

An application of above to the empirical distributions ($\eta_a = \hat{\mu}_a(t)$) weighted by sample counts, i.e., $\alpha_a = N_a(t)$, for $a \in \{1, 2\}$, results in unweighted sums over the samples. Also observe that the representation in Proposition 3.5 is 1 dimension smaller compared to that obtained by using Theorem 3.4 in (6), and hence, is faster to optimize numerically.

Recall that while $\mathrm{KL}^U_{\inf}$ is a convex optimization problem, $\mathrm{KL}^L_{\inf}$ is not (see Section 2). To handle this, we use the min formulation for CVaR in 2 to turn it into a one-dimensional family of linear constraints, which appears as the outer $\min_{x_0}$ in the expression in (b) in Theorem 3.4, and Proposition 3.5 above, with the range constraint from Lemma 2.1. This renders the remanining problem as a convex optimization problem. To simplify the CVaR constraint in $\mathrm{KL}^U_{\inf}$, we use (3). Rest is the Lagrangian duality. Complete proofs for Theorem 3.4, and Proposition 3.5, are given in Appendix D.

The equality in Proposition 3.5, and the dual formulations in Theorem 3.4, are important statistically and computationally. First, our stopping rule will threshold the $Z$ statistics to determine when to safely stop. So we need to bound the deviations of $Z$. For this, we will use the dual formulations from Theorem 3.4 in (6) to construct mixtures of super-martingales that dominate the deviations of $Z$. Second, our sampling rule will sample according to the optimal proportions evaluated for the empirical distribution vector, $\hat{\mu}(t)$, in (5). For this, we use Proposition 3.5 in our experiments to solve the inner optimization problem in (5). These will be made precise in Section 4.

Computing a gradient for the objective of the maximisation problem (5), seen as a function of the sampling weights $t$, takes one $Z$ evaluation per suboptimal arm. The inner maximisation over $\mathcal{D}_{x_0}$ is a constrained concave program, for which standard algorithms apply. The outer $\min_{x_0}$ problem requires a different approach, as it is not even quasiconvex. Empirically it does become quasiconvex after seeing enough samples, so we employ a heuristic bisection search for which we measure the impact on the error probability (there is none). Numerical results are presented in Section 4.4.

# 4 The algorithm

Given a bandit problem $\mu \in \mathcal{M}$, our algorithm is a specification of three things: a sampling rule, a stopping rule, and a recommendation rule.

**Sampling rule:** At each iteration, the algorithm has access to the empirical distribution vector, $\hat{\mu}(n)$. It first projects $\hat{\mu}(n)$ to $\mathcal{L}^K$ in the Kolmogorov metric, $d_K$, using the projection map, $\Pi$, defined below. It then computes $t^*(\Pi(\hat{\mu}(n)))$ and allocates samples using the C-tracking rule of [27], which we state in Appendix E for completeness. The map $\Pi = (\tilde{\Pi}, \ldots, \tilde{\Pi})$, where $\tilde{\Pi} : \mathcal{P}(\Re) \to \mathcal{L}$, is given by

$$\tilde{\Pi}(\eta) \in \operatorname{argmin}_{\kappa \in \mathcal{L}} \ d_K(\eta, \kappa), \quad \text{where} \quad d_K(\eta, \kappa) := \sup_{x \in \Re} \ |F_\eta(x) - F_\kappa(x)| .$$

We show in Appendix G that this projection has a simple form and can be computed easily.

**Stopping rule:** We use a modification of the generalized likelihood ratio test (GLRT) (see, [17]) as our stopping criterion. At any time, the vector of empirical distributions, $\hat{\mu}(n)$, suggests an arm with minimum CVaR (empirically best-CVaR arm), say arm $i$. This is our null hypothesis, which we test against all the alternatives. Formally, the log of the GLRT statistic, denoted by $S_i(n)$, is $\inf_{\nu' \in \mathcal{A}_i^c} \ \sum_{a=1}^K N_a(n) \operatorname{KL}(\hat{\mu}_a(n), \nu'_a)$. This is exactly the scaled inner optimization problem in the expression of $V(\hat{\mu}(n))$ in (4), except that $\hat{\mu}(n)$ may not belong to $\mathcal{M}$ (recall the $Z$ statistic defined in (6)). Let $Z_i(n)$ equal $\min_{a \neq i} \ \inf_{x \leq y} \ \{N_i(n) \operatorname{KL}_{\text{inf}}^{\text{U}}(\hat{\mu}_i(n), y) + N_a(n) \operatorname{KL}_{\text{inf}}^{\text{L}}(\hat{\mu}_a(n), x)\}$. It equals $S_i(n)$ when $\hat{\mu}(n) \in \mathcal{M}$ (Lemma 3.1). Our stopping rule corresponds to checking

$$Z_i(n) \geq \beta(n, \delta) \quad \text{where} \quad \beta(n, \delta) = \log\left((K-1)\delta^{-1}\right) + 5\log(n+1) + 2. \tag{7}$$

**Recommendation rule:** After stopping, the algorithm outputs the arm with the minimum CVaR of the corresponding empirical distribution, i.e., if $\tau$ is the stopping time of the algorithm, then it outputs $\operatorname{argmin}_a \ c_\pi(\hat{\mu}_a(\tau))$.

## 4.1 Theoretical guarantees

For a given confidence $\delta$, let $\tau_\delta$ denote the stopping time for the algorithm. The algorithm makes an error if at time $\tau_\delta$, there is an arm $j \neq 1$ such that $c_\pi(\hat{\mu}_j(\tau_\delta)) < c_\pi(\hat{\mu}_1(\tau_\delta))$. Let the error event be denoted by $\mathcal{E}$.

**Theorem 4.1.** *For $\delta > 0$ and $\mu \in \mathcal{M}^o$, the proposed algorithm with $\beta(t, \delta)$ chosen as in (7), satisfies*

$$\mathbb{P}(\mathcal{E}) \leq \delta \quad \text{and} \quad \limsup_{\delta \to 0} \ \frac{\mathbb{E}_\mu(\tau_\delta)}{\log(1/\delta)} \leq \frac{1}{V(\mu)}.$$

We first sketch the proof for the $\delta$-correctness part of the theorem. Proof ideas for sample complexity are presented later in this section. The detailed proof for Theorem 4.1 can be found in Appendix F.

$\delta$**-correctness:** Recall that the algorithm makes an error if at time $\tau_\delta$, the empirically best-CVaR arm is not arm 1. As in the best-mean arm case, it can be argued that this probability is at most

$$\sum_{i=2}^K \mathbb{P}\left(\exists n : \ N_i(n) \operatorname{KL}_{\text{inf}}^{\text{U}}(\hat{\mu}_i(n), c_\pi(\mu_i)) + N_1(n) \operatorname{KL}_{\text{inf}}^{\text{L}}(\hat{\mu}_1(n), c_\pi(\mu_1)) \geq \beta\right). \tag{8}$$

See Appendix F for a proof of (8). The following proposition will be helpful in bounding each of the summands above. Setting $j = 1$, and $x = \log \frac{K-1}{\delta}$ in Proposition 4.2, along with $\beta$ from (7), we get that each summand in (8) is at most $\delta/(K-1)$, proving that the proposed algorithm is $\delta$-correct.

**Proposition 4.2.** *For $i \in [K], j \in [K], i \neq j, h(n) = 5\log(n+1) + 2$, and $x \geq 0$,*

$$\mathbb{P}\left(\exists n : N_i(n) \operatorname{KL}_{\text{inf}}^{\text{U}}(\hat{\mu}_i(n), c_\pi(\mu_i)) + N_j(n) \operatorname{KL}_{\text{inf}}^{\text{L}}(\hat{\mu}_j(n), c_\pi(\mu_j)) - h(n) \geq x\right) \leq e^{-x}.$$

A key step in proving Proposition 4.2 is constructing mixtures of super-martingales that dominate the exponentials of $N_i(n) \operatorname{KL}_{\text{inf}}^{\text{U}}(\hat{\mu}_i(n), c_\pi(\mu_i))$ and $N_i(n) \operatorname{KL}_{\text{inf}}^{\text{L}}(\hat{\mu}_i(n), c_\pi(\mu_i))$. From Theorem 3.4(a) and (2), it can be shown that for fixed dual-variables, the objective is a sum of logs of random variables with mean at-most 1. Hence, its exponential is a non-negative candidate super-martingale. Since we want to bound the maximum over the dual parameters, we construct a mixture of these candidates, over the dual-parameters, and show that it dominates the exponential of $N_i(n) \operatorname{KL}_{\text{inf}}^{\text{U}}(\hat{\mu}_i(n), c_\pi(\mu_i))$.

**Sample complexity:** Our sample complexity proof follows that of [27] for a parametric family. However, we work with a more general non-parametric class, in which we establish continuity of the KL-projection functionals (Lemma 3.2). Our proof also differs from that in [2] in that we only have upper-hemicontinuity of the set of optimal sampling allocations ($t^*$) (Lemma 3.3). A nuance in our analysis is that the empirical distribution may not belong to the class $\mathcal{L}$, in which case we project the empirical distribution onto that class, and the sampling rule uses this projected distribution to compute $t^*$. Our careful choice of the projection map aids in the proof of this result.

**Computational complexity:** The computational cost of these KL-projection functionals, and hence, that of the oracle weights, is linear in the number of samples taken (see, [2, 16, 30]). As a result, the overall run-time is quadratic in $\tau_\delta$. We propose a modification in which we update the weights only at geometrically spaced times. This modification improves the computational-cost to almost linear in $\tau_\delta$, while its sample complexity is optimal up to a multiplicative constant depending on the choice of geometrical-spacing factor, thus providing a controlled trade-off between the two costs. Recently, [3] propose a similar yet different "doubling" trick in the regret-minimization setting. However, our approach differs from theirs in that our update-times for weights are not random. We refer the reader to Appendix K for details of the algorithm and proofs for its theoretical guarantees.

**Mean-CVaR problem:** We now extend the methodology for the CVaR problem to the more general mean-CVaR problem. For a distribution $\eta \in \mathcal{L}$ (for example, a random loss in a financial investment), the metric associated with the "badness" of a distribution is $\alpha_1 m(\eta) + \alpha_2 c_\pi(\eta)$, for $\alpha_1 > 0$ and $\alpha_2 > 0$, and the best-arm is the one with minimum value of this conic combination of mean and CVaR. For $\alpha_1 = 0$ this is the CVaR-problem, which we have studied in this work. For $\alpha_2 = 0$, this is the mean-problem, extensively studied in [2, 27].

As in (1), we can define corresponding KL-projection functionals, with the CVaR constraints replaced with those on the modified metric. The above theory, with this updated $\mathrm{KL}_{\mathrm{inf}}^{\mathrm{L}}$ and $\mathrm{KL}_{\mathrm{inf}}^{\mathrm{U}}$, gives the corresponding results for this setting. In particular, the lower bound on $\mathbb{E}(\tau_\delta)$ for $\delta$-correct algorithms for mean-CVaR BAI is given by $V(\mu)^{-1} \log \frac{1}{4\delta}$, where $V(\mu)$ is defined in (5) with the updated $\mathrm{KL}_{\mathrm{inf}}^{\mathrm{U}}$ and $\mathrm{KL}_{\mathrm{inf}}^{\mathrm{L}}$.

**Theorem 4.3** (Informal). *For $\mu \in \mathcal{M}^o$, the proposed algorithm for CVaR, with $\mathrm{KL}_{\mathrm{inf}}^{\mathrm{U}}$ and $\mathrm{KL}_{\mathrm{inf}}^{\mathrm{L}}$ defined with mean-CVaR constraints instead, is $\delta$-correct and asymptotically optimal.*

The proof of this theorem parallels that for CVaR above. We give the formal statement with proof-details in Appendix I.

## 4.2 The VaR problem

In this section we present the main ideas for an analogous approach for the optimum VaR-problem. Here, we will not impose any conditions (viz. membership in $\mathcal{L}$) on the arm-distributions, as the VaR lower bound is defined without it, i.e., arm distributions are allowed to be arbitrary probability measures on $\Re$. For a probability measure $\eta$, let $F_\eta(y)$, denote its CDF evaluated at $y$ and $F_\eta^-(y) = \lim_{z\uparrow y} F_\eta(z)$ denote the left limit of the CDF. Moreover, for $r, q \in (0, 1)$ let $d_2(r, q)$ denote the KL divergence between the Bernoulli random variables with mean $r$ and $q$. For $y \in \Re$, let $\mathrm{KL}_{\mathrm{inf}}^{\mathrm{L}}(\eta, y)$ and $\mathrm{KL}_{\mathrm{inf}}^{\mathrm{U}}(\eta, y)$ be defined as in (1), with VaR constraints, instead. These simplify as follows.

**Lemma 4.4.** $\mathrm{KL}_{\mathrm{inf}}^{\mathrm{L}}(\eta, y) = d_2(\min\{F_\eta(y), \pi\}, \pi)$ *and* $\mathrm{KL}_{\mathrm{inf}}^{\mathrm{U}}(\eta, y) = d_2(\max\{F_\eta^-(y), \pi\}, \pi)$.

Unlike in the CVaR-problem, we show that $\mathrm{KL}_{\mathrm{inf}}^{\mathrm{L}}$ and $\mathrm{KL}_{\mathrm{inf}}^{\mathrm{U}}$ for the VaR problem are not jointly continuous functionals (see Remark H.2). The discontinuity occurs at $y$ being the jump points of $F_\eta$ in Lemma 4.4 above. However, we prove in Appendix H (Corollary H.3.1) that the set of optimal proportions, $t^*$, is still upper-hemicontinuous and convex.

The algorithm for CVaR with $\mathrm{KL}_{\mathrm{inf}}^{\mathrm{U}}$ and $\mathrm{KL}_{\mathrm{inf}}^{\mathrm{L}}$ replaced by those in the lemma above, and setting

$$\beta(t, \delta) = 6 \log\left(1 + \log\frac{t}{2}\right) + \log\frac{K-1}{\delta} + 8 \log\left(1 + \log\frac{K-1}{\delta}\right),$$

we get our algorithm for the VaR-problem.

**Theorem 4.5** (Informal). *The proposed algorithm for the VaR-problem is $\delta$-correct and asymptotically optimal.*

We refer the reader to Appendix H for a detailed discussion of the VaR-problem and proofs.

### 4.3 Tight $\mathrm{KL}_{\inf}$-based confidence intervals for CVaR

We now present tight anytime-valid confidence interval for the CVaR of a distribution in $\mathcal{L}$. Let $\hat{\eta}_n$ denote the empirical distribution corresponding to $n$ samples from $\eta \in \mathcal{L}$. Our proposed upper $(U_n)$ and lower $(L_n)$ confidence intervals for $c_\pi(\eta)$ are of the form $U_n = \max\left\{x \in \Re: \ n\,\mathrm{KL}_{\inf}^{\mathrm{U}}(\hat{\eta}_n, x) \leq C\right\}$ and $L_n = \min\left\{x \in \Re: \ n\,\mathrm{KL}_{\inf}^{\mathrm{L}}(\hat{\eta}_n, x) \leq C\right\}$, for an appropriately chosen threshold $C \approx \log \delta^{-1} + 3\log n$. Similar confidence intervals for the mean of heavy-tailed distributions were proposed in [3]. Let $\hat{x}_{\pi,n}$ denote the $\pi^{th}$ quantile for $\hat{\eta}_n$. Recall that the popular truncation-based estimator for $c_\pi(\eta)$ is given by $\hat{c}_{\pi,n} = n^{-1}(1-\pi)^{-1}\sum_i X_i \mathbb{1}(\hat{x}_{\pi,n} \leq X_i \leq u_n)$, for appropriately chosen truncation levels, $u_n$ (see, [39]). Observe that there are 2 sources of error in this estimator, first, the estimation of the quantile, and second, the estimation of the tail-expectation. On the other hand, our confidence intervals do not rely on estimation of the true quantile, $x_\pi$. In Appendix J, we show that even given the correct estimation of $\hat{x}_{\pi,n}$, confidence intervals for $\hat{c}_{\pi,n}$ perform poorly compared to those based on $\mathrm{KL}_{\inf}^{\mathrm{U}}$ and $\mathrm{KL}_{\inf}^{\mathrm{L}}$, in some applications.

### 4.4 Numerical Results

This is only a brief teaser section on the experiments, which are detailed in Appendix L. We are interested in the question whether the asymptotic sample complexity result of Theorem 4.1 is representative at reasonable confidence levels $\delta$. Whether this is the case or not differs greatly between pure exploration setups: [27] see state-of-the-art numerical results in Bernoulli arms for Track-and-Stop with $\delta = 0.1$, while [22] present a Minimum Threshold problem instance where the Track-and-Stop asymptotics have not kicked in yet at $\delta = 10^{-20}$. Our experiments confirm that our approach is indeed practical at moderate confidence $\delta$.

In our experiments we implement a version of Track-and-Stop including C-tracking and forced exploration and apply it to Fisher-Tippett $(F(\mu, \sigma, \gamma))$, Pareto $(P(\mu, \sigma, \gamma))$, and mixtures of Fisher-Tippett arms (these heavy-tailed distributions arise in extreme value theory).

Figure 1 shows the distribution of the stopping time as a function of $\delta$ in a synthetic three-arm task: arm 1 is a uniform mixture of $F(-1, 0.5, 0.4)$ and $F(-3, 0.5, -0.4)$, arm 2 is $P(0, 0.2, 0.55)$ and arm 3 is $F(-0.5, 1, 0.1)$ with respective CVaRs at quantile $\pi = 0.6$ being $-0.1428$, $0.974$ and $1.547$. We select $\epsilon = 0.7$ and $B = 4.5$. This is a moderately hard problem of complexity $V^{-1}(\mu) = 49.7$. We conclude that even at moderate $\delta$ the average sample complexity closely matches the lower bound, especially after adjusting it for the lower-order terms in the employed stopping threshold $\beta(n, \delta)$. This demonstrates that our asymptotic optimality is in fact indicative of the performance in practice.

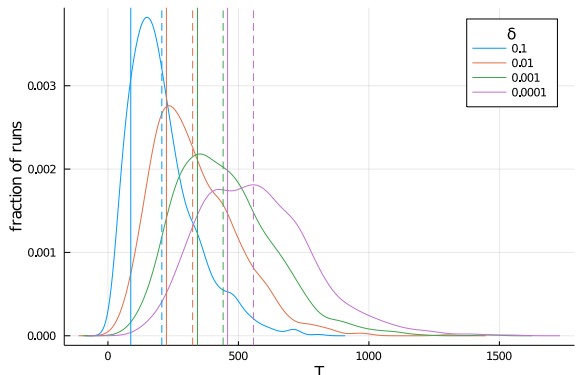

Figure 1: Histogram of stopping times among 1000 runs on 3 arms, as a function of confidence $\delta$. Vertical bars (solid) indicate the lower bound (4), and (dashed) a version adjusted to our stopping threshold (7), i.e., the $n$ that solves $n = \beta(n, \delta)V(\mu)^{-1}$.

We do additional experiments to show the dependence of the algorithm's performance on the number of arms and the input parameter $B$. We see that the average stopping time of our algorithm increases linearly in the number of arms. Moreover, the sample complexity is sensitive to $B$, indicating the importance of correctly estimating it. We refer to Appendix L for details of these experiments.

**Conclusion:** We developed asymptotically optimal algorithms that identify the arm with the minimum risk, measured in terms of CVaR, VaR, or a conic combination of mean and CVaR. Our algorithms operate in non-parametric settings with possibly heavy-tailed distributions. Although similar plug-and-play algorithms have been developed in simpler settings, our algorithms for tail-risk measures require more nuanced analysis. The techniques developed may be generalizable to a much broader class of problems.

## Acknowledgments and Disclosure of Funding

We acknowledge the support of the Department of Atomic Energy, Government of India, to TIFR under project no. RTI4001.

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
