## A    Equivalence in canonical SPEF setting

In this section, we will show that $x_\pi(\eta_\theta)$ and $c_\pi(\eta_\theta)$ are monotonic functions of $\theta$ when $\eta_\theta$ belongs to a canonical SPEF with parameter $\theta$, as is the mean. Thus, the problem of identifying the best-(CVaR/VaR/mean-CVaR) arm in this setting, is equivalent to identifying the arm with minimum mean.

Let $\nu$ be the reference measure on $\Re$ for the SPEF, and $\Theta \subset \Re$ be the parameter space, i.e., $\Theta = \{\theta \in \Re : \int_\Re \exp(\theta y)\, d\nu(y) < \infty\}$. Then, for $\eta_\theta$ in the SPEF, for $y \in \Re$, $(d\eta_\theta/d\nu)(y) = \exp\{\theta y - A(\theta)\}$, where $A(\theta)$ is the normalizing factor. By direct computation, it can be verified that $A'(\theta)$ equals the mean, and $A''(\theta)$ equals the variance of $\eta_\theta$. Hence, mean is an increasing function of $\theta$ ([16]).

Next, for $a \in \Re$, define $\bar{F}_\theta(a) = \int_{y \geq a} d\eta_\theta(y)$ to be the tail-CDF of $\eta_\theta$. Clearly, $\bar{F}_\theta(a)$ increases on increasing $\theta$ since $d\bar{F}_\theta(a)/d\theta = \bar{F}_\theta(a)\left(\mathbb{E}_{\eta_\theta}(X|X \geq a) - m(\eta_\theta)\right)$, which is positive. The fact that $x_\pi(\eta_\theta)$ is a non-decreasing function of $\theta$ now follows from its definition. From (2), we see that $c_\pi(\eta_\theta)$ is non-decreasing in $\theta$.

## B    Bounds on $\mathrm{CVaR}$ and $\mathrm{VaR}$ for distributions in $\mathcal{L}$: Proof of Lemma 2.1

We first recall the definitions and different representations of CVaR, which will be useful in this section. Given a probability measure $\kappa$, let $x_\pi(\kappa)$ and $c_\pi(\kappa)$ denote its VaR and CVaR at level $\pi$. Then, recall that

$$c_\pi(\kappa) = \frac{F(x_\pi(\kappa)) - \pi}{1 - \pi} x_\pi(\kappa) + \frac{1}{1 - \pi} \int_\Re (y - x_\pi(\kappa))_+ \, d\kappa(y)$$

$$= \min_{z \in \Re} \left\{ z + \frac{1}{1 - \pi}\mathbb{E}_\kappa\left((X - z)_+\right) \right\},$$

where the infimum of the set of minimizers in the second representation, is VaR at level $\pi$ for $\kappa$.

Consider a probability measure $\eta \in \mathcal{L}$. Recall that for $\epsilon > 0$, $\mathcal{L} = \{\eta \in \mathcal{P}(\Re) : \mathbb{E}_\eta(f(X) \leq B)\}$, where $f(y) = |y|^{1+\epsilon}$, and

$$C = \left[-f^{-1}(B\pi^{-1}), f^{-1}(B(1-\pi)^{-1})\right] \quad \text{and} \quad D = \left[-f^{-1}(B), f^{-1}(B(1-\pi)^{-1})\right].$$

Let $x_\pi^-(\eta) < x_\pi(\eta)$. Then for $x_\pi(\eta) < 0$,

$$\pi \leq \int_{-\infty}^{x_\pi^-(\eta)} d\eta(y) = \int_{-\infty}^{x_\pi^-(\eta)} \frac{f(y)}{f(y)} d\eta(y) \leq \int_{-\infty}^{x_\pi^-(\eta)} \frac{f(y)}{f(x_\pi(\eta))} d\eta(y) \leq \frac{B}{f(x_\pi(\eta))},$$

and for $x_\pi(\eta) \geq 0$,

$$1 - \pi \leq \int_{x_\pi^-(\eta)}^\infty d\eta(y) = \int_{x_\pi^-(\eta)}^\infty \frac{f(y)}{f(y)} d\eta(y) \leq \int_{x_\pi^-(\eta)}^\infty \frac{f(y)}{f(x_\pi(\eta))} d\eta(y) \leq \frac{B}{f(x_\pi(\eta))}.$$

Combining the two, we get $-f^{-1}(B\pi^{-1}) \leq x_\pi(\eta) \leq f^{-1}(B(1-\pi)^{-1})$, where $f^{-1}(c)$ is defined as $\max\{y : f(y) = c\}$, which equals $c^{\frac{1}{1+\epsilon}}$. To get a bound on $c_\pi(\eta)$, consider the following inequalities.

$$B \geq \mathbb{E}_\eta(f(X)) \geq (F(x_\pi(\eta)) - \pi)f(x_\pi(\eta)) + \int_{x_\pi(\eta)}^\infty f(y)\, d\eta(y)$$

$$= (1 - \pi)\left(\frac{F(x_\pi(\eta)) - \pi}{1 - \pi} f(x_\pi(\eta)) + \frac{1}{1 - \pi} \int_{x_\pi(\eta)}^\infty f(y)\, d\eta(y)\right),$$

where the first inequality follows since $\eta$ is in $\mathcal{L}$, and the second follows since $f$ is non-negative. Furthermore, since $f$ is convex, using conditional Jensen's inequality, the above can be bounded from below by

$$(1 - \pi)f\left(\frac{F(x_\pi(\eta)) - \pi}{1 - \pi} x_\pi(\eta) + \frac{1}{1 - \pi} \int_{x_\pi(\eta)}^\infty y\, d\eta(y)\right),$$

which is $(1-\pi)f\left(c_\pi(\eta)\right)$. Thus we have, $-f^{-1}(B(1-\pi)^{-1}) \leq c_\pi(\eta) \leq f^{-1}(B(1-\pi)^{-1})$. However, the lower bound for $c_\pi(\eta)$ obtained above can be further tightened. Recall that

$$\min_{\eta \in \mathcal{L}} c_\pi(\eta) = \min_{z \in C} \min_{\eta \in \mathcal{L}} \left\{ z + \frac{1}{1-\pi} \mathbb{E}_\eta \left( (X-z)_+ \right) \right\}.$$

In the inner minimization problem in r.h.s. above, the objective is minimizing expectation under $\eta$ of convex functions of $X$, under the constraint that expectation under $\eta$ of a convex function being smaller that $B$. Thus, the minimizer concentrates at a single point, i.e., the minimizer $\eta = \delta_x$, for some $x \in \Re$ such that $f(x) \leq B$. Thus, the above problem equals

$$\min_{z \in C} \min_{x \in [-f^{-1}(B), f^{-1}(B)]} \left\{ z + \frac{1}{1-\pi}(x-z)_+ \right\},$$

which is increasing in $x$. Thus, at optimal $x = -f^{-1}(B)$, it equals

$$\min_{z \in C} \max \left\{ z, \frac{-f^{-1}(B) - \pi z}{1-\pi} \right\}.$$

Clearly, the minimum is attained at $z = -f^{-1}(B)$, with the optimal value being $-f^{-1}(B)$. Combining with the previous bounds on $c_\pi(\eta)$, we have that for $\eta \in \mathcal{L}$, $c_\pi(\eta) \in D$.

## C  Details of proofs in Section 3

We first review some notation that will be useful in this section. Recall that for a non-negative constant $B$, arm distributions belong to class $\mathcal{L}$ which equals $\{\eta \in \mathcal{P}(\Re) : \mathbb{E}_\eta(f(X)) \leq B\}$, where $f(x) = |x|^{1+\epsilon}$, for some $\epsilon > 0$. Define $f^{-1}(c) = \max\{y : f(y) = c\} = c^{\frac{1}{1+\epsilon}}$.

We denote by $\mathcal{M}$ the collection of all $K$-vectors of distributions, each belonging to $\mathcal{L}$ and by $\mathcal{A}_j$ the collection of vectors in $\mathcal{M}$ with arm $j$ having the minimum CVaR. Furthermore, for $\eta \in \mathcal{P}(\Re)$ and $\pi \in (0, 1)$, $c_\pi(\eta)$ and $x_\pi(\eta)$ denote the CVaR and VaR at confidence level $\pi$, for measure $\eta$. Also, for $x \in \Re$, we define the KL projection functionals

$$\mathrm{KL}^{\mathrm{U}}_{\inf}(\eta, x) := \inf_{\kappa \in \mathcal{L}: c_\pi(\kappa) \geq x} \mathrm{KL}(\eta, \kappa) \quad \text{and} \quad \mathrm{KL}^{\mathrm{L}}_{\inf}(\eta, x) = \inf_{\kappa \in \mathcal{L}: c_\pi(\kappa) \leq x} \mathrm{KL}(\eta, \kappa).$$

Furthermore, recall that

$$D = \left[-f^{-1}(B), f^{-1}\left(B(1-\pi)^{-1}\right)\right] \quad \text{and} \quad C = \left[-f^{-1}(B\pi^{-1}), f^{-1}\left(B(1-\pi)^{-1}\right)\right],$$

and $D^o$ and $C^o$ denote the interior of sets $D$ and $C$, respectively. For $v \in D^o$, $x_0 \in C$, $\boldsymbol{\lambda} \in \Re^3$, $\boldsymbol{\gamma} \in \Re^2$, and $X \in \Re$,

$$g^U(X, \boldsymbol{\lambda}, v) := 1 + \lambda_1 v - \lambda_2(1-\pi) + \lambda_3\left(f(X) - B\right) - \left(\frac{\lambda_1 X}{1-\pi} - \lambda_2\right)_+, \tag{9}$$

and

$$g^L(X, \boldsymbol{\gamma}, v, x_0) := 1 - \gamma_1\left(v - x_0 - \frac{(X-x_0)_+}{1-\pi}\right) - \gamma_2(B - f(X)). \tag{10}$$

Furthermore,

$$\hat{S}(v) := \left\{\lambda_1 \geq 0, \lambda_2 \in \Re, \lambda_3 \geq 0 : \forall x \in \Re, \ g^U(x, \boldsymbol{\lambda}, v) \geq 0\right\}, \tag{11}$$

and

$$\mathcal{R}_2(x_0, v) := \left\{\gamma_1 \geq 0, \gamma_2 \geq 0, \forall y \in \Re, \ g^L(y, \boldsymbol{\gamma}, x_0, v) \geq 0\right\}. \tag{12}$$

Later, in Theorem 3.4 we show that for $\eta \in \mathcal{P}(\Re)$,

$$\mathrm{KL}^{\mathrm{U}}_{\inf}(\eta, v) = \max_{\boldsymbol{\lambda} \in \hat{S}(v)} \mathbb{E}_\eta\left(\log\left(g^U(X, \boldsymbol{\lambda}, v)\right)\right)$$

and

$$\mathrm{KL}^{\mathrm{L}}_{\inf}(\eta, v) = \min_{x_0 \in C} \max_{\boldsymbol{\gamma} \in \mathcal{R}_2(x_0, v)} \mathbb{E}_\eta\left(\log\left(g^L(X, \boldsymbol{\gamma}, x_0, v)\right)\right).$$

## C.1 Proof of Lemma 3.1

Recall that arm 1 is the arm with minimum CVaR in $\mu$, and $V(\mu) = \sup_{t \in \Sigma_K} \inf_{\nu \in \mathcal{A}_1^c} \sum_{i=1}^K t_i \, \mathrm{KL}(\mu_i, \nu_i)$, where $\mathcal{A}_1^c = \mathcal{M} \setminus \mathcal{A}_1$. Clearly, the inner optimization problem satisfies

$$\inf_{\nu \in \mathcal{A}_1^c} \sum_{i=1}^K t_i \, \mathrm{KL}(\mu_i, \nu_i) = \min_{j \neq 1} \inf_{\nu \in \mathcal{A}_j} \sum_{i=1}^K t_i \, \mathrm{KL}(\mu_i, \nu_i). \tag{13}$$

Next, for $\mu \in \mathcal{M}$ the infimum in the expression in r.h.s. above is attained by $\nu \in \mathcal{A}_j$ such that $\nu_i = \mu_i$ for all arms $i$ not in $\{1, j\}$, as otherwise, the value of the summation can be decreased by setting them equal to $\mu_i$. Thus,

$$\inf_{\nu \in \mathcal{A}_j} \sum_{i=1}^K t_i \, \mathrm{KL}(\mu_i, \nu_i) = \inf_{\substack{\nu_1, \nu_j \in \mathcal{L}, \, x \leq y, \\ c_\pi(\nu_j) \leq x, \, c_\pi(\nu_1) \geq y}} \left\{ t_1 \, \mathrm{KL}(\mu_1, \nu_1) + t_j \, \mathrm{KL}(\mu_j, \nu_j) \right\}.$$

Now, from the definition of $\mathrm{KL}_{\mathrm{inf}}^{\mathrm{L}}$ and $\mathrm{KL}_{\mathrm{inf}}^{\mathrm{U}}$, the r.h.s. in above equation equals

$$\inf_{x \leq y} \left\{ t_1 \, \mathrm{KL}_{\mathrm{inf}}^{\mathrm{U}}(\mu_1, y) + t_j \, \mathrm{KL}_{\mathrm{inf}}^{\mathrm{L}}(\mu_j, x) \right\}.$$

Combining this with (13) gives the desired result. □

## C.2 Towards proving Lemma 3.2: Continuity of the KL-projection functionals

We first establish the properties of $\mathcal{L}$ stated in the Lemma.

**Uniform integrability of $\mathcal{L}$:** Since each probability measure $\eta$ in $\mathcal{L}$ has a uniformly bounded $p^{th}$ moment for a fixed $p > 1$, their Skorokhod transforms $\{u \mapsto F_\eta^{-1}(u) \mid \eta \in \mathcal{L}\}$ form a uniformly integrable collection ([57]).

**Compactness of $\mathcal{L}$:** It is sufficient to show that $\mathcal{L}$ is closed and tight. Prohorov's Theorem then gives that it is a compact set in the topology of weak convergence ([9]). We first show that it is a closed set. Towards this, consider a sequence $\eta_n$ of probability measures in $\mathcal{L}$, converging weakly to $\eta \in \mathcal{P}(\Re)$. By Skorohod's Representation Theorem (see, [9]), there exist random variables $Y_n, Y$ defined on a common probability space, say $(\Omega, \mathcal{F}, q)$, such that $Y_n \sim \eta_n$, $Y \sim \eta$, and $Y_n \xrightarrow{a.s.} Y$. Then, by Fatou's Lemma,

$$\mathbb{E}_\eta \left( |X|^{1+\epsilon} \right) = \mathbb{E}_q \left( |Y|^{1+\epsilon} \right) = \mathbb{E}_q \left( \liminf_{n \to \infty} |Y_n|^{1+\epsilon} \right) \leq \liminf_{n \to \infty} \mathbb{E}_q \left( |Y_n|^{1+\epsilon} \right) \leq B.$$

Hence, $\eta$ is in $\mathcal{L}$ and the class is closed in the weak topology. To see that it is tight, consider $K_\epsilon := \left[ -\left( 2B\epsilon^{-1} \right)^{\frac{1}{1+\epsilon}}, \left( 2B\epsilon^{-1} \right)^{\frac{1}{1+\epsilon}} \right]$. For $\eta \in \mathcal{L}$, $\eta(K_\epsilon^c) \leq \epsilon$.

**Convexity of $\mathrm{KL}_{\mathrm{inf}}^{\mathrm{U}}(., x)$:** Consider two measures, $\eta_1, \eta_2$ in $\mathcal{L}$, and let $\lambda \in (0, 1)$. Let $\kappa_1$ be such that $\mathrm{KL}_{\mathrm{inf}}^{\mathrm{U}}(\eta_1, x) = \mathrm{KL}(\eta_1, \kappa_1)$. Existence of $\kappa_1$ is guaranteed by continuity of $\mathrm{KL}_{\mathrm{inf}}^{\mathrm{U}}$ in its arguments, and compactness of the domain of optimization, which follows from Lemma C.2 below. Similarly, let $\kappa_2$ satisfy $\mathrm{KL}_{\mathrm{inf}}(\eta_2, x) = \mathrm{KL}(\eta_2, \kappa_2)$. Let

$$\eta_{12} = \lambda \eta_1 + (1 - \lambda)\eta_2, \quad \text{and} \quad \kappa_{12} = \lambda \kappa_1 + (1 - \lambda)\kappa_2.$$

Clearly, $\kappa_{12}$ is in $\mathcal{L}$. Moreover, by concavity of $c_\pi(\cdot)$, $c_\pi(\kappa_{12}) \geq \lambda c_\pi(\kappa_1) + (1 - \lambda)c_\pi(\kappa_2)$. Then, $\mathrm{KL}_{\mathrm{inf}}^{\mathrm{U}}(\eta_{12}, x)$ is at most $\mathrm{KL}(\eta_{12}, \kappa_{12})$, which, by joint convexity of KL, is bounded by $\lambda \, \mathrm{KL}(\eta_1, \kappa_1) + (1 - \lambda) \, \mathrm{KL}(\eta_2, \kappa_2)$. This bound then equals $\lambda \, \mathrm{KL}_{\mathrm{inf}}^{\mathrm{U}}(\eta_1, x) + (1 - \lambda) \, \mathrm{KL}_{\mathrm{inf}}(\eta_2, x)$.

**Joint continuity of $\mathrm{KL}_{\mathrm{inf}}^{\mathrm{U}}$ and $\mathrm{KL}_{\mathrm{inf}}^{\mathrm{L}}$:** We show upper- and lower-semicontinuity separately for the KL projection functionals restricted to $\mathcal{L}$ (see Lemmas C.3, C.4, and C.5). The following results will assist in the proofs of these.

**Lemma C.1.** *For $\eta_n$ and $\eta \in \mathcal{L}$, $c_\pi(\eta_n) \longrightarrow c_\pi(\eta)$ whenever $\eta_n \xRightarrow{D} \eta$.*

*Proof.* Consider a sequence $\eta_n \in \mathcal{L}$ weakly converging to $\eta \in \mathcal{L}$. Then, there exist random variables $Y_n, Y$ defined on a common probability space $(\Omega, \mathcal{F}, q)$ such that $Y_n \sim \eta_n$, $Y \sim \eta$, and $Y_n \xrightarrow{a.s.} Y$ (Skorohod's Theorem, see, [9]). Furthermore, since $\eta_n, \eta$ are uniformly integrable, $\mathbb{E}_q(|Y_n|) \to \mathbb{E}_q(|Y|)$ (see [57, Theorem 13.7])

Consider a sequence of real numbers $z_n \to z$. Then, $Y_n - z_n \xrightarrow{a.s.} Y - z$, whence $(Y_n - z_n)_+ \xrightarrow{a.s.} (Y - z)_+$. Clearly,

$$(Y_n - z_n)_+ \le |Y_n| + |z_n| \;, \quad |Y_n| + |z_n| \xrightarrow{a.s.} |Y| + |z| \quad \text{and} \quad \mathbb{E}_q(|Y_n|) + |z_n| \to \mathbb{E}_q(|Y|) + |z| < \infty.$$

Then, by generalized Dominated Convergence Theorem, $\mathbb{E}_q((Y_n - z_n)_+) \to \mathbb{E}_q((Y - z)_+)$. Now, for $\eta \in \mathcal{L}$, $c_\pi(\eta)$ equals

$$\min_{z \in C} g(z, \eta), \quad \text{where} \quad g(z, \eta) = z + \frac{1}{1 - \pi} \mathbb{E}_\eta((X - z)_+).$$

From the above discussion, $g(z, \eta)$ restricted to $C \times \mathcal{L}$, is a jointly continuous function. Berge's Theorem ([8, Maximum Theorem, Page 116]) then gives the desired result. □

**Lemma C.2.** *The sets $\mathcal{D}_v^L \triangleq \{\eta \in \mathcal{L} : c_\pi(\eta) \le v\}$ and $\mathcal{D}_v^U \triangleq \{\eta \in \mathcal{L} : c_\pi(\eta) \ge v\}$ are compact sets in the topology of weak convergence.*

*Proof.* Since $\mathcal{L}$ is compact, it is sufficient to show that the sets $\mathcal{D}_v^L$ and $\mathcal{D}_v^U$ are closed, which follows from Lemma C.1. □

**Lemma C.3.** *For $\eta \in \mathcal{P}(\Re)$ and $v \in D$, the functionals $\mathrm{KL}_{\mathrm{inf}}^U(\eta, v)$ and $\mathrm{KL}_{\mathrm{inf}}^L(\eta, v)$ are jointly lower-semicontinuous in $(\eta, v)$.*

*Proof.* Recall that

$$\mathrm{KL}_{\mathrm{inf}}^L(\eta, v) = \min_{\kappa \in \mathcal{L}: \, c_\pi(\kappa) \le v} \mathrm{KL}(\eta, \kappa) \quad \text{and} \quad \mathrm{KL}_{\mathrm{inf}}^U(\eta, v) = \min_{\kappa \in \mathcal{L}: \, c_\pi(\kappa) \ge v} \mathrm{KL}(\eta, \kappa).$$

For $\eta, \kappa \in \mathcal{P}(\Re)$, $\mathrm{KL}(\eta, \kappa)$ is jointly lower-semicontinuous function in the topology of weak convergence (see, [45]) and a jointly lower-semicontinuous function of $(\eta, \kappa, v)$. Let $D_v = \{\kappa \in \mathcal{L} : c_\pi(\kappa) \le v\}$. Since $D_v$ is a compact set for each $v$ (Lemma C.2), it is sufficient to show that $D_v$ is an upper-hemicontinuous correspondence (see, [8, Theorem 1, Page 115]).

Consider a sequence $v_n$ in $D$, converging to $\tilde{v}$ in $D$. Let $\eta_n \in D_{v_n}$, which exist since $D_{v_n}$ are non-empty sets. Since $\mathcal{L}$ is a tight, and hence relatively compact collection of probability measures, and $\eta_n \in \mathcal{L}$, $\eta_n$ has a weakly convergent sub-sequence, say $\eta_{n_i}$ converging to $\eta \in \mathcal{L}$ (since $\mathcal{L}$ is also closed). Furthermore, $c_\pi(\eta_{n_i}) \le v_{n_i}$. From Lemma C.1, $c_\pi(\eta) = \lim_{n_i} c_\pi(\eta_{n_i}) \le \tilde{v}$, which implies that $\eta \in D_{\tilde{v}}$, proving upper-hemicontinuity of the set $D_v$ in $v$ (see, [52, Proposition 9.8] for sequential characterization of upper-hemicontinuity).

Similar arguments hold for $\mathrm{KL}_{\mathrm{inf}}^L(\cdot, \cdot)$. □

**Lemma C.4.** $\mathrm{KL}_{\mathrm{inf}}^L$, *viewed as a function from $\mathcal{L} \times \left(-f^{-1}(B), f^{-1}\left(\frac{B}{1-\pi}\right)\right]$, is a jointly upper-semicontinuous function.*

*Proof.* Let

$$\underline{D} = \left(-f^{-1}(B), f^{-1}\left(\frac{B}{1-\pi}\right)\right] \quad \text{and} \quad C_v = \left[-f^{-1}\left(\frac{B}{\pi}\right), v\right].$$

We prove in Theorem 3.4(b) that for $v \in \underline{D}$, $\mathrm{KL}_{\mathrm{inf}}^L(\eta, v) = \min_{x_0 \in C_v} h^*(x_0, v, \eta)$, where for $g^L$ and $\mathcal{R}_2$ defined in (10) and (12) above,

$$h^*(x_0, v, \eta) := \max_{\boldsymbol{\gamma} \in \mathcal{R}_2(x_0, v)} \mathbb{E}_\eta\left(\log g^L(X, \boldsymbol{\gamma}, x_0, v)\right).$$

**Joint upper-semicontinuity for $v > c_\pi(\eta)$:** Observe that for $\eta \in \mathcal{L}$ and $v \geq c_\pi(\eta)$, $\mathrm{KL}^{\mathrm{L}}_{\mathrm{inf}}(\eta, v) = 0$. Consider a sequence $(\eta_n, v_n)$ converging to $(\eta, v)$. Then, $\exists n_0$ such that for all $n \geq n_0$, $c_\pi(\eta_n) \leq v_n$. To see this, suppose not, i.e., for all $n$, $c_\pi(\eta_n) > v_n$. Taking limits, this gives $c_\pi(\eta) \geq v$, which is a contradiction. Thus, for $n \geq n_0$, $\mathrm{KL}^{\mathrm{L}}_{\mathrm{inf}}(\eta_n, v_n) = 0$, proving continuity in this case.

We next prove the joint upper-semicontinuity for $v < c_\pi(\eta)$, and handle the joint upper-semicontinuity at $(\eta, c_\pi(\eta))$ separately.

**Joint upper-semicontinuity for $v < c_\pi(\eta)$:** It can be argued that for $\eta \in \mathcal{L}$ and $v < c_\pi(\eta)$,

$$\mathrm{KL}^{\mathrm{L}}_{\mathrm{inf}}(\eta, v) = \min_{x_0 \in C_v \setminus \{v\}} h^*(x_0, v, \eta). \tag{14}$$

To see this, $v < c_\pi(\eta)$ implies that $\eta \notin \mathcal{P}(\mathrm{Supp}(-\infty, v])$ and from Lemma D.8 and the remark following it, $h^*(v, v, \eta) = \infty$, giving (14).

Clearly, $C_v \setminus \{v\}$ is a lower-hemicontinuous correspondence. To show that $\mathrm{KL}^{\mathrm{L}}_{\mathrm{inf}}$ is jointly upper-semicontinuous, it suffices to show that $h^*(x_0, v, \eta)$ is jointly upper-semicontinuous ([8, Theorem 1, Page 115]).

**Joint upper-semicontinuity of $h^*$:** It follows from the definition that $\mathcal{R}_2(x_0, v) \neq \emptyset$ as $\mathbf{0} \in \mathcal{R}_2(x_0, v)$, and for $x_0 \neq v$, $\mathcal{R}_2(x_0, v)$ is compact (Lemma D.8). Furthermore, suppose $\mathcal{R}_2(x_0, v)$ is jointly upper-hemicontinuous correspondence, and for $\gamma \in \mathcal{R}_2(x_0, v)$, $\mathbb{E}_\eta\left(\log g^L(X, \gamma, x_0, v)\right)$ is jointly upper-semicontinuous in $(x_0, v, \eta, \gamma)$, then $h^*(x_0, v, \eta)$ is upper-semicontinuous ([8, Theorem 2, Page 116]). It then suffices to prove the following:

1. For $x_0 \neq v$, $\gamma \in \mathcal{R}_2(x_0, v)$, $h(x_0, v, \eta, \gamma) = \mathbb{E}_\eta\left(\log g^L(X, \gamma, x_0, v)\right)$ is a jointly upper-semicontinuous function.

2. For $x_0 \in C_v \setminus \{v\}$ and $v \in \underline{D}$, $\mathcal{R}_2(x_0, v)$ is an upper-hemicontinuous correspondence.

***Proof of (1):*** Consider a sequence $(x_n, v_n, \eta_n, \gamma_n) \in C_{v_n} \times \underline{D} \times \mathcal{L} \times \mathcal{R}_2(x_n, v_n)$ converging to $(x_0, v, \eta, \gamma) \in C_v \times \underline{D} \times \mathcal{L} \times \mathcal{R}_2(x_0, v)$, where convergence is defined coordinate wise, and $\eta_n$ converges to $\eta$ in topology of weak convergence. It is sufficient to show that

$$\limsup_{n \to \infty} h(x_n, v_n, \eta_n, \gamma_n) \leq h(x_0, v, \eta, \gamma).$$

Since $\eta_n \overset{D}{\Rightarrow} \eta$, by Skorokhod's Representation Theorem (see, [9]), there are random variables, $Y_n, Y$ defined on a common probability space, $(\Omega, \mathcal{F}, q)$, such that $Y_n \overset{a.s.}{\longrightarrow} Y$ and $Y_n \sim \eta_n$ and $Y \sim \eta$. Then, $\log\left(g^L(Y_n, \gamma_n, x_n, v_n)\right) \overset{a.s.}{\longrightarrow} \log\left(g^L(Y, \gamma, x_0, v)\right)$, and

$$h(x_n, v_n, \eta_n, \gamma_n) = \mathbb{E}_q\left(\log g^L(Y_n, \gamma_n, x_n, v_n)\right) \quad \text{and} \quad h(x_0, v, \eta, \gamma) = \mathbb{E}_q\left(\log g^L(Y, \gamma, x_0, v)\right).$$

Let

$$0 \leq Z_n \triangleq c_{1n} + c_{2n}|Y_n| + c_{3n}|Y_n|^{1+\epsilon},$$

where

$$c_{1n} = \gamma_{1n}(v_n - x_n) + \frac{\gamma_{1n}|x_n|}{1-\pi} + \gamma_{2n}B, \quad c_{2n} = \frac{\gamma_{1n}}{1-\pi}, \quad c_{3n} = \gamma_{2n}.$$

Clearly, each $c_{in}$ converge to $c_i < \infty$. With these notation, $\log\left(g^L(Y_n, \gamma_n, x_n, v_n)\right)$ is bounded by $\log(1 + Z_n)$, and $Z_n \overset{n \to \infty}{\longrightarrow} Z$. Thus, there exist $c_{0n} \overset{n \to \infty}{\longrightarrow} c_0 < \infty$ such that $\log(1 + Z_n) \leq c_{0n} + |Z_n|^{1/(1+\epsilon)}$ and using the form of $Z_n$ from above, there also exist constants $c_{4n} \overset{n \to \infty}{\longrightarrow} c_4$ and $c_{5n} \overset{n \to \infty}{\longrightarrow} c_5$ such that

$$|Z_n|^{1/(1+\epsilon)} \leq c_{4n} + c_{5n}|Y_n|.$$

Thus, there exist constants $c_{0n}, c_{4n}, c_{5n}$ converging to $c_0, c_4, c_5$ such that

$$\log\left(g^L(Y_n, \gamma_n, x_n, v_n)\right) \leq c_{0n} + c_{4n} + c_{5n}|Y_n| \triangleq f^L(Y_n, \gamma_n, x_n, v_n).$$

Furthermore,

$$f^L(Y_n, \gamma_n, x_n, v_n) \overset{a.s.}{\longrightarrow} f^L(Y, \gamma, x_0, v) \quad \text{and} \quad \mathbb{E}_q\left(f^L(Y_n, \gamma_n, x_n, v_n)\right) \to \mathbb{E}_q\left(f^L(Y, \gamma, x_0, v)\right),$$

since $\eta_n, \eta \in \mathcal{L}$ which is a collection of uniformly integrable measures (see, [57]). Since, $f^L(Y_n, \boldsymbol{\gamma}_n, x_n, v_n) - \log g^L(Y_n, \boldsymbol{\gamma}_n, x_n, v_n) \geq 0$, by Fatou's Lemma,

$$\mathbb{E}_q \big( \liminf_{n\to\infty} (f^L(Y_n, \boldsymbol{\gamma}_n, x_n, v_n) - \log g^L(Y_n, \boldsymbol{\gamma}_n, x_n, v_n)) \big)$$
$$\leq \mathbb{E}_q \left( f^L(Y, \boldsymbol{\gamma}, x_0, v) \right) - \limsup_{n\to\infty} \mathbb{E}_q \left( \log g^L(Y_n, \boldsymbol{\gamma}_n, x_n, v_n) \right),$$

which implies

$$h(x_0, v, \eta, \boldsymbol{\gamma}_n) = \mathbb{E}_q \left( \limsup_{n\to\infty} \log \left( g^L(Y_n, \boldsymbol{\gamma}_n, x_n, v_n) \right) \right) \geq \limsup_{n\to\infty} \mathbb{E}_q \left( \log \left( g^L(Y_n, \boldsymbol{\gamma}_n, x_n, v_n) \right) \right)$$
$$= \limsup_{n\to\infty} h(x_n, v_n, \eta_n, \boldsymbol{\gamma}_n).$$

***Proof of (2):*** Clearly, $(0,0) \in \mathcal{R}_2(x, v)$ for all $x \in C_v$ and $v \in \underline{D}$. Next, consider a sequence $(x_n, v_n) \longrightarrow (x_0, v) \in C_v \times \underline{D}$ and a sequence $\boldsymbol{\gamma}_n \in \mathcal{R}_2(x_n, v_n)$. Since $(x_n, v_n) \longrightarrow (x_0, v)$, there exists a closed and bounded (compact) subset, K, of $\Re \times \Re$ containing $(x_0, v)$, such that for some $J \geq 1$, and all $n \geq J$, $(x_n, v_n) \in K$. Since $\min_y g^L(y, \cdot, \cdot, \cdot)$ is a jointly continuous function, for $n \geq J$, $\boldsymbol{\gamma}_n$ also belongs to a compact subset of $\Re$. Bolzano-Weierstrass theorem then gives a convergent subsequence $\{(x_{n_i}, v_{n_i}), \boldsymbol{\gamma}_{n_i}\}$ in $\Re^3$ with the limit $\{(x_0, v), \boldsymbol{\gamma}\}$. It is then sufficient to show that $\boldsymbol{\gamma} \in \mathcal{R}_2(x_0, v)$, which follows since

$$g^L(y, \boldsymbol{\gamma}_n, x_n, v_n) \geq 0 \;\; \Rightarrow \;\; g^L(y, \boldsymbol{\gamma}, x_0, v) \geq 0,$$

proving that the correspondence $\mathcal{R}_2(\cdot, \cdot)$ is upper-hemicontinuous (see, [52, Proposition 9.8]). This completes the proof for *upper-semicontinuity of* $\mathrm{KL}^L_{\mathrm{inf}}(\eta, v)$ *for* $v < c_\pi(\eta)$.

**Joint upper-semicontinuity of** $\mathrm{KL}^L_{\mathrm{inf}}(\eta, c_\pi(\eta))$**:** Towards this, consider a sequence $(\eta_n, v_n) \in \mathcal{L} \times \underline{D}$ converging to $(\eta, c_\pi(\eta))$, where the convergence is defined coordinate-wise and in the first coordinate it is in the Lévy metric. Without loss of generality, assume that $v_n \leq c_\pi(\eta_n)$ for all $n$. It is then sufficient to argue that $\mathrm{KL}^L_{\mathrm{inf}}(\eta_n, v_n) \xrightarrow{n\to\infty} 0$.

We demonstrate a sequence of measures $\kappa_n \in \mathcal{L}$ which are feasible to $\mathrm{KL}^L_{\mathrm{inf}}(\eta_n, v_n)$ problem, such that $\mathrm{KL}(\eta_n, \kappa_n) \xrightarrow{n\to\infty} 0$, whence $\mathrm{KL}^L_{\mathrm{inf}}(\eta_n, v_n) \xrightarrow{n\to\infty} 0$. Define

$$w_n = \frac{\mathbb{E}_{\eta_n}(X - z_n)_+ - (1-\pi)(v_n - z_n)}{\mathbb{E}_{\eta_n}(X - z_n)_+} \quad \text{and} \quad \kappa_n = w_n \delta_{-f^{-1}(B) + z_n} + (1 - w_n)\eta_n,$$

where,

$$z_n = \begin{cases} x_\pi(\eta) - \frac{c_\pi(\eta) - v_n}{2}, & \text{for } v_n \leq c_\pi(\eta) \\ x_\pi(\eta), & \text{otherwise.} \end{cases}$$

It is easy to check that for $w_n \in [0, 1]$, $\kappa_n \in \mathcal{L}$. The above choice of $z_n$ ensures that $w_n \in [0, 1]$. Furthermore,

$$c_\pi(\kappa_n) \leq z_n + \frac{1}{1 - \pi} \mathbb{E}_{\kappa_n}(X - z_n)_+ \leq v_n,$$

where the last inequality follows from the choice of $w_n$, whence $\kappa_n$ are feasible.

Since $v_n \xrightarrow{n\to\infty} c_\pi(\eta)$, $\eta_n \xRightarrow{D} \eta$, and $\eta_n, \eta \in \mathcal{L}$, $\mathbb{E}_{\eta_n}(X - z_n)_+ \xrightarrow{n\to\infty} \mathbb{E}_\eta(X - x_\pi(\eta))_+$, whence, $w_n \xrightarrow{n\to\infty} 0$. With this choice of $\kappa_n$, $\mathrm{KL}^L_{\mathrm{inf}}(\eta_n, v_n)$ is bounded from above by

$$-\log(1 - w_n) \xrightarrow{n\to\infty} 0 = \mathrm{KL}^L_{\mathrm{inf}}(\eta, c_\pi(\eta)).$$

$\square$

**Lemma C.5.** $\mathrm{KL}^U_{\mathrm{inf}}$, *viewed as a function from* $\mathcal{L} \times \left[ -f^{-1}(B), f^{-1}\left(\frac{B}{1-\pi}\right) \right)$, *is a jointly upper-semicontinuous function.*

*Proof.* Proof for upper-semicontinuity of $\text{KL}_{\text{inf}}^{\text{U}}$ follows exactly as proof of the previous lemma. However, we give it for completeness. Define

$$\tilde{D} = \left[ -f^{-1}(B), f^{-1}\left( \frac{B}{1-\pi} \right) \right).$$

Consider the dual formulation of $\text{KL}_{\text{inf}}^{\text{U}}$ from Theorem 3.4(a). Since for $v \in \tilde{D}$, $\hat{S}(\eta, v)$ (defined in (11)) is a compact set (see Section D.3), and for all $y \in \Re$ $g^U(y, \cdot, \cdot)$ is a jointly continuous map, $\hat{S}(\cdot)$ can be verified to be an upper-hemicontinuous correspondence. Whence, it suffices to show that $h(v, \eta, \boldsymbol{\lambda}) := \mathbb{E}_\eta \left( \log \left( g^U(X, \boldsymbol{\lambda}, v) \right) \right)$ is a jointly upper-semicontinuous map, where $g^U$ is defined in (9) above.

Consider a sequence $(v_n, \eta_n, \boldsymbol{\lambda}_n) \in \tilde{D} \times \mathcal{L} \times \hat{S}(v_n)$ converging to $v, \eta, \boldsymbol{\lambda} \in D \times \mathcal{L} \times \hat{S}(v)$. Notice that the convergence is defined coordinate-wise, and $\eta_n$ converges to $\eta$ in weak topology. It suffices to show:

$$\limsup_{n \to \infty} \; h(v_n, \eta_n, \boldsymbol{\lambda}_n) \leq h(v, \eta, \boldsymbol{\lambda}).$$

By Skorokhod's Theorem (see, [9]), there exist random variables $Y_n, Y$ defined on a common probability space $(\Omega, \mathcal{F}, q)$ such that $Y_n \sim \eta_n$, $Y \sim \eta$ and $Y_n \xrightarrow{a.s.} Y$. Hence, $\log \left( g^U(Y_n, \boldsymbol{\lambda_n}, v_n) \right) \xrightarrow{a.s.} \log \left( g^U(Y, \boldsymbol{\lambda}, v) \right)$, and

$$h(v_n, \eta_n, \boldsymbol{\lambda}_n) = \mathbb{E}_q \left( \log \left( g^U(Y_n, \boldsymbol{\lambda}_n, v_n) \right) \right) \quad \text{and} \quad h(v, \eta, \boldsymbol{\lambda}) = \mathbb{E}_q \left( \log \left( g^U(Y, \boldsymbol{\lambda}, v) \right) \right).$$

As earlier, let

$$0 \leq Z_n = c_{1n} + c_{2n} |Y_n| + c_{3n} |Y_n|^{1+\epsilon},$$

where

$$c_{1n} = \lambda_{1n} |v_n| + |\lambda_{2n}(1 - \pi)| + \lambda_{3n} B, \quad c_{2n} = \frac{\lambda_{1n}}{1 - \pi}, \quad c_{3n} = \lambda_{3n},$$

and $Z_n \xrightarrow{n \to \infty} Z$ and $c_{in} \xrightarrow{n \to \infty} c_i < \infty$. With these notation, $\log \left( g^U(Y_n, \boldsymbol{\lambda}_n, v_n) \right)$ is bounded from above by $\log(1 + Z_n)$, and there exist constants $c_{0n} \xrightarrow{n \to \infty} c_0$ such that $\log(1 + Z_n) \leq c_{0n} + (Z_n)^{1/(1+\epsilon)}$. Using the form of $Z_n$ from above, there also exist constants $c_{4n} \xrightarrow{n \to \infty} c_4$ and $c_{5n} \xrightarrow{n \to \infty} c_5$ such that

$$(Z_n)^{1/(1+\epsilon)} \leq c_{4n} + c_{5n} |Y_n|.$$

Thus as earlier, there exist constants $c_{0n}, c_{4n}, c_{5n}$ converging to $c_0, c_4, c_5$ such that

$$\log \left( g^U(Y_n, \boldsymbol{\lambda}_n, v_n) \right) \leq c_{0n} + c_{4n} + c_{5n} |Y_n| \triangleq f^U(Y_n, \boldsymbol{\lambda}_n, v_n).$$

and

$$f^U(Y_n, \boldsymbol{\lambda}_n, v_n) \xrightarrow{a.s.} f^U(Y, \boldsymbol{\lambda}, v) \quad \text{and} \quad \mathbb{E}_q \left( f^U(Y_n, \boldsymbol{\gamma}_n, v_n) \right) \to \mathbb{E}_q \left( f^U(Y, \boldsymbol{\gamma}, v) \right),$$

since $\eta_n, \eta \in \mathcal{L}$, whence $Y_n, Y$ are uniformly integrable (see, [57]) . Since, $f^U(Y_n, \boldsymbol{\lambda}_n, v_n) - \log \left( g^U(Y_n, \boldsymbol{\lambda}_n, v_n) \right) \geq 0$, by Fatou's Lemma,

$$\mathbb{E}_q \left( \liminf_{n \to \infty} \left( f^U(Y_n, \boldsymbol{\lambda}_n, v_n) - \log \left( g^U(Y_n, \boldsymbol{\lambda}_n, v_n) \right) \right) \right) \leq \mathbb{E}_q \left( f^U(Y, \boldsymbol{\lambda}, v) \right)$$
$$- \limsup_{n \to \infty} \mathbb{E}_q \left( \log \left( g^U(Y_n, \boldsymbol{\lambda}_n, v_n) \right) \right),$$

which implies

$$h(v, \eta, \boldsymbol{\lambda}) = \mathbb{E}_q \left( \limsup_{n \to \infty} \log \left( g^U(Y_n, \boldsymbol{\lambda}_n, v_n) \right) \right) \geq \limsup_{n \to \infty} \mathbb{E}_q \left( \log \left( g^U(Y_n, \boldsymbol{\lambda}_n, v_n) \right) \right)$$
$$= \limsup_{n \to \infty} h(v_n \eta_n, \boldsymbol{\lambda}_n).$$

$\square$

**Remark C.1.** $\kappa_1 = \pi\delta_0 + (1-\pi)\delta_{f^{-1}\left(\frac{B}{1-\pi}\right)}$ and $\kappa_2 = \delta_{-f^{-1}(B)}$ are unique measures in $\mathcal{L}$ with CVaR being $f^{-1}\left(\frac{B}{1-\pi}\right)$ and $-f^{-1}(B)$, respectively. Uniqueness of $\kappa_2$ follows from the proof of Lemma 2.1. To see the uniqueness of $\kappa_1$, consider the following optimization problem, optimal value of which equals $f^{-1}\left(\frac{B}{1-\pi}\right)$:

$$\max_{\eta\in\mathcal{L}} \min_{z\in C} \left\{ z + \frac{1}{1-\pi}\mathbb{E}_\eta\left(X - z\right)_+ \right\}.$$

First observe that if $\mathbb{E}_\eta\left(f(X)\right) < B$, then $\eta$ does not belong to the set of maximizers above. Using this, it is also sufficient to restrict to 2-point distributions with $x_\pi(\eta) = 0$ and mass on 0 being $\pi$, as otherwise we can improve in the $B$ constraint, and hence, the objective. Now, $\kappa_1$ is the unique distribution satisfying the above requirements, with CVaR being $f^{-1}\left(\frac{B}{1-\pi}\right)$.

**Remark C.2.** Consider $v_n = v = f^{-1}\left(\frac{B}{1-\pi}\right)$, and let

$$\eta_n = \left(\pi - \frac{1}{n}\right)\delta_0 + \frac{1}{n}\delta_1 + (1-\pi)\delta_{f^{-1}\left(\frac{B}{1-\pi}\right)}, \quad \text{and} \quad \eta = \pi\delta_0 + (1-\pi)\delta_{f^{-1}\left(\frac{B}{1-\pi}\right)}.$$

Clearly, $d_L(\eta_n, \eta) \to 0$, as $n \to \infty$. Moreover, Remark C.1 argues that there is a unique $\kappa \in \mathcal{L}$ such that $c_\pi(\kappa) = f^{-1}\left(\frac{B}{1-\pi}\right)$, whence

$$\mathrm{KL}_{\mathrm{inf}}^{\mathrm{U}}(\eta_n, v_n) = \mathrm{KL}(\eta_n, \kappa) = \infty > 0 = \mathrm{KL}(\eta, \kappa) = \mathrm{KL}_{\mathrm{inf}}(\eta, v).$$

Thus, $\mathrm{KL}_{\mathrm{inf}}^{\mathrm{U}}(\eta, f^{-1}(B(1-\pi)^{-1}))$ is not a jointly continuous function. Similar example can be constructed for $\mathrm{KL}_{\mathrm{inf}}^{\mathrm{L}}(, -f^{-1}(B))$.

## C.3 Proof of Lemma 3.3:

**Upper-hemicontinuity of $t^*$:** Let $\nu$ be in $\mathcal{A}_j \cap \mathcal{M}$, i.e., the best-CVaR arm in $\nu$ is arm $j$, and each arm-distribution strictly satisfies the moment-constraint. Then from Lemma 2.1, for all $i \in [K]$, $c_\pi(\nu_i) \in D^o$. Let $t^*(\nu)$ be the set of maximizers in

$$V(\nu) = \max_{t\in\Sigma_K} \min_{a\neq j} g_{a,j}(\nu, t),$$

where

$$g_{a,j}(\nu, t) = \inf_{x\leq y} \left\{ t_j\,\mathrm{KL}_{\mathrm{inf}}^{\mathrm{U}}(\nu_j, x) + t_a\,\mathrm{KL}_{\mathrm{inf}}^{\mathrm{L}}(\nu_a, y) \right\}.$$

The infimum above is attained at a common point between the CVaR of the two distributions, whence the above equals

$$g_{a,j}(\nu, t) = \inf_{x\in[c_\pi(\nu_j), c_\pi(\nu_a)]} \left\{ t_j\,\mathrm{KL}_{\mathrm{inf}}^{\mathrm{U}}(\nu_j, x) + t_a\,\mathrm{KL}_{\mathrm{inf}}^{\mathrm{L}}(\nu_a, x) \right\}.$$

Using Lemma C.1, it is easy to verify that the set $[c_\pi(\nu_j), c_\pi(\nu_a)]$ is both upper- and lower- hemicontinuous in $(\nu, t)$, whence continuous. Then by Berge's Theorem and joint continuity of $\mathrm{KL}_{\mathrm{inf}}^{\mathrm{L}}$ and $\mathrm{KL}_{\mathrm{inf}}^{\mathrm{U}}$ in arguments, when viewed as functions from $\mathcal{L} \times D^o$ (Lemma 3.2), $g_{a,j}(\nu, t)$ is jointly continuous in $(\nu, t)$. Again by Berge's Theorem, $V(\nu)$, as a function from $\mathcal{M}$ to $\Re$, is a continuous function of $\nu$. Furthermore, the set of maximizers, $\{t^* : V(\nu) = \min_{a\neq j} g_{a,j}(\nu, t^*)\}$, is an upper-hemicontinuous correspondence.

**Convexity of the set of maximizers:** Let $t^{(1)}$ and $t^{(2)}$ belong to $t^*(\nu)$. Then,

$$V(\mu) = \min_{a\neq j} g_a(\nu, t^{(1)}) = \min_{a\neq j} g_a(\nu, t^{(2)}).$$

Clearly, $\min_{a\neq j} g_a(\nu, \lambda t^{(1)} + (1-\lambda)t^{(2)}) \geq \lambda \min_{a\neq j} g_a(\nu, t^{(1)}) + (1-\lambda)\min_{b\neq j} g_b(\nu, t^{(2)})$, which equals $V(\nu)$. Since $t^{(1)}$ and $t^{(2)}$ are maximizers, the above holds as an equality. Thus the set $t^*(\nu)$ is convex.

# D Dual formulations

In this section we prove the Theorem 3.4. Recall that $\mathcal{P}(\Re)$ denotes the space of all probability measures on $\Re$, and $M^+$ denotes the collection of all finite, positive measures on $\Re$. Let $\eta \in \mathcal{P}(\Re)$. Then, for $\pi \in (0,1)$, $c_\pi(\eta)$ denotes the CVaR of $\eta$ at the confidence level $\pi$. Furthermore,

$$c_\pi(\eta) = \min_{x_0 \in \Re} \left\{ x_0 + \frac{1}{1-\pi} \mathbb{E}_\eta \left( (X - x_0)_+ \right) \right\} \tag{15}$$

$$= \max_{v \in M^+(\Re)} \frac{1}{1-\pi} \int_\Re y \, dv(y) \text{ s.t. } \forall y, \; 0 \leq dv(y) \leq d\eta(y) \text{ and } \int_\Re dv(y) = 1 - \pi, \tag{16}$$

For $\eta \in \mathcal{P}(\Re)$, and $v \in D^o$,

$$\mathrm{KL}_{\inf}^{\mathrm{U}}(\eta, v) = \inf_{\kappa \in \mathcal{L}: \, c_\pi(\kappa) \geq v} \mathrm{KL}(\eta, \kappa) \quad \text{and} \quad \mathrm{KL}_{\inf}^{\mathrm{L}}(\eta, v) = \inf_{\kappa \in \mathcal{L}: \, c_\pi(\kappa) \leq v} \mathrm{KL}(\eta, \kappa).$$

Furthermore, extend the Kullback-Leibler Divergence to a function on $M^+(\Re) \times M^+(\Re)$, i.e., $\mathrm{KL} : M^+(\Re) \times M^+(\Re) \to \Re$ defined as:

$$\mathrm{KL}(\kappa_1, \kappa_2) \triangleq \int_{y \in \Re} \log \left( \frac{d\kappa_1}{d\kappa_2}(y) \right) d\kappa_1(y).$$

Note that for $\kappa_1 \in \mathcal{P}(\Re)$ and $\kappa_2 \in \mathcal{P}(\Re)$, $\mathrm{KL}(\kappa_1, \kappa_2)$ is the usual Kullback-Leibler Divergence between the probability measures.

We first present the proof for the Theorem 3.4(a).

## D.1 $\mathrm{KL}_{\inf}^{\mathrm{U}}$ problem: towards proving Theorem 3.4(a)

Consider the following optimization problem, which is equivalent to the $\mathrm{KL}_{\inf}^{\mathrm{U}}$ problem (see, (16)).

$$\min_{\substack{\kappa \in M^+ \\ W \in M^+}} \mathrm{KL}(\eta, \kappa) \qquad \text{subject to} \qquad \frac{1}{1-\pi} \int_\Re x \, dW(x) \geq v \tag{17}$$

$$\int_\Re dW(x) = 1 - \pi$$

$$\int_\Re f(x) \, d\kappa(x) \leq B$$

$$\int_\Re d\kappa(x) = 1$$

$$\forall x : 0 \leq dW(x) \leq d\kappa(x)$$

Introducing the dual variables $(\lambda_1 \geq 0, \lambda_2 \in \Re, \lambda_3 \geq 0, \lambda_4 \in \Re, \forall x \; \lambda_5(x) \geq 0)$. Then, the Lagrangian, denoted as $L(\kappa, W, \boldsymbol{\lambda})$, equals

$$\int_\Re \log \left( \frac{d\eta}{d\kappa}(y) \right) d\eta(y) + \lambda_1 \left( v - \frac{1}{1-\pi} \int_\Re x \, dW(x) \right) + \lambda_2 \left( \int_\Re dW(x) - 1 + \pi \right)$$

$$- \lambda_3 B + \lambda_3 \int_\Re f(x) \, d\kappa(x) + \lambda_4 \left( \int_\Re d\kappa(x) - 1 \right) + \int_\Re \lambda_5(x) \left( dW(x) - d\kappa(x) \right).$$

The Lagrangian dual problem is

$$\max_{\substack{\lambda_1 \geq 0, \lambda_2 \in \Re, \lambda_3 \geq 0, \\ \lambda_4 \in \Re, \forall x : \lambda_5(x) \geq 0}} \; \inf_{\substack{\kappa \in M^+ \\ W \in M^+}} L(\kappa, W, \boldsymbol{\lambda}). \tag{18}$$

Let $S = (\lambda_1 \geq 0, \lambda_2 \in \Re, \lambda_3 \geq 0, \lambda_4 \in \Re, \forall x : \lambda_5(x) \geq 0)$, and define

$$S_1 = S \cap \{ \boldsymbol{\lambda} : \forall x \in \Re, \lambda_4 + \lambda_3 f(x) - \lambda_3 B - \lambda_5(x) \geq 0 \}.$$

**Lemma D.1.** *The Lagrangian dual problem (18) satisfies*

$$\max_{\substack{\lambda_1 \geq 0, \lambda_2 \in \Re, \lambda_3 \geq 0, \\ \lambda_4 \in \Re, \forall x : \lambda_5(x) \geq 0}} \; \inf_{\substack{\kappa \in M^+ \\ W \in M^+}} L(\kappa, W, \boldsymbol{\lambda}) = \max_{\boldsymbol{\lambda} \in S_1} \; \inf_{\substack{\kappa \in M^+ \\ W \in M^+}} L(\kappa, W, \boldsymbol{\lambda}).$$

*Proof.* Consider $\boldsymbol{\lambda} \in S$ and $\boldsymbol{\lambda} \notin S_1$. Then, there exists $y_0 \in \Re$ such that

$$\lambda_4 + \lambda_3 f(y_0) - \lambda_3 B - \lambda_5(y_0) < 0.$$

Consider the measure $\kappa_M \in M_+$ such that $\kappa_M(y_0) = M$ and

$$\frac{d\eta}{d\kappa_M}(y) = 1, \quad \text{for } y \in \{\text{Supp}(\eta) \setminus y_0\}.$$

Then, $L(\kappa_M, W, \boldsymbol{\lambda})$ equals

$$\int_\Re \log\left(\frac{d\eta}{d\kappa_M}(y)\right) d\eta(y) + \int_\Re (\lambda_4 + \lambda_3 f(y) - \lambda_3 B - \lambda_5(x)) d\kappa_M(x) \tag{19}$$
$$+ \lambda_1\left(v - \frac{1}{1-\pi}\int_\Re x dW(x)\right) + \lambda_2\left(\int_\Re dW(x) - 1 + \pi\right) - \lambda_4 + \int_\Re \lambda_5(x) dW(x).$$

Clearly, the first two terms in the expression above decrease to $-\infty$ as $M$ increases to $\infty$. Thus, for $\boldsymbol{\lambda} \in S$ and $\boldsymbol{\lambda} \notin S_1$, the infimum in the inner optimization problem in (18) is $-\infty$, and we get the desired equality. $\quad\square$

Let $\mathcal{Z}(\boldsymbol{\lambda}) = \{y \in \Re : \lambda_4 + \lambda_3 f(y) - \lambda_5(y) = 0\}$.

**Lemma D.2.** *For $\boldsymbol{\lambda} \in S_1$, $\kappa^*$ that minimizes $L(\kappa, W, \boldsymbol{\lambda})$ satisfies $\text{Supp}(\kappa^*) \subset \text{Supp}(\eta) \cup \mathcal{Z}(\boldsymbol{\lambda})$. Furthermore, for $y \in \text{Supp}(\eta)$, $\lambda_4 + \lambda_3 f(y) - \lambda_3 B - \lambda_5(y) > 0$, and*

$$\frac{d\kappa^*}{d\eta}(y) = (\lambda_4 + \lambda_3 f(y) - \lambda_3 B - \lambda_5(y))^{-1}. \tag{20}$$

*Proof.* Clearly, for $\boldsymbol{\lambda} \in S_1$, $L(\boldsymbol{\kappa}, \boldsymbol{W}, \boldsymbol{\lambda})$ is a strictly convex function of $\kappa$ being minimized over a convex set $M^+$. Thus, if there is a minimizer of $L(\kappa, W, \boldsymbol{\lambda})$ over $M^+$, it is unique. It is then sufficient to show that $\kappa^*$ satisfying the conditions of the Lemma minimizes $L(\kappa, W, \boldsymbol{\lambda})$. Let $\kappa_1 \neq \kappa^*$ and $\kappa_1 \in M^+$. For $t \in [0,1]$, define $\kappa_{2,t} = (1-t)\kappa^* + t\kappa_1$. Then $\kappa_{2,t} \in M^+$ and it suffices to show that

$$\left.\frac{\partial L(\kappa_{2,t}, W, \boldsymbol{\lambda})}{\partial t}\right|_{t=0} \geq 0.$$

To see this, substituting for $\kappa_{2,t}$ in (19), $L(\kappa_{2,t}, W, \boldsymbol{\lambda})$ equals

$$\int_{\text{Supp}(\eta)} \log\left(\frac{d\eta}{d\kappa_{2,t}}(y)\right) d\eta(y) + \int_\Re (\lambda_4 + \lambda_3 f(y) - \lambda_5(x)) d\kappa_{2,t}(x)$$
$$+ \lambda_1\left(v - \frac{1}{1-\pi}\int_\Re x dW(x)\right) - \lambda_3 B + \lambda_2\left(\int_\Re dW(x) - 1 + \pi\right) - \lambda_4 + \int_\Re \lambda_5(x) dW(x).$$

Differentiating with respect to $t$ and evaluating at $t = 0$, the derivative $\left.\frac{\partial L(\kappa_{2,t}, W, \boldsymbol{\lambda})}{\partial t}\right|_{t=0}$ equals

$$\int_{\text{Supp}(\eta)} \frac{d\eta}{d\kappa^*}(y)(d\kappa^* - d\kappa_1)(y) + \int_\Re (\lambda_4 + \lambda_3 f(y) - \lambda_3 B - \lambda_5(y))(d\kappa_1 - d\kappa^*)(y).$$

Now, using the form of $\kappa^*$ from (20), the above expression simplifies to

$$\int_{\Re\setminus\text{Supp}(\eta)} (\lambda_4 + \lambda_3 f(y) - \lambda_3 B - \lambda_5(y)) d\kappa_1(y) - \int_{\Re\setminus\text{Supp}(\eta)} (\lambda_4 + \lambda_3 f(y) - \lambda_3 B - \lambda_5(y)) d\kappa^* \geq 0,$$

where the inequality above follows since the integrand is 0 in the second term, while it is non-negative in the first term. $\quad\square$

### D.1.1 Proof of Theorem 3.4(a)

We first show that the dual problem in (18) simplifies to the alternative expression for $\mathrm{KL}_{\mathrm{inf}}^{\mathrm{U}}(\eta, v)$ in the Theorem. Then we argue that both the $\mathrm{KL}_{\mathrm{inf}}^{\mathrm{U}}$ primal problem in (17) and the dual problems are feasible, and that strong duality holds.

Using the expression for the optimizer for optimal $\kappa^*$ (Lemma D.2) in the Lagrangian dual in Lemma D.1, (18) equals

$$
\max_{\boldsymbol{\lambda} \in S_1} \inf_{W \in M^+} \int_{\Re} \log \left( \lambda_4 + \lambda_3 f(y) - \lambda_3 B - \lambda_5(y) \right) d\eta(y)
$$
$$
+ \int_{\Re} dW(x) \left( -\frac{\lambda_1 x}{1 - \pi} + \lambda_2 + \lambda_5(x) \right) + 1 + \lambda_1 v - \lambda_2(1 - \pi) - \lambda_4.
$$

Since $W \in M^+$, and if $\boldsymbol{\lambda}$ are such that the integrand in the second term above is negative, then the value of the expression above will be $-\infty$. Thus, it suffices to restrict $\boldsymbol{\lambda}$ so that this does not happen. Let

$$
S_2 = S_1 \cap \left\{ \boldsymbol{\lambda} : \forall x, -\frac{\lambda_1 x}{1 - \pi} + \lambda_2 + \lambda_5(x) \geq 0 \right\}.
$$

Then the dual problem simplifies to

$$
\max_{\boldsymbol{\lambda} \in S_2} \int_{\Re} \log \left( \lambda_4 + \lambda_3 f(y) - \lambda_3 B - \lambda_5(y) \right) d\eta(y) + 1 + \lambda_1 v - \lambda_2(1 - \pi) - \lambda_4.
$$

Optimizing over the common scaling of the dual variables, we get

$$
\max_{\boldsymbol{\lambda} \in S_2} \int_{\Re} \log \left( \frac{\lambda_4 + \lambda_3 f(y) - \lambda_3 B - \lambda_5(y)}{-\lambda_1 v + \lambda_2(1 - \pi) + \lambda_4} \right) d\eta(y).
$$

Observe that $-\lambda_1 v + \lambda_2(1 - \pi) + \lambda_4 \geq 0$, for the dual optimal variables. Thus, it is sufficient to restrict the variables to satisfy this constraint. This follows from the complementary slackness condition and the restrictions in the set $S_2$. We later show that strong duality holds. Since the problem is a convex optimization problem, the dual optimal variables satisfy the complementary slackness conditions.

Setting $\tilde{\lambda}_4 = -\lambda_1 v + \lambda_2(1 - \pi) + \lambda_4$, and substituting in the above expression, we get

$$
\max_{\boldsymbol{\lambda} \in S_3(v)} \int_{\Re} \log \left( 1 + \lambda_1 v - \lambda_2(1 - \pi) + \lambda_3 f(y) - \lambda_3 B - \lambda_5(y) \right) d\eta(y),
$$

where $S_3(v)$ is $S_2$ with the above modifications, and is given by intersection of the set $S$ with the set

$$
\left\{ \boldsymbol{\lambda} : \forall y, \ 1 + \lambda_1 v - \lambda_2(1 - \pi) + \lambda_3 f(y) - \lambda_3 B - \lambda_5(y) \geq 0, \ \& \ \forall x \ \lambda_5(x) \geq \left( \frac{\lambda_1 x}{1 - \pi} - \lambda_2 \right)_+ \right\}.
$$

Further, optimizing over $\lambda_5(x)$, the dual representation simplifies to

$$
\max_{\boldsymbol{\lambda} \in \hat{S}(v)} \mathbb{E}_\eta \left( \log \left( 1 + \lambda_1 v - \lambda_2(1 - \pi) + \lambda_3 f(X) - \lambda_3 B - \left( \frac{\lambda_1 X}{1 - \pi} - \lambda_2 \right)_+ \right) \right).
$$

Thus, it suffices to show that both the primal problem in (17) and the dual in (18) are feasible, and strong duality holds.

Consider $\boldsymbol{\lambda}^1 = (0, 0, 0, 1, 0)$. To show that dual is feasible, it suffices to show

$$
\min_{\kappa \in M^+, W \in M^+} L(\kappa, W, \boldsymbol{\lambda}^1) = \min_{\kappa \in M^+, W \in M^+} \mathrm{KL}(\eta, \kappa) - 1 + \int_{\Re} d\kappa(y) > -\infty.
$$

Let $\tilde{\kappa}$ be the minimizer of the above expression. Then, $\mathrm{Supp}(\tilde{\kappa}) = \mathrm{Supp}(\eta)$, as otherwise if there is a point $y$ in $\mathrm{Supp}(\eta) \setminus \mathrm{Supp}(\tilde{\kappa})$, then the above expression is $\infty$, and if there is a point in

$\mathrm{Supp}(\tilde{\kappa}) \setminus \mathrm{Supp}(\eta)$, then the value of the above expression can be improved by removing that mass. Furthermore, from (20),

$$\frac{d\tilde{\kappa}}{d\eta}(y) = 1.$$

We next argue the feasibility of primal problem, and show that strong duality holds. For $v \leq 0$, define $\kappa_1 := \delta_{\epsilon_1}$, where $\epsilon_1 > v$ and $f(\epsilon_1) < B$. Similarly, for $v > 0$, define $\kappa_2 := q\delta_{f^{-1}\left(\frac{B}{1-\pi}\right)} + (1-q)\delta_0$, where $q < 1 - \pi$ is chosen to satisfy $c_\pi(\kappa_2) > v$ and $\mathbb{E}_{\kappa_2}(f(X)) < B$.

Clearly, $\kappa_1$ and $\kappa_2$ defined above, lie in the interior of the feasible region of the primal problem. Hence, strong duality holds if the primal is feasible. To see feasibility, define

$$\tilde{\kappa}_1 := p_1\eta + (1-p_1)\kappa_1 \quad \text{and} \quad \tilde{\kappa}_2 := p_2\eta + (1-p_2)\kappa_2,$$

where $p_1$ and $p_2$ are chosen to satisfy

$$c_\pi(\tilde{\kappa}_1) > v, \ \mathbb{E}_{\tilde{\kappa}_1}(f(X)) < B \quad \text{and} \quad c_\pi(\tilde{\kappa}_2) > v, \ \mathbb{E}_{\tilde{\kappa}_2}(f(X)) < B.$$

It is easy to see the existence of $p_1, p_2, \epsilon_1$, and $q$ satisfying the above requirement.

## D.2 $\mathrm{KL}^{\mathrm{L}}_{\mathrm{inf}}$ problem

For $\eta \in \mathcal{P}(\Re)$, and $v \in \Re$, using (15), the $\mathrm{KL}^{\mathrm{L}}_{\mathrm{inf}}$ optimization problem is equivalent to the following optimization problem (we refer to the inner optimization problem in the following as $\mathcal{O}_1$).

$$\inf_{-f^{-1}\left(\frac{B}{\pi}\right) \leq x_0 \leq v} \ \min_{\kappa \in M^+(\Re)} \ \mathrm{KL}(\eta, \kappa) \quad \text{subject to} \quad x_0 + \frac{1}{1-\pi}\int_\Re (y - x_0)_+ d\kappa(y) \leq v$$

$$\int_\Re f(y)\, d\kappa(y) \leq B$$

$$\int_\Re d\kappa(y) = 1.$$

We first characterize the solution to the inner optimization problem for a fixed $x_0$, $\mathcal{O}_1$. The proof is similar to that for the duality result in [2].

Let $\gamma = (\gamma_1, \gamma_2, \gamma_3)$. For $\kappa \in M^+(\Re)$, the Lagrangian, denoted by $L(\kappa, \gamma, x_0)$, for the Problem $\mathcal{O}_1$ is given by,

$$\mathrm{KL}(\eta, \kappa) + \gamma_1 \left( x_0 + \frac{1}{1-\pi}\int_\Re (y - x_0)_+ d\kappa(y) - v \right)$$

$$+ \gamma_2 \left( \int_\Re f(x)\, d\kappa(x) - B \right) + \gamma_3 \left( \int_\Re d\kappa(x) - 1 \right). \tag{21}$$

Define

$$L(\gamma, x_0) := \inf_{\kappa \in M^+(\Re)} L(\kappa, \gamma, x_0). \tag{22}$$

The Lagrangian dual problem corresponding to the Problem ($\mathcal{O}_1$) is given by

$$\max_{\gamma_1 \geq 0, \gamma_2 \geq 0, \gamma_3 \in \Re} \left( \inf_{\kappa \in M^+(\Re)} L(\kappa, \gamma, x_0) \right). \tag{23}$$

Let $\mathrm{Supp}(\kappa)$ denote the support of measure $\kappa$,

$$h(y, \gamma, x_0) \triangleq \frac{\gamma_1}{1-\pi}(y - x_0)_+ + \gamma_3 + \gamma_2 f(y), \quad \mathcal{Z}(\gamma) = \{y \in \Re : h(y, \gamma, x_0) = 0\},$$

and

$$\mathcal{R}_3(x_0) = \left\{ \gamma \in \Re^3 : \gamma_1 \geq 0, \ \gamma_2 \geq 0, \gamma_3 \in \Re, \ \inf_{y \in \Re} h(y, \gamma, x_0) \geq 0 \right\}.$$

Observe that for $\gamma \in \mathcal{R}_3(x_0)$, there is a unique element in $\mathcal{Z}(\gamma)$.

**Lemma D.3.** *The Lagrangian dual problem (23) is simplified as below.*

$$\max_{\gamma_3 \in \Re, \gamma_1 \geq 0, \gamma_2 \geq 0} \left( \inf_{\kappa \in M^+(\Re)} L(\kappa, \boldsymbol{\gamma}, x_0) \right) = \max_{\boldsymbol{\gamma} \in \mathcal{R}_3(x_0)} \left( \inf_{\kappa \in M^+(\Re)} L(\kappa, \boldsymbol{\gamma}, x_0) \right).$$

*Proof.* For $\boldsymbol{\gamma} \in \Re^3 \setminus \mathcal{R}_3(x_0)$, there exists $y_0 \in \Re$ such that $h(y_0, \boldsymbol{\gamma}, x_0) < 0$ and it suffices to show that $L(\boldsymbol{\gamma}, x_0) = -\infty$, where $L(\boldsymbol{\gamma}, x_0)$ is defined in (22). Observe that for every $M > 0$, there exists a measure $\kappa_M \in M^+(\Re)$ satisfying $\kappa_M(y_0) = M$ and for $y \in \text{Supp}(\eta) \setminus \{y_0\}$,

$$\frac{d\eta}{d\kappa_M}(y) = 1.$$

Then, (21) can be re-written as:

$$L(\kappa_M, \boldsymbol{\gamma}, x_0) = \underbrace{\int_{y \in \Re} \log\left(\frac{d\eta}{d\kappa_M}(y)\right) d\eta(y)}_{\triangleq A_1} + \underbrace{\int_{y \in \Re} h(y, \boldsymbol{\gamma}, x_0) d\kappa_M(y) + \gamma_1(x_0 - v) - \gamma_3 - \gamma_2 B}_{\triangleq A_2}.$$

From above, it can be easily seen that $L(\kappa_M, \boldsymbol{\lambda}) \xrightarrow{M \to \infty} -\infty$, since $A_1 + A_2 \to -\infty$. Thus, for $\boldsymbol{\gamma} \in \Re^3 \setminus \mathcal{R}_3(x_0)$, $L(\boldsymbol{\gamma}, x_0) = -\infty$ and we get the desired result. $\qquad\square$

**Lemma D.4.** *For $\boldsymbol{\gamma} \in \mathcal{R}_3(x_0)$, $\kappa^* \in M^+(\Re)$ that minimizes $L(\kappa, \boldsymbol{\gamma}, x_0)$, satisfies*

$$\text{Supp}(\kappa^*) \subset \{\text{Supp}(\eta) \cup \mathcal{Z}(\boldsymbol{\gamma})\}. \tag{24}$$

*Furthermore, for $y \in \text{Supp}(\eta)$, $h(y, \boldsymbol{\gamma}, x_0) > 0$, and*

$$\frac{d\kappa^*}{d\eta}(y) = \left(\frac{\gamma_1}{1 - \pi}(y - x_0)_+ + \gamma_3 + \gamma_2 f(y)\right)^{-1}. \tag{25}$$

*Proof.* For $\boldsymbol{\gamma} \in \mathcal{R}_3(x_0)$, $L(\kappa, \boldsymbol{\gamma}, x_0)$ is a strictly convex function of $\kappa$ being minimized over a convex set. Hence, if the minimizer of $L(\kappa, \boldsymbol{\gamma}, x_0)$ exists, it is unique. It then suffices to show that $\kappa^*$ satisfying (24) and (25) minimizes $L(\kappa, \boldsymbol{\gamma}, x_0)$.

Let $\kappa_1$ be any measure in $M^+(\Re)$ that is different from $\kappa^*$. Since $M^+(\Re)$ is a convex set, for $t \in [0, 1]$, $\kappa_{2,t} \triangleq (1 - t)\kappa^* + t\kappa_1$ belongs to $M^+(\Re)$. Since $L(\kappa, \boldsymbol{\gamma}, x_0)$ is convex in $\kappa$, to show that $\kappa^*$ minimizes $L(\kappa, \boldsymbol{\gamma}, x_0)$, it suffices to show

$$\left.\frac{\partial L(\kappa_{2,t}, \boldsymbol{\gamma})}{\partial t}\right|_{t=0} \geq 0.$$

Substituting for $\kappa_{2,t}$ in (21), $L(\kappa_{2,t}, \boldsymbol{\gamma})$ equals

$$\int_{y \in \text{Supp}(\eta)} \log\left(\frac{d\eta}{d\kappa_{2,t}}(y)\right) d\eta(y) + (\gamma_1(x_0 - v) - \gamma_3 - \gamma_2 B) + \int_\Re h(y, \boldsymbol{\gamma}, x_0) d\kappa_{2,t}(y).$$

Evaluating the derivative with respect to $t$ at $t = 0$,

$$\left.\frac{\partial L(\kappa_{2,t}, \boldsymbol{\gamma})}{\partial t}\right|_{t=0} = \int_{y \in \text{Supp}(\eta)} \frac{d\eta}{d\kappa^*}(y)(d\kappa^* - d\kappa_1)(y) + \int_\Re h(y, \boldsymbol{\gamma}, x_0)(d\kappa_1 - d\kappa^*)(y).$$

For $y \in \text{Supp}(\eta)$, $\partial\eta/\partial\kappa^* = h(y, \boldsymbol{\gamma}, x_0)$. Substituting this in the above expression, we get:

$$\left.\frac{\partial L(\kappa_{2,t}, \boldsymbol{\gamma})}{\partial t}\right|_{t=0} = \int_{y \in \text{Supp}(\eta)} h(y, \boldsymbol{\gamma}, x_0)(d\kappa^* - d\kappa_1)(y) - \int_\Re h(y, \boldsymbol{\gamma}, x_0)(d\kappa^* - d\kappa_1)(y)$$

$$= \int_{y \in \{\Re \setminus \text{Supp}(\eta)\}} h(y, \boldsymbol{\gamma}, x_0) d\kappa_1(y) - \int_{y \in \{\Re \setminus \text{Supp}(\eta)\}} h(y, \boldsymbol{\gamma}, x_0) d\kappa^*(y)$$

$$\geq 0,$$

where, for the last inequality, we have used the fact that for $y \in \{\text{Supp}(\kappa^*) \setminus \text{Supp}(\eta)\}$, $h(y, \boldsymbol{\gamma}, x_0) = 0$ and $h(y, \boldsymbol{\gamma}, x_0) \geq 0$, otherwise. $\qquad\square$

### D.2.1   Proof of Theorem 3.4(b)

To prove the alternative expression for $\mathrm{KL}_{\mathrm{inf}}^{\mathrm{L}}$ given by this theorem, we first show that both the primal and dual problems ($\mathcal{O}_1$ and $\mathcal{O}_2$, respectively) are feasible and that strong duality holds for the Problem $\mathcal{O}_1$. We then show that the alternative formulation for $\mathrm{KL}_{\mathrm{inf}}^{\mathrm{L}}$ is its simplified dual formulation.

Let $\delta_y$ denote a unit mass at point $y$. For $x_0 > 0$, define $\kappa_1 = (1-\pi)\delta_{x_0-\epsilon_0} + \pi\delta_0$, where $\epsilon_0$ is chosen to satisfy $f(x_0 - \epsilon_0)(1-\pi) < B$. Similarly, for the other case ($x_0 \leq 0$), for $x_0 \neq v$ and $x_0 > -f^{-1}\left(\frac{B}{\pi}\right)$, define $\kappa_2 := \pi\delta_{x_0-\epsilon_1} + (1-\pi)\delta_{x_0+\epsilon_2}$, where $\epsilon_1$ and $\epsilon_2$ are chosen to satisfy

$$x_0 + \epsilon_2 < v \quad \text{and} \quad \pi f|x_0 - \epsilon_1| + (1-\pi)f(x_0 + \epsilon_2) < B.$$

Also, for $x_0 = -f^{-1}\left(\frac{B}{\pi}\right)$, for $v > 0$, define $\kappa_3 = (1-\pi)\delta_0 + \pi\delta_{\left(-f^{-1}\left(\frac{B}{\pi}\right)+\epsilon_3\right)}$, where $\epsilon_3$ is chosen to satisfy $0 < \epsilon_3 < \frac{1-\pi}{\pi}v$. Similarly, for $x_0 = -f^{-1}\left(\frac{B}{\pi}\right)$ and $v < 0$, define $\kappa_4 = (\pi - \epsilon_4)\delta_{-f^{-1}\left(\frac{B}{\pi}\right)} + (1-\pi+\epsilon_4)\delta_{v-\epsilon_5}$, where $\epsilon_4 > 0$ and $\epsilon_5 > 0$ are chosen to satisfy

$$\frac{\epsilon_4}{1-\pi+\epsilon_4}\left(f^{-1}\left(\frac{B}{\pi}\right) + v\right) < \epsilon_5 < f^{-1}\left(\frac{B}{1-\pi+\epsilon_4}\right) + v.$$

Define

$$\kappa_0 := \kappa_1\mathbf{1}(x_0 > 0) + \kappa_2\mathbf{1}(x_0 \leq 0)\mathbf{1}\left(-f^{-1}\left(\frac{B}{\pi}\right) < x_0 < v\right) + \delta_v\mathbf{1}(x_0 = v)\mathbf{1}(x_0 \leq 0)$$

$$+ \mathbf{1}\left(x_0 = -f^{-1}\left(\frac{B}{\pi}\right)\right)(\kappa_3\mathbf{1}(v > 0) + \kappa_4\mathbf{1}(v \leq 0)).$$

Clearly, $\kappa_0$ defined above satisfies all the inequality constraints in the primal problem strictly, whence, lies in the interior of the feasible region.

Recall that $c_\pi(\eta)$ is a concave function of $\eta$ (see (15)). It is then easy to check that there exists $0 < p < 1$ such that $\kappa' := p\eta + (1-p)\kappa_0$ is feasible for the primal problem, and $\mathrm{KL}(\eta, \kappa') < \infty$. Hence, primal problem $\mathcal{O}_1$ is feasible.

Next, we claim that $\boldsymbol{\gamma}^{\mathbf{1}} = (0, 0, 1)$ is a dual feasible solution. To this end, it is sufficient to show that

$$\min_{\kappa \in M^+(\Re)} L(\kappa, (0, 0, 1), x_0) > -\infty.$$

Observe that for $\kappa \in \mathcal{M}^+(\Re)$, $\mathrm{KL}(\eta, \kappa)$ defined to extend the usual definition of Kullback-Leibler Divergence to include all measures in $M^+(\Re)$, can be negative with arbitrarily large magnitude. From (21),

$$L(\kappa, \boldsymbol{\gamma}^{\mathbf{1}}, x_0) = \mathrm{KL}(\eta, \kappa) - 1 + \int_\Re d\kappa(y).$$

Let $\tilde{\kappa}$ denote the minimizer of $L(\kappa, \boldsymbol{\gamma}^{\mathbf{1}}, x_0)$. Then, as earlier, $\mathrm{Supp}(\tilde{\kappa}) = \mathrm{Supp}(\eta)$. Furthermore, from Lemma D.4, for $y$ in $\mathrm{Supp}(\eta)$, the optimal measure $\tilde{\kappa}$ must satisfy

$$\frac{d\tilde{\kappa}}{d\eta}(y) = 1.$$

Thus, $\tilde{\kappa} = \eta$ and $\min_{\kappa \in M^+(\Re)} L(\kappa, \boldsymbol{\gamma}^{\mathbf{1}}, x_0) = 0$. This proves the feasibility of the dual problem $\mathcal{O}_2$.

Since both primal and dual problems are feasible, both have optimal solutions. Furthermore, $\kappa_0$ defined earlier satisfies all the inequality constraints of $(\mathcal{O}_1)$ strictly, hence lies in the interior of the feasible region (Slater's conditions are satisfied). Thus strong duality holds for the problem $(\mathcal{O}_1)$ and there exists optimal dual variable $\boldsymbol{\gamma}^* = (\gamma_1^*, \gamma_2^*, \gamma_3^*)$ that attains maximum in the problem $\mathcal{O}_2$ (see, [41, Theorem 1, Page 224]).

Also, since the primal problem (for fixed $x_0$) is minimization of a strictly-convex function (which is non-negative on the feasible set) with an optimal solution over a closed and convex set, it attains its infimum within the set. Strong duality implies

$$\mathrm{KL}_{\mathrm{inf}}(\eta, v) = \min_{-f^{-1}\left(\frac{B}{\pi}\right) \leq x_0 \leq v} \max_{\boldsymbol{\gamma} \in \mathcal{R}_3(x_0)} \inf_{\kappa \in M^+(\Re)} L(\kappa, \boldsymbol{\lambda}, x_0).$$

Let $\kappa^*$ and $\boldsymbol{\gamma}^*$ denote the optimal primal and dual variables. Since strong duality holds, and the problem $(\mathcal{O}_1)$ is a convex optimization problem, KKT conditions are necessary and sufficient for $\kappa^*$ and $\boldsymbol{\gamma}^*$ to be optimal variables (see, [11, page 224]). Hence $\kappa^*, \gamma_3^* \in \Re, \gamma_1^* \geq 0$, and $\gamma_2^* \geq 0$ must satisfy the following conditions (KKT):

$$\kappa^* \in M^+(\Re), \ \int_\Re d\kappa^*(y) = 1, \ x_0 + \frac{1}{1-\pi} \int_{x_0}^\infty (y - x_0) d\kappa^*(y) \leq v, \ \int_\Re f(y) \, d\kappa^*(y) \leq B,$$

$$\int_{x_0}^\infty \gamma_1^* \frac{y - x_0}{1 - \pi} d\kappa^*(y) = \gamma_1^*(v - x_0), \ \int_\Re \gamma_3^* d\kappa^*(y) = \gamma_3^* \quad , \ \int_\Re \gamma_2^* f(y) \, d\kappa^*(y) = \gamma_2^* B. \quad (26)$$

and $(\gamma_1^*, \gamma_2^*, \gamma_3^*) \in \mathcal{R}_3(x_0)$. Furthermore, $\kappa^*$ minimizes $L(\kappa, \boldsymbol{\gamma}^*, x_0)$. From conditions (26), and Lemma D.4, $L(\kappa^*, \boldsymbol{\gamma}^*) = \mathbb{E}_\eta (h(X, \boldsymbol{\gamma}^*, x_0))$, where $X$ is the random variable distributed as $\eta$. Adding the equations in (26), and using the form of $\kappa^*$ from Lemma D.4, we get $\gamma_3^* = 1 - \gamma_1^*(v - x_0) - \gamma_2^* B$.

For $\tilde{\boldsymbol{\gamma}} = (\tilde{\gamma}_1, \tilde{\gamma}_2)$ let,

$$g^L(X, \tilde{\boldsymbol{\gamma}}, v, x_0) := 1 - \tilde{\gamma}_1 \left( v - x_0 - \frac{(X - x_0)_+}{1 - \pi} \right) - \tilde{\gamma}_2 (B - f(X)).$$

and

$$\mathcal{R}_2(x_0, v) := \left\{ \gamma_1 \geq 0, \gamma_2 \geq 0 : \forall y \in \Re, \ g^L(y, (\gamma_1, \gamma_2), x_0, v) \geq 0 \right\}.$$

With this condition on $\gamma_3^*$, the region $\mathcal{R}_3(x_0, v)$ reduces to the region $\mathcal{R}_2(x_0)$. Since we know that the optimal $\boldsymbol{\gamma}^*$ in $\mathcal{R}_3(x_0)$ with the corresponding minimizer, $\kappa^*$, satisfies the conditions in (26) and that $\gamma_3^*$ has the specific form given above, the dual optimal value remains unaffected by adding these conditions as constraints in the dual optimization problem. With these conditions, the dual reduces to

$$\max_{(\gamma_1, \gamma_2) \in \mathcal{R}_2(x_0, v)} \mathbb{E}_\eta \left( \log \left( g^L(X, \boldsymbol{\gamma}, x_0, v) \right) \right),$$

and by strong duality, this is also the value of $\mathrm{KL}_{\inf}(\eta, x)$.

### D.3 Compactness of the dual regions

In this section we show that for valid values of $v$ and $x_0$, the regions $\hat{S}(v)$ and $\mathcal{R}_2(x_0, v)$ are closed and bounded, i.e., compact. Recall that for $v \in D^o = \left( -f^{-1}(B), f^{-1}\left( \frac{B}{1-\pi} \right) \right), x_0 \in C = \left[ -f^{-1}\left( \frac{B}{\pi} \right), f^{-1}\left( \frac{B}{1-\pi} \right) \right], \boldsymbol{\lambda} \in \Re^3$ and $\boldsymbol{\gamma} \in \Re^2$,

$$g^U(X, \boldsymbol{\lambda}, v) = 1 + \lambda_1 v - \lambda_2(1 - \pi) + \lambda_3 (f(X) - B) - \left( \frac{\lambda_1 X}{1 - \pi} - \lambda_2 \right)_+,$$

$$g^L(X, \boldsymbol{\gamma}, v, x_0) = 1 - \gamma_1 \left( v - x_0 - \frac{(X - x_0)_+}{1 - \pi} \right) - \gamma_2(B - f(X)),$$

$$\hat{S}(v) = \left\{ \lambda_1 \geq 0, \lambda_2 \in \Re, \lambda_3 \geq 0 : \forall x \in \Re, \ g^U(x, \boldsymbol{\lambda}, v) \geq 0 \right\},$$

and

$$\mathcal{R}_2(x_0, v) = \left\{ \gamma_1 \geq 0, \gamma_2 \geq 0 : \forall y \in \Re, \ g^L(y, (\gamma_1, \gamma_2), x_0, v) \geq 0 \right\}, \quad (27)$$

where $f(y) = |y|^{1+\epsilon}$ for some $\epsilon > 0$.

We first state a few results, which are easy to prove, and will be used later.

**Lemma D.5.** *For $a > 0, b > 0$, and $z \in \Re$, the minimizer, $x^*$, for $a|x|^{1+\epsilon} + b(x - z)_+$ satisfies*

$$x^* = \begin{cases} 0, & \text{if } z \geq 0 \\ z, & \text{if } z < 0, \text{ and } z \geq -\left( \frac{b}{a(1+\epsilon)} \right)^{\frac{1}{\epsilon}} \\ -\left( \frac{b}{a(1+\epsilon)} \right)^{\frac{1}{\epsilon}}, & \text{otherwise.} \end{cases}$$

**Lemma D.6.** *For $\epsilon > 0$, $b \geq 0$, $a \in \Re$, and $c \geq 0$, the set of $p \geq 0$, $q \geq 0$, satisfying*

$$a - bp + cq - \frac{\epsilon}{p^{\frac{1}{\epsilon}}} \left( \frac{q}{1+\epsilon} \right)^{1+\frac{1}{\epsilon}} \geq 0$$

*is compact, provided $c^{1+\epsilon} \leq b$. Moreover,*

$$p \in \left[ 0, \frac{a}{c^{1+\epsilon} - b} \right]; \qquad q \in \left[ 0, \frac{a}{c - b^{\frac{1}{1+\epsilon}}} \right].$$

**Lemma D.7.** *For $v \in \left[ -f^{-1}(B), f^{-1}\left( \frac{B}{1-\pi} \right) \right)$, $\hat{S}(v)$ is compact.*

*Proof.* For $\boldsymbol{\lambda} \in \hat{S}(v)$,

$$\min_y \; 1 + \lambda_1 v - \lambda_2(1 - \pi) + \lambda_3 \left( f\left( y \right) - B \right) - \left( \frac{\lambda_1 y}{1 - \pi} - \lambda_2 \right) \geq 0,$$

and

$$\min_y \; 1 + \lambda_1 v - \lambda_2(1 - \pi) + \lambda_3 \left( f\left( y \right) - B \right) \geq 0.$$

The l.h.s. in the first inequality above is a convex function which is minimized at $y_1$ for which the derivative of the l.h.s. is 0, while the l.h.s. of the second inequality above is minimized at $y_2 = 0$. Substituting for $y_1$ and $y_2$ in the above inequalities,

$$\frac{1 + \lambda_1 v - \lambda_3 B}{1 - \pi} \geq \lambda_2 \geq \frac{\lambda_1^{1+1/\epsilon}}{\lambda_3^{1/\epsilon}} \frac{\epsilon}{(1+\epsilon)^{1+1/\epsilon}} \frac{1}{(1-\pi)^{1+1/\epsilon}} \frac{1}{\pi} - \frac{1 + \lambda_1 v - \lambda_3 B}{\pi}. \tag{28}$$

Eliminating $\lambda_2$ and simplifying,

$$\frac{\lambda_1^{1+1/\epsilon} \epsilon}{(1+\epsilon)^{1+1/\epsilon}} \frac{1}{(1-\pi)^{1/\epsilon}} - \lambda_3^{\frac{1}{\epsilon}} - \lambda_1 \lambda_3^{\frac{1}{\epsilon}} v + \lambda_3^{1+1/\epsilon} B \leq 0. \tag{29}$$

L.h.s. above is a convex function of $\lambda_1$ which is minimized at

$$\lambda_1^* = \lambda_3 v^\epsilon (1 - \pi)(1 + \epsilon).$$

In particular, (29) holds for $\lambda_1^*$. On substituting $\lambda_1^*$ in (29), we get

$$\lambda_3 \leq \left( B - v^{1+\epsilon}(1 - \pi) \right)^{-1}.$$

Again, observe that l.h.s. in (29) is a convex function of $\lambda_3$ which is minimized at

$$\lambda_3^* = \frac{1 + \lambda_1 v}{B(1 + \epsilon)}.$$

Substituting $\lambda_3^*$ in (29), we get

$$\lambda_1 \leq \left( \left( \frac{B}{\epsilon^\epsilon (1 - \pi)} \right)^{1+\frac{1}{\epsilon}} - v \right)^{-1}.$$

These bounds on $\lambda_1$ and $\lambda_3$, together with the fact that $\lambda_1 \geq 0$ and $\lambda_3 \geq 0$, and bounding $\lambda_2$ using (28), we get that the region specified by $\hat{S}(v)$ is compact. $\qquad\square$

**Lemma D.8.** *For $v \in \left( -f^{-1}(B), f^{-1}\left( \frac{B}{1-\pi} \right) \right]$ and $x_0 \in \left[ -f^{-1}\left( \frac{B}{\pi} \right), v \right)$, $\mathcal{R}_2(x_0, v)$ is compact.*

*Proof.* For $\boldsymbol{\gamma} \in \mathcal{R}_2(v, x_0)$,

$$\min_y \; 1 - \gamma_1 \left( v - x_0 \right) + \gamma_1 \frac{(y - x_0)_+}{1 - \pi} - \gamma_2(B - |y|^{1+\epsilon}) \geq 0. \tag{30}$$

Using Lemma D.5, the region-constraint above can be re-written as

$$0 \leq 1 - \gamma_1(v - x_0) - \gamma_2 B + \begin{cases} 0, & \text{if } x_0 \geq 0, \\ \gamma_2 |x_0|^{1+\epsilon}, & \text{if } 0 \geq x_0 \geq -\left(\frac{\gamma_1}{\gamma_2(1-\pi)(1+\epsilon)}\right)^{\frac{1}{\epsilon}}, \\ -\gamma_1 \frac{x_0}{1-\pi} - \frac{\epsilon}{\gamma_2^{\frac{1}{\epsilon}}}\left(\frac{\gamma_1}{(1-\pi)(1+\epsilon)}\right)^{1+\frac{1}{\epsilon}}, & \text{if } -\left(\frac{\gamma_1}{\gamma_2(1-\pi)(1+\epsilon)}\right)^{\frac{1}{\epsilon}} \geq x_0. \end{cases}$$

Let $x_0 \neq v$. In this case, for $x_0 \geq 0$,

$$\gamma_1 \leq (v - x_0)^{-1} \quad \text{and} \quad \gamma_2 \leq B^{-1}.$$

For $x_0 \neq v$, in the bottom case, optimizing out $\gamma_1$ and setting derivative to 0, together with the fact that $\gamma_1 \geq 0$, we get that the minimizer

$$\gamma_1^* = \max\left\{0, \gamma_2(1+\epsilon)(1-\pi)\left(-\pi x_0 - (1-\pi)v\right)^\epsilon\right\}.$$

This gives

$$\gamma_2 \leq \left(B - \max\left\{0, -\pi x_0 - (1-\pi)v\right\}^{1+\epsilon}\right)^{-1}.$$

Furthermore, the case constraint gives that

$$\gamma_1 \leq (-x_0)^\epsilon \gamma_2 (1-\pi)(1+\epsilon) \leq \frac{(-x_0)^\epsilon(1-\pi)(1+\epsilon)}{B - \max\left\{0, -\pi x_0 - (1-\pi)v\right\}^{1+\epsilon}}.$$

Note that for $v > -f^{-1}(B)$, the denominators in the bounds above is strictly positive.

Let us now consider the center case with $x_0 \neq v$. In this case, the region constraint is

$$0 \leq 1 - \gamma_1(v - x_0) - \gamma_2\left(B - |x_0|^{1+\epsilon}\right).$$

Consider optimizing over $\gamma_1$ to get a bound on $\gamma_2$. Since the coefficient of $\gamma_1$ is positive, optimal value of $\gamma_1$ equals 0, in which case, $\gamma_2$ is at most $(B - |x_0|^{1+\epsilon})^{-1}$ if $|x_0|^{1+\epsilon} \leq B$, otherwise the maximum occurs either at the case-line, or in the bottom case.

Similarly for $\gamma_1$, for $|x_0|^{1+\epsilon} \leq B$, $\gamma_1 \leq (v - x_0)^{-1}$, and the maximum occurs either at the boundary, or in the bottom case, otherwise.

Thus, overall bounds for $x_0 \neq v$ are :

$$\gamma_1 \leq (v - x_0)^{-1}, \quad \gamma_2 \leq B^{-1}, \quad \text{for } x_0 \geq 0,$$
$$\gamma_1 \leq (v - x_0)^{-1}, \quad \gamma_2 \leq (B - |x_0|^{1+\epsilon})^{-1}, \quad \text{for } x_0 \leq 0, \ (-x_0)^{1+\epsilon} \leq B,$$

and for $x_0 \leq 0$, $(-x_0)^{1+\epsilon} > B$,

$$\gamma_1 \leq \frac{(-x_0)^\epsilon(1-\pi)(1+\epsilon)}{B - \max\left\{0, -\pi x_0 - (1-\pi)v\right\}^{1+\epsilon}}, \quad \gamma_2 \leq \left(B - \max\left\{0, -\pi x_0 - (1-\pi)v\right\}^{1+\epsilon}\right)^{-1}.$$

It is important to note that for $v > -f^{-1}(B)$, the bounds above blow-up only for $x_0 = v$. This case will be handled separately. Let us now look at the case when $x_0 = v$. In this case, (30) simplifies to

$$\min_y \ 1 + \gamma_1 \frac{(y - x_0)_+}{1 - \pi} - \gamma_2 B + \gamma_2 |y|^{1+\epsilon} \geq 0.$$

When $x_0 \geq 0$, $y = 0$ is the minimizer for l.h.s. above. Substituting this, we get

$$\gamma_2 \leq B^{-1}.$$

**Remark D.1.** When $x_0 = v \geq 0$, $\gamma_1$ is unbounded. However, if the given probability measure, $\eta$, is such that $\eta(v, \infty) = 0$, then $\gamma_1$ doesn't appear in the objective function. Thus, it is sufficient to restrict $\mathcal{R}_2(x_0(= v), v)$ to the $\gamma_2$ axis. Hence, the modified region is again compact in this special case.

For the other case, i.e., when $x_0 = v < 0$, l.h.s. in minimized at $y^* \in [x_0, 0]$, which is given by

$$y^* = (-1)\left(\gamma_1\gamma_2^{-1}\right)^{1/\epsilon}\left((1-\pi)^{1/\epsilon}(1+\epsilon)^{1/\epsilon}\right)^{-1}.$$

Substituting and simplifying as above, (or substituting y = $x_0$), we get that

$$\gamma_2 \le \left(B - |x_0|^{1+\epsilon}\right)^{-1}.$$

Observe that the denominator is positive for $x_0 = v$. Since $y^* \ge x_0$, we get that

$$\gamma_1 \le (-x_0)^\epsilon \gamma_2 (1-\pi)(1+\epsilon) \le \frac{(-x_0)^\epsilon(1-\pi)(1+\epsilon)}{B - |x_0|^{1+\epsilon}}.$$

Hence, the region $\mathcal{R}_2(x_0, v)$ is compact in this case too. $\qquad\square$

### D.4 Discussion on possibility of uniform priors on the dual feasible regions

Consider an arm distribution $\mu_i$ for $i \in \{1, \dots, K\}$. Since $\hat{S}(c_\pi(\mu_i))$ is compact (see Lemma D.7), uniform measure on the set is well defined. For the region $\mathcal{R}_2(x_\pi(\mu_i), c_\pi(\mu_i))$, whenever $x_\pi(\mu_i) \ne c_\pi(\mu_i)$, the region $\mathcal{R}_2(x_\pi(\mu_i), c_\pi(\mu_i))$ is compact, and uniform prior on this set is well defined. When $\mu_i$ is such that $x_\pi(\mu_i) = c_\pi(\mu_i)$(=v, say), then $\text{Supp}(\mu_i) \subset (-\infty, v]$. In this case $\mathcal{R}_2(v, v)$ is unbounded along the $\gamma_1$ axis. However, from the Remark D.1, it is sufficient to restrict the region along $\gamma_2$ axis, and this restricted region is then compact. We put uniform prior on this modified region.

### D.5 The joint-dual problem

Recall that $\mathcal{L}$ is the class of all probability measures on $\Re$, say $\eta$, with moment bound, i.e. $\mathbb{E}_\eta(f(X)) \le B$, where $f(x) = |x|^{1+\epsilon}$ for some $\epsilon > 0$.

In this sub-section, we look at the joint optimization problem, which appears in the lower bound as a weighted sum of $\text{KL}_{\text{inf}}^{\text{L}}$ and $\text{KL}_{\text{inf}}^{\text{U}}$ for two arms. Specifically, for $\eta_1, \eta_2 \in \mathcal{P}(\Re)$, and non-negative weights $\alpha_1, \alpha_2$, we denote the inner optimization problem in (5) by

$$Z = \inf_{x \le y}\left\{\alpha_1\,\text{KL}_{\text{inf}}^{\text{U}}(\eta_1, y) + \alpha_2\,\text{KL}_{\text{inf}}^{\text{L}}(\eta_2, x)\right\},$$

which is equivalent to the following problem:

$$
\begin{aligned}
\text{minimise}\quad & \alpha_1\,\text{KL}\,(\eta_1, \kappa_1) + \alpha_2\,\text{KL}\,(\eta_2, \kappa_2) \\
\text{subject to}\quad & \kappa_1, \kappa_2 \in \mathcal{L} \\
& c_\pi(\kappa_2) \le c_\pi(\kappa_1)
\end{aligned}
$$

Using the maximization form of CVaR for $\kappa_1$ and the minimization form for $\kappa_2$ from (16) and (15), the above problem is equivalent to

$$
\begin{aligned}
\text{minimise}\quad & \alpha_1\,\text{KL}\,(\eta_1, \kappa_1) + \alpha_2\,\text{KL}\,(\eta_2, \kappa_2) \\
\text{subject to}\quad & \kappa_1, \kappa_2 \in \mathcal{L}, z \in \mathbb{R}, W \in M_+(\Re) \\
& z + \frac{1}{1-\pi}\mathbb{E}_{\kappa_2}\left((X-z)_+\right) \le \frac{1}{1-\pi}\int_{x \in \Re} x\,dW(x) \\
& \forall x: 0 \le dW(x) \le d\kappa_1(x) \\
& \int_{x \in \Re} dW(x) = 1 - \pi.
\end{aligned}
$$

Introducing the dual variables $(\rho_1 \ge 0, \rho_2 \in \Re, \lambda_1 \in \Re, \lambda_2 \ge 0, \gamma_1 \in \Re, \gamma_2 \ge 0, \forall x\, \lambda_3(x) \ge 0)$. Then, single out the minimisation over $z$, the Lagrangian in terms of $\kappa_1, \kappa_2, W$, denoted as

$L(\kappa_1, \kappa_2, W, \boldsymbol{\lambda}, \boldsymbol{\gamma}, \boldsymbol{\rho})$, equals

$$\alpha_1 \int_{\Re} \log\left(\frac{d\eta_1}{d\kappa_1}(y)\right) d\eta_1(y) + \alpha_2 \int_{\Re} \log\left(\frac{d\eta_2}{d\kappa_2}(y)\right) d\eta_2(y) + \lambda_1 \left(\int_{\Re} d\kappa_1(x) - 1\right)$$

$$+ \lambda_2 \left(\int_{\Re} f(x) d\kappa_1(x) - B\right) + \gamma_1 \left(\int_{\Re} d\kappa_2(x) - 1\right) + \gamma_2 \left(\int_{\Re} f(x) d\kappa_2(x) - B\right)$$

$$+ \rho_1 \left(z + \frac{1}{1-\pi}\int (x-z)_+ d\kappa_2(x) - \frac{1}{1-\pi}\int_{\Re} x dW(x)\right)$$

$$+ \int \lambda_3(x)\left(dW(x) - d\kappa_1(x)\right) + \rho_2 \left(\int_{\Re} dW(x) - (1-\pi)\right).$$

Then the Lagrangian dual problem is

$$\min_{z \in \mathbb{R}} \quad \max_{\substack{\rho_1 \geq 0, \rho_2 \in \Re, \\ \lambda_1 \in \Re, \lambda_2 \geq 0, \lambda_3(x) \geq 0, \\ \gamma_1 \in \Re, \gamma_2 \geq 0.}} \quad \min_{\substack{\kappa_1 \in M^+ \\ \kappa_2 \in M^+ \\ W \in M^+}} L(\kappa_1, \kappa_2, W, \boldsymbol{\lambda}, \boldsymbol{\gamma}, \boldsymbol{\rho}). \tag{31}$$

Let $S = \{\rho_1 \geq 0, \rho_2 \in \Re, \lambda_1 \in \Re, \lambda_2 \geq 0, \gamma_1 \in \Re, \gamma_2 \geq 0, \forall x\, \lambda_3(x) \geq 0\}$, and let $S_1$ be the set obtained by intersection of $S$ and set of $(\boldsymbol{\lambda}, \boldsymbol{\gamma}, \boldsymbol{\rho})$ such that

$$\min_{x \in \Re} \lambda_1 + \lambda_2 f(x) - \lambda_2 B - \lambda_3(x) \geq 0, \quad \min_{x \in \Re} \gamma_1 + \gamma_2 f(x) - \gamma_2 B + \rho_1 \frac{(x-z)_+}{1-\pi} \geq 0.$$

**Lemma D.9.** *The Lagrangian dual problem (31) satisfies*

$$\min_{z \in \mathbb{R}} \quad \max_{\substack{\rho_1 \geq 0, \rho_2 \in \Re, \\ \lambda_1 \in \Re, \lambda_2 \geq 0, \\ \lambda_3(x) \geq 0, \\ \gamma_1 \in \Re, \gamma_2 \geq 0.}} \min_{\substack{\kappa_1 \in M^+ \\ \kappa_2 \in M^+ \\ W \in M^+}} L(\kappa_1, \kappa_2, W, \boldsymbol{\lambda}, \boldsymbol{\gamma}, \boldsymbol{\rho}) = \min_{z \in \Re} \quad \max_{(\boldsymbol{\lambda}, \boldsymbol{\gamma}, \boldsymbol{\rho}) \in S_1} \quad \inf_{\substack{\kappa_1 \in M^+ \\ \kappa_2 \in M^+ \\ W \in M^+}} L(\kappa_1, \kappa_2, W, \boldsymbol{\lambda}, \boldsymbol{\gamma}, \boldsymbol{\rho}).$$

*Proof.* Consider $(\boldsymbol{\lambda}, \boldsymbol{\gamma}, \boldsymbol{\rho}) \in S$ and $\boldsymbol{\lambda} \notin S_1$. Then, there exist $y_1 \in \Re$ such that

$$\lambda_1 + \lambda_2 f(y_1) - \lambda_2 B - \lambda_3(y_1) < 0.$$

Set $\kappa_2 = \eta_2$ and define $\kappa_{1M} \in M_+$ such that $\kappa_{1M}(y_1) = M$ and

$$\frac{d\eta_1}{d\kappa_{1M}}(y) = 1, \quad \text{for } y \in \{\text{Supp}(\eta_1) \setminus y_1\}.$$

Then, $L(\kappa_{1M}, \kappa_2, W, \boldsymbol{\lambda}, \boldsymbol{\gamma}, \boldsymbol{\rho})$ equals

$$\alpha_1 \int_{\Re} \log\left(\frac{d\eta_1}{d\kappa_{1M}}\right)(y) d\eta_1(y) + \int_{\Re} \left(\lambda_1 + \lambda_2 f(x) - \lambda_2 B - \lambda_3(x)\right) d\kappa_{1M}(x)$$

$$+ \int_{\Re} \left(\gamma_1 + \gamma_2 f(x) - \gamma_2 B + \frac{\rho}{1-\pi}(x-z)_+\right) d\kappa_2(x)$$

$$+ \int_{\Re} \left(-\frac{\rho_1 x}{1-\pi} + \lambda_3(x) + \rho_2\right) dW(x) - \lambda_1 - \gamma_1 + \rho_1 z - \rho_2(1-\pi).$$

Clearly, the first two terms in the expression above decrease to $-\infty$ as $M$ increases to $\infty$. The other cases, specifically $(\boldsymbol{\gamma}, \boldsymbol{\rho}) \notin S_1$ and $(\boldsymbol{\lambda}, \boldsymbol{\gamma}, \boldsymbol{\rho}) \notin S_1$ can be handled similarly. Thus, the infimum in the inner optimization problem in (31) is $-\infty$, and we get the desired equality. $\qquad \square$

Using arguments similar to those in Lemma D.2 and Lemma D.4, it can be shown that the optimal $\kappa_1^*$ and $\kappa_2^*$, that solve the inner minimization problem in (31) have the following form:

$$\frac{d\kappa_1^*}{d\eta_1}(y) = \frac{\alpha_1}{\lambda_1 + \lambda_2 (f(y) - B) - \lambda_3(y)} \quad \text{for } y \in \text{Supp}(\eta_1), \tag{32}$$

$$\frac{d\kappa_2^*}{d\eta_2}(y) = \frac{\alpha_2}{\gamma_1 + \gamma_2 (f(y) - B) + \rho_1 \frac{(y-z)_+}{1-\pi}} \quad \text{for } y \in \text{Supp}(\eta_2). \tag{33}$$

Furthermore, for $y \in \text{Supp}(\kappa_1^*) \setminus \text{Supp}(\eta_1)$, $\lambda_1 + \lambda_2 (f(y) - B) - \lambda_3(y) = 0$ and for $y \in \text{Supp}(\kappa_2^*) \setminus \text{Supp}(\eta_2)$, $\gamma_1 + \gamma_2 (f(y) - B) + \rho_1 \frac{(y-z)_+}{1-\pi} = 0$.

### D.5.1 Proof of Proposition 3.5

We first show that the dual problem in (31) simplifies to the alternative expression given in the Theorem. Then we argue that both the primal and the dual problems are feasible, and that strong duality holds.

Using the expressions for the optimizers $\kappa_1^*$ and $\kappa_2^*$ from above in the dual in Lemma D.9, the dual in (31) equals

$$\min_{z\in\Re} \max_{(\boldsymbol{\lambda},\boldsymbol{\gamma},\boldsymbol{\rho})\in S_1} \inf_{W\in M^+} \alpha_1\mathbb{E}_{\eta_1}\left(\log\left(\lambda_1 + \lambda_2\left(f\left(Y\right)-B\right)-\lambda_3(Y)\right)\right) - \alpha_1\log\alpha_1 - \alpha_2\log\alpha_2 - \lambda_1$$

$$+ \alpha_2\mathbb{E}_{\eta_2}\left(\log\left(\gamma_1 + \gamma_2\left(f\left(Y\right)-B\right) + \rho_1\frac{(Y-z)_+}{1-\pi}\right)\right) + \alpha_1 + \alpha_2$$

$$+ \int_{\Re} dW(x)\left(-\frac{\rho_1 x}{1-\pi} + \rho_2 + \lambda_3(x)\right) - \rho_2(1-\pi) + \rho_1 z - \gamma_1.$$

Since $W\in M^+$, and if $(\boldsymbol{\lambda},\boldsymbol{\gamma},\boldsymbol{\rho})$ are such that the integrand in the integral above is negative, then the value of the expression above will be $-\infty$. Thus, it suffices to restrict $(\boldsymbol{\lambda},\boldsymbol{\gamma},\boldsymbol{\rho})$ so that this does not happen. Let

$$S_2 = S_1 \cap \left\{(\boldsymbol{\lambda},\boldsymbol{\gamma},\boldsymbol{\rho}) : \forall x, \lambda_3(x) \geq \left(\frac{\rho_1 x - \rho_2(1-\pi)}{1-\pi}\right)_+\right\}.$$

Then the dual problem simplifies to

$$\min_{z\in\Re} \max_{(\boldsymbol{\lambda},\boldsymbol{\gamma},\boldsymbol{\rho})\in S_2} \alpha_1\mathbb{E}_{\eta_1}\left(\log\left(\lambda_1 + \lambda_2\left(f\left(Y\right)-B\right)-\lambda_3(Y)\right)\right) - \alpha_1\log\alpha_1 - \alpha_2\log\alpha_2 - \gamma_1$$

$$+ \alpha_2\mathbb{E}_{\eta_2}\left(\log\left(\gamma_1 + \gamma_2\left(f\left(Y\right)-B\right) + \rho_1\frac{(Y-z)_+}{1-\pi}\right)\right) + \alpha_1 + \alpha_2 + \rho_1 z - \lambda_1 - \rho_2(1-\pi).$$

Let $(\boldsymbol{\lambda},\boldsymbol{\gamma},\boldsymbol{\rho}) = (\lambda_1, \lambda_2, \gamma_1, \gamma_2, \rho_1, \rho_2)$ and

$$S_3 = S \cap \left\{(\boldsymbol{\lambda},\boldsymbol{\gamma},\boldsymbol{\rho}) : \min_y \lambda_1 + \lambda_2\left(f\left(y\right)-B\right) \geq \left(\frac{\rho_1 y - \rho_2}{1-\pi}\right)_+\right\},$$

and

$$S_4 = S_3 \cap \left\{(\boldsymbol{\lambda},\boldsymbol{\gamma},\boldsymbol{\rho}) \in S : \min_y \gamma_1 + \gamma_2\left(f\left(y\right)-B\right) + \rho_1\frac{(y-z)_+}{1-\pi} \geq 0\right\}.$$

Optimizing over the choice of $\lambda_3(x)$, and renaming $\rho_2(1-\pi)$ to $\rho_2$, we get

$$\min_{z\in\Re} \max_{(\boldsymbol{\lambda},\boldsymbol{\gamma},\boldsymbol{\rho})\in S_4} \alpha_1\mathbb{E}_{\eta_1}\left(\log\left(\lambda_1 + \lambda_2\left(f\left(Y\right)-B\right) - \left(\frac{\rho_1 Y - \rho_2}{1-\pi}\right)_+\right)\right) + \alpha_1 + \alpha_2 - \lambda_1 - \gamma_1$$

$$+ \alpha_2\mathbb{E}_{\eta_2}\left(\log\left(\gamma_1 + \gamma_2\left(f\left(Y\right)-B\right) + \rho_1\frac{(Y-z)_+}{1-\pi}\right)\right) - \alpha_1\log\alpha_1 - \alpha_2\log\alpha_2 - \rho_2 + \rho_1 z.$$

$$(34)$$

Optimizing over the common scaling of the dual variables, the inner problem in (34) above can be re-written as

$$\max_{(\boldsymbol{\lambda},\boldsymbol{\gamma},\boldsymbol{\rho})\in S_4} \alpha_1\mathbb{E}_{\eta_1}\left(\log\frac{\lambda_1 + \lambda_2\left(f\left(Y\right)-B\right) - \left(\frac{\rho_1 Y - \rho_2}{1-\pi}\right)_+}{\lambda_1 + \rho_2 - \rho_1 z + \gamma_1}\right) + (\alpha_1+\alpha_2)\log(\alpha_1+\alpha_2)$$

$$+ \alpha_2\mathbb{E}_{\eta_2}\left(\log\frac{\gamma_1 + \gamma_2\left(f\left(Y\right)-B\right) + \rho_1\frac{(Y-z)_+}{1-\pi}}{\lambda_1 + \rho_2 - \rho_1 z + \gamma_1}\right) - \alpha_1\log\alpha_1 - \alpha_2\log\alpha_2.$$

Observe that as earlier, $\lambda_1 + \rho_2 - \rho_1 z + \gamma_1 \geq 0$. This follows from the conditions on the dual variables in $S_4$ and complementary slackness conditions, which hold as the problem is convex optimization and satisfies strong duality (proved later).

Setting $\tilde{\gamma}_1 = \lambda_1 + \rho_2 - \rho_1 z + \gamma_1$, and $\lambda_1 + \rho_2 = \tilde{\lambda}_1$ and substituting in the above expression and renaming the variables, we get

$$\max_{(\boldsymbol{\lambda},\boldsymbol{\gamma},\boldsymbol{\rho}) \in S_5} \alpha_1 \mathbb{E}_{\eta_1} \left( \log \left( \lambda_1 + \lambda_2 \left( f(Y) - B \right) - \rho_2 - \left( \frac{\rho_1 Y - \rho_2}{1 - \pi} \right)_+ \right) \right) - \alpha_1 \log \alpha_1 - \alpha_2 \log \alpha_2$$

$$+ \alpha_2 \mathbb{E}_{\eta_2} \left( \log \left( 1 - \lambda_1 + \gamma_2 \left( f(Y) - B \right) + \rho_1 z + \rho_1 \frac{(Y-z)_+}{1-\pi} \right) \right) + (\alpha_1 + \alpha_2) \log(\alpha_1 + \alpha_2),$$

where $S_5$ is given by intersection of the set $S$ with the sets

$$\left\{ (\boldsymbol{\lambda},\boldsymbol{\gamma},\boldsymbol{\rho}) : \min_y \left( \lambda_1 + \lambda_2 \left( f(y) - B \right) - \rho_2 - \left( \frac{\rho_1 y - \rho_2}{1-\pi} \right)_+ \right) \geq 0 \right\},$$

and

$$\left\{ (\boldsymbol{\lambda},\boldsymbol{\gamma},\boldsymbol{\rho}) : \min_y \left( 1 - \lambda_1 + \gamma_2 \left( f(Y) - B \right) + \rho_1 z + \rho_1 \frac{(Y-z)_+}{1-\pi} \right) \geq 0 \right\}.$$

Re-parameterize again by setting $\tilde{\lambda}_1 = \lambda_1 - 1/2$ and scaling every dual variable by $1/2$, we get the desired dual formulation.

It now suffices to show that both the primal and dual problems are feasible, and strong duality holds.

Consider $(\boldsymbol{\lambda},\boldsymbol{\gamma},\boldsymbol{\rho})^1 = (1, 0, 0, 1, 0, 0, 0)$. To show that dual is feasible, it suffices to show

$$\min_{\substack{\kappa_1 \in M^+ \\ \kappa_2 \in M^+}} L(\kappa_1, \kappa_2, , W, (\boldsymbol{\lambda},\boldsymbol{\gamma},\boldsymbol{\rho})^1) = \min_{\substack{\kappa_1 \in M^+ \\ \kappa_2 \in M^+}} \alpha_1 \, \mathrm{KL}(\eta_1, \kappa_1) + \alpha_2 \, \mathrm{KL}(\eta_2, \kappa_2) - 2$$

$$+ \int_{\Re} d\kappa_1(y) + \int_{\Re} d\kappa_2(y) > -\infty.$$

Let $\tilde{\kappa}_1$ and $\tilde{\kappa}_2$ be the minimizers in the above expression. Then, $\mathrm{Supp}(\tilde{\kappa}_1) = \mathrm{Supp}(\eta_1)$ and $\mathrm{Supp}(\tilde{\kappa}_2) = \mathrm{Supp}(\eta_2)$, and from (32) and (33),

$$\frac{d\tilde{\kappa}_1}{d\eta_1}(y) = \alpha_1 \quad \text{and} \quad \frac{d\tilde{\kappa}_2}{d\eta_2}(y) = \alpha_2.$$

We next argue the feasibility of primal problem, and show that strong duality holds. Consider $\kappa_1 = \kappa_2 = p_1 \eta_1 + p_2 \eta_2 + (1 - p_1 - p_2)\delta_0$, where, $p_1, p_2 \in (0, 1)$ and $\mathbb{E}_{\kappa_1}(f(X)) < B$. Clearly, $\mathrm{KL}(\eta_1, \kappa_1) < \infty$ and $\mathrm{KL}(\eta_2, \kappa_2) < \infty$ and are feasible for the primal problem. Furthermore, the measures $\tilde{\kappa}_1 = (1 - \epsilon)\delta_{-f^{-1}(B)} + \epsilon \delta_0$ and $\tilde{\kappa}_2 = (\pi + \epsilon)\delta_0 + (1 - \pi - \epsilon)\delta_{f^{-1}\left(\frac{B}{\pi}\right)}$ lie in the interior of the primal region. Hence, strong duality holds.

## E   Description of the C-tracking rule of [27] used in Section 4

Recall that at each time $n$, the algorithm evaluates the optimization problem (5) for the projected empirical distribution vector, i.e., it computes $V(\Pi(\hat{\mu}(n)))$. Let $t^*(\Pi(\hat{\mu}(n)))$ be a maximizer. In order to track these with some forced-exploration, for $\zeta \in (0, \frac{1}{K}]$, let $t^\zeta(\Pi(\hat{\mu}(n)))$ be a $L^\infty$ projection of $t^*(\Pi(\hat{\mu}(n)))$ onto $\Sigma_K^\zeta = \{(t_1, \ldots, t_K) \in [\zeta, 1] : t_1 + \cdots + t_K = 1\}$. The algorithm sets $\zeta_n = (K^2 + n)^{-\frac{1}{2}}/2$, and chooses

$$A_{n+1} \in \operatorname*{argmax}_{1 \leq a \leq K} \sum_{s=0}^{n} \left( t_a^{\zeta_n}(\Pi(\hat{\mu}(n))) - N_a(n) \right).$$

See [27, Section 3.1] for details of the C-tracking rule, and its properties.

## F   Theoretical guarantees for the algorithm: Proof for Theorem 4.1

## F.1 Proof of Theorem 4.1: $\delta$-correctness

In this section, we prove that the algorithm presented is $\delta$-correct, i.e., the first part of Theorem 4.1 holds. Recall that the error occurs when at the stopping time $\tau_\delta$, the arm with minimum CVaR is not arm 1. Let the event $\{\hat{\mu}(n) \text{ suggests arm j as answer}\}$ be denoted by $\mathcal{E}_n(j)$. Then, using the stopping rule in (7), $\mathcal{E}$ is contained in

$$\left\{ \exists n: \bigcup_{i \neq 1} \left\{ \min_{a \neq i} \inf_{x \leq y} \left\{ N_i(n) \operatorname{KL}_{\inf}^U(\hat{\mu}_i(n), y) + N_a(n) \operatorname{KL}_{\inf}^L(\hat{\mu}_a(n), x) \right\} \geq \beta; \ \mathcal{E}_n(i) \right\} \right\},$$

which is further contained in

$$\left\{ \exists n: \bigcup_{i \neq 1} \left\{ \inf_{x \leq y} \left\{ N_i(n) \operatorname{KL}_{\inf}^U(\hat{\mu}_i(n), y) + N_1(n) \operatorname{KL}_{\inf}^L(\hat{\mu}_1(n), x) \right\} \geq \beta; \ \mathcal{E}_n(i) \right\} \right\}. \quad (35)$$

Clearly, $x = c_\pi(\mu_1)$ and $y = c_\pi(\mu_i)$ are feasible for the infimum problem above. Using these with the union bound, the probability of the error event is bounded by

$$\sum_{i=2}^{K} \mathbb{P} \left( \exists n: \ N_i(n) \operatorname{KL}_{\inf}^U(\hat{\mu}_i(n), c_\pi(\mu_i)) + N_1(n) \operatorname{KL}_{\inf}^L(\hat{\mu}_1(n), c_\pi(\mu_1)) \geq \beta \right). \quad (36)$$

Whence, it is sufficies to show that each summand in (36) is at most $\frac{\delta}{K-1}$.

Recall that $f^{-1}(c) = \max \{y : f(y) = c\} = c^{\frac{1}{1+\epsilon}}$, and for $v \in D^o = \left( -f^{-1}(B), f^{-1}\left( \frac{B}{1-\pi} \right) \right)$, $x_0 \in C = \left[ -f^{-1}\left( \frac{B}{\pi} \right), v \right]$,

$$\hat{S}(v) := \left\{ \lambda_1 \geq 0, \lambda_2 \in \Re, \lambda_3 \geq 0 : \forall x \in \Re, \ g^U(x, \boldsymbol{\lambda}, v) \geq 0 \right\},$$

and

$$\mathcal{R}_2(x_0, v) := \left\{ \gamma_1 \geq 0, \gamma_2 \geq 0 : \forall y \in \Re, g^L(y, \boldsymbol{\gamma}, v, x_0) \geq 0 \right\},$$

where,

$$g^U(X, \boldsymbol{\lambda}, v) = 1 + \lambda_1 v - \lambda_2(1 - \pi) + \lambda_3 (f(X) - B) - \left( \frac{\lambda_1 X}{1 - \pi} - \lambda_2 \right)_+,$$

and

$$g^L(X, \boldsymbol{\gamma}, v, x_0) = 1 - \gamma_1 \left( v - x_0 - \frac{(X - x_0)_+}{1 - \pi} \right) - \gamma_2(B - f(X)).$$

Clearly, $g^U$ and $g^L$ are concave functions of $\boldsymbol{\lambda}$ and $\boldsymbol{\gamma}$. Recall from Theorems 3.4 that for each arm i,

$$N_i(n) \operatorname{KL}_{\inf}^U(\hat{\mu}_i(n), c_\pi(\mu_i)) = \max_{\boldsymbol{\lambda} \in \hat{S}(c_\pi(\mu_i))} \sum_{j=1}^{N_i(n)} \log \left( g^U(X_j^i, \boldsymbol{\lambda}, c_\pi(\mu_i)) \right), \quad (37)$$

and

$$N_i(n) \operatorname{KL}_{\inf}^L(\hat{\mu}_i(n), c_\pi(\mu_i)) \leq \max_{\boldsymbol{\gamma} \in \mathcal{R}_2(x_\pi(\mu_i), c_\pi(\mu_i))} \sum_{j=1}^{N_i(n)} \log \left( g^L(X_j^i, \boldsymbol{\gamma}, c_\pi(\mu_i), x_\pi(\mu_i)) \right), \quad (38)$$

where, $X_j^i : j \in \{1, \ldots, N_i(n)\}$ are samples from $\mu_i$.

The following lemma bounds the maximum of a sum of exp-concave functions, i.e., functions whose exponentials are concave. It is essentially the regret bound for the continuous exponentially weighted average predictor, which was shown for the core portfolio optimisation case by [10] and stated in general by [29, Theorem 7].

**Lemma F.1.** *Let $\Lambda \subseteq \mathbb{R}^d$ be a compact and convex subset and $q$ be the uniform distribution on $\Lambda$. Let $g_t : \Lambda \to \mathbb{R}$ be any series of exp-concave functions. Then*

$$\max_{\boldsymbol{\lambda} \in \Lambda} \sum_{t=1}^{T} g_t(\boldsymbol{\lambda}) \leq \log \mathbb{E}_{\boldsymbol{\lambda} \sim q} \left( e^{\sum_{t=1}^{T} g_t(\boldsymbol{\lambda})} \right) + d \log(T + 1) + 1.$$

Let $q_{1i}$ be a uniform prior on the set $\hat{S}(c_\pi(\mu_i))$, and $q_{2i}$ be the uniform prior on the set $\mathcal{R}_2(x_\pi(\mu_i), c_\pi(\mu_i))$. See Sections D.3 and D.4 for a discussion on the possibility of having uniform priors on these sets. For samples $X_j^i : j \in \{1, \ldots, N_i(n)\}$, define

$$L_i(n) = \mathbb{E}_{\boldsymbol{\gamma} \sim q_{2i}} \left( \prod_{j=1}^{N_i(n)} g^L(X_j^i, \boldsymbol{\gamma}, c_\pi(\mu_i), x_\pi(\mu_i)) \Big| X_1^i, \ldots, X_{N_i(n)}^i \right),$$

and

$$U_i(n) = \mathbb{E}_{\boldsymbol{\lambda} \sim q_{1i}} \left( \prod_{j=1}^{N_i(n)} g^U(X_j^i, \boldsymbol{\lambda}, c_\pi(\mu_i)) \Big| X_1^i, \ldots, X_{N_i(n)}^i \right).$$

Then, using Lemma F.1 with $g_t(\boldsymbol{\lambda}) = \log g^U(X_t, \boldsymbol{\lambda}, c_\pi(\mu_i))$, $d = 3$, and (37), on each sample path, we have

$$N_i(n) \, \mathrm{KL}_{\inf}^U(\hat{\mu}_i(n), c_\pi(\mu_i)) \leq \log U_i(n) + 3\log(N_i(n) + 1) + 1.$$

Also using Lemma F.1 with $g_t(\boldsymbol{\gamma}) = g^L(X_t, \boldsymbol{\gamma}, c_\pi(\mu_i), x_\pi(\mu_i))$, $d = 2$, (38) and Remark D.1, on each sample path,

$$N_i(n) \, \mathrm{KL}_{\inf}^L(\hat{\mu}_i(n), c_\pi(\mu_i)) \leq \log L_i(n) + 2\log(N_i(n) + 1) + 1, \quad \text{a.s.}$$

For each arm $i$, let

$$Y_i^L(n) = N_i(n) \, \mathrm{KL}_{\inf}^L(\hat{\mu}_i(n), c_\pi(\mu_i)) - 2\log(N_i(n) + 1) - 1. \tag{39}$$

and

$$Y_i^U(n) = N_i(n) \, \mathrm{KL}_{\inf}^U(\hat{\mu}_1(n), c_\pi(\mu_i)) - 3\log(N_i(n) + 1) - 1. \tag{40}$$

Then we have that

$$e^{Y_i^L(n)} \leq L_i(n) \quad \text{and} \quad e^{Y_i^U(n)} \leq U_i(n), \ \ a.s.$$

Furthermore, it is easy to verify that for each arm i, $L_i(n)$ and $U_i(n)$ are non-negative, super-martingales satisfying $\mathbb{E}(U_i(n)) \leq 1$ and $\mathbb{E}(L_i(n)) \leq 1$. Thus, $U_i(n)L_1(n)$ is a non-negative super-martingale with mean at most 1, and satisfies that the event

$$\left\{ \exists n : N_i(n) \, \mathrm{KL}_{\inf}^U(\hat{\mu}_i(n), c_\pi(\mu_i)) + N_1(n) \, \mathrm{KL}_{\inf}^L(\hat{\mu}_1(n), c_\pi(\mu_1)) \geq \beta(n, \delta) \right\}$$

is contained in

$$\left\{ \exists n : L_1(n)U_i(n) \geq \frac{K-1}{\delta} \right\}.$$

Using Ville's inequality (see, [55]), the probability of the above event is bounded by $\frac{\delta}{K-1}$. $\quad\square$

### F.1.1 Proof of Lemma F.1

Recall that $q$ is uniform over $\Lambda$. Let $\lambda^*$ denote the maximizer for $\max_{\lambda \in \Lambda} \sum_1^T g_t(\lambda)$. Then, for any distribution $r$ over $\Lambda$, Donsker-Varadhan variational form for $\mathrm{KL}(r, q)$ gives that

$$\max_{\lambda \in \Lambda} \sum_{t=1}^T g_t(\lambda) \leq \mathrm{KL}(r, q) + \sum_{t=1}^T \mathbb{E}_{\lambda \sim r} (g_t(\lambda^*) - g_t(\lambda)) + \log \mathbb{E}_{\lambda \sim q} \left( e^{\sum_1^T g_t(\lambda)} \right). \tag{41}$$

Fix $\alpha \in (0, 1)$. Define the set $\tilde{\Lambda} = \{\alpha\lambda^* + (1 - \alpha)\lambda_0 : \lambda_0 \in \Lambda\}$, and choose $r$ to be uniform over $\tilde{\Lambda}$. Then, $\mathrm{KL}(r, q) = -d\log(1 - \alpha)$. Moreover, since $e^{g_t}$ is concave, for $\lambda \in \tilde{\Lambda}$ such that $\lambda = \alpha\lambda^* + (1 - \alpha)\lambda_0$,

$$e^{g_t(\lambda)} \geq \alpha e^{g_t(\lambda^*)} + (1 - \alpha)e^{g_t(\lambda_0)} \geq \alpha e^{g_t(\lambda^*)}.$$

Taking negative logarithm of the above inequality, we get $g_t(\lambda^*) - g_t(\lambda) \leq -\log\alpha$, for all $\lambda \in \tilde{\Lambda}$. Using this and the bound for $\mathrm{KL}(r, q)$ in (41), along with setting $\alpha = \frac{T}{T+d}$, we get

$$\max_{\lambda \in \Lambda} \sum_{t=1}^T g_t(\lambda) \leq (T + d)H_2\left(\frac{T}{T+d}\right) + \log \mathbb{E}_{\lambda \sim q}\left(e^{\sum_1^T g_t(\lambda)}\right),$$

where $H_2(a)$ is the entropy of Bernoulli(a) random variable. The required inequality then follows by observing that $(T + d)H_2\left(\frac{T}{T+d}\right) \leq d\log(T + 1) + 1$. $\quad\square$

### F.2 Sample complexity

We now prove that the sample complexity of the algorithm matches the lower bound asymptotically, as $\delta \to 0$, i.e., it satisfies

$$\limsup_{\delta \to 0} \frac{\mathbb{E}_\mu(\tau_\delta)}{\log \frac{1}{\delta}} \leq \frac{1}{V(\mu)}.$$

The proof works for any projection map $\Pi$, which is continuous at $\mathcal{L}$. However, we give proof for the map that projects onto the element in $\mathcal{L}$ which is closest in the Kolmorogov metric, i.e., $\Pi = (\Pi_1, \Pi_2, \ldots, \Pi_K)$, where

$$\Pi_i(\eta) \in \operatorname*{argmin}_{\kappa \in \mathcal{L}} d_K(\kappa, \eta), \quad \text{and} \quad d_K(\kappa, \eta) = \sup_{x \in \Re} |F_\kappa(x) - F_\eta(x)|,$$

and $F_\kappa$ and $F_\eta$ denote the CDF functions for the measures $\eta$ and $\kappa$.

Let $\epsilon' > 0$ and $n \in \mathbb{N}$. Define $\mathcal{I}_{\epsilon'} \triangleq B_\zeta(\mu_1) \times B_\zeta(\mu_2) \times \ldots \times B_\zeta(\mu_K)$, where $B_\zeta(\mu_i) = \{\kappa \in \mathcal{P}(\Re) : d_K(\kappa, \mu_i) \leq \zeta\}$, and $\zeta > 0$ is chosen to satisfy the following:

$$\mu' \in \mathcal{I}_{\epsilon'} \implies \forall t' \in t^*(\Pi(\mu')), \ \exists t \in t^*(\mu) \text{ s.t. } \|t' - t\|_\infty \leq \epsilon'.$$

Observe that for $\zeta \to 0$, probability measures in $B_\zeta(\mu_i)$ weakly converge to $\mu_i$, for all $i$. Also, for all $\kappa \in B_\zeta(\mu_i)$, $d_K(\kappa, \mu_i) \leq \zeta$ which implies that $d_K(\Pi(\kappa), \kappa) \leq \zeta$, and hence, $d_K(\Pi(\kappa), \mu_i) \leq 2\zeta$, where the last inequality follows from triangle inequality for $d_K$.

Recall that $\mu \in \mathcal{M}$ is such that $-f^{-1}(B) < c_\pi(\mu_1) < \max_{j \neq 1} c_\pi(\mu_j) < f^{-1}\left(\frac{B}{1-\pi}\right)$, where $f^{-1}(c) := \max\{y : f(y) = c\}$. Existence of $\zeta = \zeta(\epsilon)$ is then guaranteed by the upper-hemicontinuity of the set $t^*(\mu)$ (Lemma 3.3). See, [21, Theorem 9] for a proof in the parametric setting, when the optimal proportions are only upper-hemicontinuous.

For $T \in \mathbb{N}$, set $l_0(T) = T^{1/4}$, and define

$$\mathcal{G}_T(\epsilon') = \bigcap_{n=l_0(T)}^{T} \{\hat{\mu}(n) \in \mathcal{I}_{\epsilon'}\}.$$

Let $\mu'$ be a vector of $K$, 1-dimensional distributions from $\mathcal{P}(\Re)$, $[K] = \{1, \ldots, K\}$, and let $t' \in \Sigma_K$. Define

$$g(\mu', t') \triangleq \max_{a \in [K]} \min_{b \neq a} \inf_{x \in \left[-f^{-1}(B), f^{-1}\left(\frac{B}{1-\pi}\right)\right]} \left(t'_a \operatorname{KL}^{\mathrm{U}}_{\mathrm{inf}}(\mu'_1, x) + t'_b \operatorname{KL}^{\mathrm{L}}_{\mathrm{inf}}(\mu'_b, x)\right).$$

Note that, for $\mu \in (\mathcal{P}(\Re))^K$, from Lemma C.3 and Berge's Theorem (see, [8, Theorem 2, Page 116]), $g(\mu, t)$ is a jointly lower-semicontinuous function of $(\mu, t)$. Let $\|.\|_\infty$ be the maximum norm in $\Re^K$, and

$$C^*_{\epsilon'}(\mu) \triangleq \inf_{\substack{\mu' \in \mathcal{I}_{\epsilon'} \\ t' : \inf_{t \in t^*(\mu)} \|t' - t\|_\infty \leq 4\epsilon'}} g(\mu', t').$$

From Lemma F.4, for $T \geq T_{\epsilon'}$, on $\mathcal{G}_T(\epsilon')$, for $t \geq T^{1/4}$, the modified log generalized likelihood ratio statistic for $\hat{\mu}(n)$, used in the stopping rule, is given by $Z(n) = \max_a \min_{b \neq a} Z_{a,b}(n)$, where

$$Z_{a,b}(n) = n \inf_{x \in \left[-f^{-1}(B), f^{-1}\left(\frac{B}{1-\pi}\right)\right]} \left(\frac{N_a(n)}{n} \operatorname{KL}^{\mathrm{U}}_{\mathrm{inf}}(\hat{\mu}_a(n), x) + \frac{N_b(n)}{n} \operatorname{KL}^{\mathrm{L}}_{\mathrm{inf}}(\hat{\mu}_b(n), x)\right).$$

In particular, on $\mathcal{G}_T(\epsilon')$, for $T \geq T_{\epsilon'}$ and $n \geq l_0(T)$,

$$Z(n) = n \max_a \min_{b \neq a} \inf_{x \in \left[-f^{-1}(B), f^{-1}\left(\frac{B}{1-\pi}\right)\right]} \left(\frac{N_a(n)}{n} \operatorname{KL}^{\mathrm{U}}_{\mathrm{inf}}(\hat{\mu}_a(n), x) + \frac{N_b(n)}{n} \operatorname{KL}^{\mathrm{L}}_{\mathrm{inf}}(\hat{\mu}_b(n), x)\right)$$

$$= n \, g\left(\hat{\mu}(n), \left\{\frac{N_1(n)}{n}, \ldots, \frac{N_K(n)}{n}\right\}\right)$$

$$\geq n \, C^*_{\epsilon'}(\mu).$$

Furthermore, the stopping time is at most $\inf\{n \geq l_0(T) : Z(n) \geq \beta(n,\delta), l \in \mathbb{N}\}$. For $T \geq T_{\epsilon'}$, on $\mathcal{G}_T(\epsilon')$,

$$
\begin{aligned}
\min\{\tau_\delta, T\} &\leq \sqrt{T} + \sum_{l=\sqrt{T}+1}^{T} \mathbf{1}\left(t < \tau_\delta\right) \\
&\leq \sqrt{T} + \sum_{l=\sqrt{T}+1}^{T} \mathbf{1}\left(Z\left(l\right) < \beta\left(l,\delta\right)\right) \\
&\leq \sqrt{T} + \sum_{l=\sqrt{T}+1}^{T} \mathbf{1}\left(l < \frac{\beta\left(l,\delta\right)}{C_{\epsilon'}^*(\mu)}\right) \\
&\leq \sqrt{T} + \frac{\beta(T,\delta)}{C_{\epsilon'}^*(\mu)}.
\end{aligned}
\tag{42}
$$

Define,

$$
T_0(\delta) = \inf\left\{l \in \mathbb{N} : \sqrt{l} + \frac{\beta(l,\delta)}{C_{\epsilon'}^*(\mu)} \leq l\right\}.
$$

On $\mathcal{G}_T$, for $T \geq \max\{T_0(\delta), T_{\epsilon'}\}$, from (42) and definition of $T_0(\delta)$,

$$
\min\{\tau_\delta, T\} \leq \sqrt{T} + \frac{\beta(T,\delta)}{C_{\epsilon'}^*(\mu)} \leq T,
$$

which gives that for such a $T$, $\tau_\delta \leq T$. Thus, for $T \geq T_0(\delta)$, we have $\mathcal{G}_T(\epsilon') \subset \{\tau_\delta \leq T\}$ and hence, $\mathbb{P}_\mu(\tau_\delta > T) \leq \mathbb{P}_\mu(\mathcal{G}_T^c)$. Since $\tau_\delta \geq 0$,

$$
\mathbb{E}_\mu(\tau_\delta) \leq T_0(\delta) + T_{\epsilon'} + \sum_{T=T_0(\delta)+1}^{\infty} \mathbb{P}_\mu\left(\mathcal{G}_T^c(\epsilon')\right).
\tag{43}
$$

For $\tilde{e} > 0$, it can be shown that

$$
\limsup_{\delta \longrightarrow 0} \frac{T_0(\delta)}{\log(1/\delta)} \leq \frac{(1+\tilde{e})}{C_{\epsilon'}^*(\mu)}.
\tag{44}
$$

Then, from (43), (44), and Lemma F.5,

$$
\limsup_{\delta \to 0} \frac{\mathbb{E}_\mu(\tau_\delta)}{\log(1/\delta)} \leq \frac{(1+\tilde{e})}{C_{\epsilon'}^*(\mu)}.
$$

From lower-semicontinuity of $g(\mu', t')$ in $(\mu', t')$ for $\mu' \in (\mathcal{P}(\Re))^K$, it follows that $\liminf_{n\to\infty} C_{\epsilon'}^*(\mu) \geq V(\mu)$. First letting $\tilde{e} \to 0$ and then letting $\epsilon' \to 0$, we get

$$
\limsup_{\delta \to 0} \frac{\mathbb{E}_\mu(\tau_\delta)}{\log(1/\delta)} \leq \frac{1}{V(\mu)}.
$$

**Lemma F.2** ([27], Lemma 7). *For $n \geq 1$ and $a \in [K]$, the C-tracking rule ensures that $N_a(n) \geq \sqrt{n + K^2} - 2K$ and that*

$$
\max_{a \in [K]} \left| N_a(n) - \sum_{s=0}^{n-1} t_a(s) \right| \leq K(1 + \sqrt{n}), \quad \text{where} \quad t(s) \in t^*\left(\Pi(\hat{\mu}(s))\right).
$$

**Lemma F.3** ([21], Lemma 33). *Let $\epsilon > 0$ and $A \subset \Sigma_K$ be a convex set and let $t(1), t(2), \ldots, t(n) \in \Sigma_K$ be such that for $s \in \{1, \ldots, n\}$, $\inf_{t \in A} \|t(s) - t\|_\infty \leq \epsilon'$. Then $\inf_{t \in A} \|\frac{1}{n}\sum_{s=1}^{n} t(s) - t\|_\infty \leq \epsilon$.*

**Lemma F.4.** *For $\epsilon' > 0$, there exists a constant $T_\epsilon'$ such that for $T \geq T_{\epsilon'}$, it holds that on $\mathcal{G}_T$ for tracking rule*

$$
\forall n \geq \sqrt{T}, \quad \inf_{t^* \in t^*(\mu)} \max_{a \in [K]} \left| \frac{N_a(n)}{n} - t_a^* \right| \leq 3\epsilon'.
$$

*Proof.* The proof follows along the lines of [27, Lemma 20] and [21, Lemma 35]. For any $n \geq \sqrt{T} = l_0(T)$, using the Lemma F.2, for all a,

$$\inf_{t \in t^*(\mu)} \max_{a \in [K]} \left| \frac{N_a(n)}{n} - t_a \right| \leq \max_{a \in [K]} \left| \frac{N_a(n)}{n} - \frac{1}{n} \sum_{s=0}^{n-1} t_a(s) \right| + \inf_{t \in t^*(\mu)} \max_{a \in [K]} \left| \frac{1}{n} \sum_{s=0}^{n-1} t_a(s) - t_a \right|$$

$$\leq \frac{K(1 + \sqrt{n})}{n} + \frac{l_0(T)}{n} + \inf_{t \in t^*(\mu)} \left\| \frac{1}{n} \sum_{s=l_0(T)}^{n-1} (t(s) - t) \right\|_\infty.$$

On the set $\mathcal{G}_T$, from the definition of this set, for all $n \geq l_0(T)$, $\forall t' \in t^* (\Pi(\hat{\mu}(n)))$, $\inf_{t \in t^*(\mu)} \|t' - t\|_\infty \leq \epsilon'$. Since $t^*(\mu)$ is a convex set, by Lemma F.3, the last term in the expression above is at most $\epsilon'$. Thus,

$$\inf_{t \in t^*(\mu)} \max_{a \in [K]} \left| \frac{N_a(n)}{n} - t_a \right| \leq \frac{2K}{l_0(T)} + \frac{1}{l_0(T)} + \epsilon' \leq 3\epsilon',$$

for $T \geq ((2K + 1)/2\epsilon')^4$. □

**Lemma F.5.**

$$\limsup_{\delta \to 0} \frac{\sum_{T=1}^{\infty} \mathbb{P}_\mu(\mathcal{G}_T^c(\epsilon'))}{\log(1/\delta)} = 0.$$

*Proof.* Recall that for $T \in \mathbb{N}$, $l_0(T) = T^{1/4}$, and

$$\mathcal{G}_T(\epsilon') = \bigcap_{n=l_0(T)}^{T} \{\hat{\mu}(n) \in \mathcal{I}_{\epsilon'}\}.$$

Using union bounds,

$$\mathbb{P}_\mu(\mathcal{G}_T^c(\epsilon')) \leq \sum_{l=l_0(T)}^{T} \mathbb{P}_\mu(\hat{\mu}(l) \notin \mathcal{I}_{\epsilon'}) \leq \sum_{l=l_0(T)}^{T} \sum_{a=1}^{K} \mathbb{P}\left(\sup_x \left|F_{\hat{\mu}_a(l)}(x) - F_a(x)\right| \geq \epsilon'\right).$$

From Lemma F.2, C-Tracking ensures at least $\sqrt{l}/2$ samples to each arm till time $l$. Using this, each summand in the above can be bounded as follows:

$$\mathbb{P}\left(\sup_x \left|F_{\hat{\mu}_a(l)}(x) - F_a(x)\right| \geq \epsilon'\right) \leq \mathbb{P}\left(\sup_x \left|F_{\hat{\mu}_a(l)}(x) - F_a(x)\right| \geq \epsilon'; N_a(l) \geq \frac{\sqrt{l}}{2}\right).$$

R.h.s. in the above inequality can be bounded using union bound and DKW inequality by

$$\sum_{j=\sqrt{l}/2}^{l} e^{-2j\epsilon'^2} = e^{-\epsilon'^2\sqrt{l}} \left(1 - e^{-2\epsilon'^2}\right)^{-1}.$$

Thus,

$$\mathbb{P}_\mu(\mathcal{G}_T^c(\epsilon)) \leq KTe^{-\epsilon'^2 T^{1/8}} \left(1 - e^{-2\epsilon'^2}\right)^{-1},$$

completing the proof. □

# G  Computing the projection in Kolmogorov metric

In this section, we describe a method for computing the projection of $F$ (where the typical application has $F$ as the empirical CDF) onto $\mathcal{L}$ in the Kolmogorov metric, i.e.

$$\operatorname*{argmin}_{G:\, \mathbb{E}_G(f(X)) \leq B} d_K(F, G),$$

where recall that

$$d_K(F, G) := \sup_{x \in \Re} |F(x) - G(x)|.$$

To project $F$ onto $\mathcal{L}$ in the Kolmogorov metric, we will essentially relocate equal mass from the extreme left and right tails to $0$. This is because relocating some mass, say $\xi$, from points in the right tail already incurs $\xi$ cost in the Kolmogorov metric (observe the shift in the CDF at $0$ due to this mass). Thus relocating an equal mass from points in the left tail to $0$ is for free in the metric under consideration. However, if the left tail does not have the required $\xi$ mass, then the additional mass needed to bring down the $(1 + \epsilon)^{th}$ moment will only be relocated from points in the right tail. Lemma G.1 essentially shows that there exists a projection of $F$ on $\mathcal{L}$ which satisfies this.

**Lemma G.1.** *There exists $z \geq 0$ such that an optimal measure in $\mathcal{L}$ has CDF of the following form:*

$$G_z(x) = \begin{cases} \max\{0, F(x) - z\}, & \text{for } x < 0 \\ \min\{1, F(x) + z\}, & \text{for } x \geq 0. \end{cases}$$

*Proof.* Let $G^*$ be a minimiser, and let $z = d_K(F, G^*)$. Clearly, $G_z$ as defined above is a CDF, and $d_K(F, G_z) \leq z$. It then suffices to show that $G_z$ is also a feasible solution, i.e., $\mathbb{E}_{G_z}\left(|X|^{1+\epsilon}\right) \leq B$.

For $\epsilon > 0$, since $f(x) = |x|^{1+\epsilon}$ is a non-negative function, and $f^{-1}(c) := \max\{y : f(y) = c\}$,

$$\mathbb{E}_{G_z}(f(X)) = \int_{x \geq 0} \mathbb{P}_{G_z}(f(X) \geq x)\, dx$$

$$= \int_{x \geq 0} \mathbb{P}_{G_z}\left(X \geq f^{-1}(x)\right) dx + \int_{x \geq 0} \mathbb{P}_{G_z}\left(X \leq -f^{-1}(x)\right) dx. \tag{45}$$

Since $d_K(F, G^*) \leq z$,

$$G^*(x) \geq G_z(x), \text{ for } x < 0 \quad \text{and} \quad G^*(x) \leq G_z(x), \text{ for } x \geq 0.$$

Using this in (45), we have that

$$\mathbb{E}_{G_z}(f(X)) \leq \int_{x \geq 0} \mathbb{P}_{G^*}\left(X \geq f^{-1}(x)\right) dx + \int_{x \geq 0} \mathbb{P}_{G^*}\left(X \leq -f^{-1}(x)\right) dx = \mathbb{E}_{G^*}(f(X)),$$

which is bounded from above by $B$, as desired. $\square$

Since there is an optimizer of the specific form considered in Lemma G.1, to compute the projection, it suffices to search over only such probability measures. Hence, it only remains to compute the smallest $z$ for which $G_z \in \mathcal{L}$ is feasible. We can see from the expression in (45) that $\mathbb{E}_{X \sim G_z}[f(X)]$ is a convex, decreasing function of $z$, which is moreover piecewise linear for discrete (i.e. empirical) $F$. This means we can use many techniques to find the argument $z$ for which it first reaches $0$ (binary search, Newton, explicitly enumerating the segments/knots, etc.).

## H  Discussion on the VaR problem

In the main text, we mainly focused on the minimum CVaR arm identification problem. In this section we formally present the ideas for the analogous approach of the optimum VaR arm identification problem. As before, our treatment is based on the lower bound problem. In this section we will not (need to) impose a moment constraint, i.e., $\mu \in \mathcal{P}(\Re)$, as the VaR lower bound is defined without it. The main object of study is the optimization problem that appears in the lower bound, given below:

$$V(\mu) = \sup_{t \in \Sigma_K} \inf_{\nu \in \mathcal{A}_1^c} \sum_{a=1}^{K} t_a \, \mathrm{KL}(\mu_a, \nu_a), \text{ and } \mathcal{A}_j^c = \tilde{\mathcal{M}} \setminus \mathcal{A}_j \tag{46}$$

where $\tilde{\mathcal{M}} \subset \mathcal{P}(\Re)^K$ is the set of all bandit models with a unique best VaR arm, and $\mathcal{A}_j \subseteq \tilde{\mathcal{M}}$ is the set of bandit models with $j$ being the arm of lowest VaR.

Recall that for a distribution $\eta$, VaR at quantile $\pi$, denoted as $x_\pi(\eta)$, is defined as

$$x_\pi(\eta) = \inf\{x \in \Re : F_\eta(x) \geq \pi\}.$$

As in the CVaR case, for $\mu \in \tilde{\mathcal{M}}$, (46) can be shown to simplify as

$$V(\mu) = \sup_{t \in \Sigma_K} \min_{b \neq 1} \inf_y \left\{ t_1 \; \mathrm{KL}_{\mathrm{inf}}^{\mathrm{U}}(\mu_1, y) + t_b \; \mathrm{KL}_{\mathrm{inf}}^{\mathrm{L}}(\mu_b, y) \right\},$$

where $\mathrm{KL}_{\mathrm{inf}}^{\mathrm{L}}$ and $\mathrm{KL}_{\mathrm{inf}}^{\mathrm{U}}$ are defined similar to (1) earlier, with the CVaR constraints replaced with the corresponding VaR constraints, and are given as:

$$\mathrm{KL}_{\mathrm{inf}}^{\mathrm{U}}(\eta, y) := \min_{\kappa \in \mathcal{P}(\Re): \; x_\pi(\kappa) \geq y} \mathrm{KL}(\eta, \kappa) \quad \text{and} \quad \mathrm{KL}_{\mathrm{inf}}^{\mathrm{L}}(\eta, y) := \min_{\kappa \in \mathcal{P}(\Re): \; x_\pi(\kappa) \leq y} \mathrm{KL}(\eta, \kappa).$$

$$(47)$$

Let $x_\pi^+(\eta) := \sup \{x \in \Re : F_\eta(x) \leq \pi\}$. Then the set of $\pi^{th}$-quantiles for the distribution $\eta$ is given by $(x_\pi(\eta), x_\pi^+(\eta))$.

**Remark H.1.** We assume that the given bandit problem, $\mu$, has the set of $\pi^{th}$-quantiles disjoint from that of every other arm distribution , as otherwise the bandit instance is not learnable. To see this, fix $\pi = 0.8$ and consider a two-armed bandit problem, $\mu$, where $\mu_1 = \mathrm{Ber}(0.2)$ and $\mu_2 = \mathrm{Ber}(0.2 + \epsilon)$, for an arbitrarity small $\epsilon > 0$. Then $x_\pi(\mu_1) = 0$, $x_\pi^+(\mu_1) = 1$, and $x_\pi(\mu_2) = 1$. Clearly $V(\mu) = 0$, whence, $\mu$ is unlearnable.

Let us now understand some structural properties of the $\mathrm{KL}_{\mathrm{inf}}$ functionals which will be helpful for proving $\delta$-correctness and optimality of the proposed algorithm. For a probability measure $\eta$, let $F_\eta(x) := \eta((-\infty, x])$, denote its CDF evaluated at $x$ and $F_\eta^-(x) = \lim_{y \uparrow x} F_\eta(y)$ denote the left limit of the CDF of $\eta$. Moreover, for $p, q \in (0, 1)$ let $d_2(p, q)$ denote the KL divergence between the Bernoulli random variables with mean $p$ and $q$.

**Lemma H.1** (Restating Lemma 4.4). *For* $y \in \Re$,

$$\mathrm{KL}_{\mathrm{inf}}^{\mathrm{L}}(\eta, y) = d_2(\min \{F_\eta(y), \pi\}, \pi) \quad \text{and} \quad \mathrm{KL}_{\mathrm{inf}}^{\mathrm{U}}(\eta, y) = d_2(\max \{F_\eta^-(y), \pi\}, \pi).$$

*Proof.* Recall that $\mathrm{KL}_{\mathrm{inf}}^{\mathrm{L}}(\eta, y)$ and $\mathrm{KL}_{\mathrm{inf}}^{\mathrm{U}}(\eta, y)$ equal the optimal values of the following problems, respectively:

$$\min \; \mathrm{KL}(\eta, \kappa) \quad \text{s.t.} \quad \kappa \in \mathcal{P}(\Re), \quad F_\kappa(y) \geq \pi, \quad 1 - F_\kappa(y) \leq 1 - \pi, \tag{48}$$

and

$$\min \; \mathrm{KL}(\eta, \kappa) \quad \text{s.t.} \quad \kappa \in \mathcal{P}(\Re), \quad F_\kappa^-(y) \leq \pi, \quad 1 - F_\kappa^-(y) \geq 1 - \pi. \tag{49}$$

Clearly,

$$\mathrm{KL}(\eta, \kappa) = \int_{-\infty}^y \left( \frac{d\eta}{d\kappa}(x) \log \frac{d\eta}{d\kappa}(x) \right) d\kappa(x) + \int_{y^+}^\infty \left( \frac{d\eta}{d\kappa}(x) \log \frac{d\eta}{d\kappa}(x) \right) d\kappa(x),$$

where the first term in the summation above equals

$$F_\eta(y) \int_{-\infty}^y \frac{d\eta/F_\eta(y)}{d\kappa/F_\kappa(y)}(x) \log \left( \frac{d\eta/F_\eta(y)}{d\kappa/F_\kappa(y)}(x) \right) \frac{d\kappa(x)}{F_\kappa(y)} + F_\eta(y) \log \frac{F_\eta(y)}{F_\kappa(y)},$$

which can be lower bounded using Jensen's ineqality by

$$F_\eta(y) \log \frac{F_\eta(y)}{F_\kappa(y)}.$$

Similarly, the second term in the definition of $\mathrm{KL}(\eta, \kappa)$ above can be lower bounded by

$$(1 - F_\eta(y)) \log \frac{1 - F_\eta(y)}{1 - F_\kappa(y)},$$

giving

$$\mathrm{KL}(\eta, \kappa) \geq d_2(F_\eta(y), F_\kappa(y)). \tag{50}$$

Let $\mathrm{Supp}(\eta)$ denote the support of measure $\eta$. Consider $\kappa^*$ defined below.

$$\kappa^*(x) \triangleq \begin{cases} \eta(x)\pi \left(\min\left\{\pi, F_\eta(y)\right\}\right)^{-1}, & \text{for } x \in \{x : x \leq y\}, \\ \eta(x)\left(1 - \pi\right)\left(1 - \min\left\{\pi, F_\eta(y)\right\}\right)^{-1}, & \text{for } x \in \{x : x > y\}, \end{cases}$$

and $\kappa^*(x) \geq 0$ for $x \leq y$ such that $x \notin \text{Supp}(\eta)$, if $\{x : x \leq y\} \cap \text{Supp}(\eta) = \emptyset$.

Clearly, $\kappa^*$ satisfies the constraints of 48 and is feasible to the $\text{KL}_{\text{inf}}^{\text{L}}(\eta, y)$ problem. Moreover,

$$\text{KL}(\eta, \kappa^*) = d_2(\min\left\{\pi, F_\eta(y)\right\}, \pi) \leq d_2(F_\eta(y), F_{\kappa^*}(y)),$$

where the inequality above follows from the monotonicity of $d_2$ in the second argument. This, along with the bound in (50) implies that the above inequality holds as an equality. Whence, $\kappa^*$ is optimal for $\text{KL}_{\text{inf}}^{\text{L}}(\eta, y)$ problem, and we get the desired equality for $\text{KL}_{\text{inf}}^{\text{L}}(\eta, y)$.

The equality for the $\text{KL}_{\text{inf}}^{\text{U}}(\eta, y)$ problem can be argued similarly using $\zeta^*$ defined below:

$$\zeta^*(x) = \begin{cases} \eta(x)\pi \left(\max\left\{\pi, F_\eta^-(y)\right\}\right)^{-1} & \text{for } x \in \{x : x < y\} \\ \eta(x)\left(1 - \pi\right)\left(1 - \max\left\{\pi, F_\eta^-(y_2)\right\}\right)^{-1} & \text{for } x \in \{x : x \geq y\}, \end{cases}$$

and $\zeta^*(x) \geq 0$ for $x \geq y$ and $x \notin \text{Supp}(\eta)$, if $\{x : x \geq y\} \cap \text{Supp}(\eta) = \emptyset$. $\qquad \square$

Thus, $V(\mu)$ in the lower bound equals

$$\sup_{t \in \Sigma_K} \min_{b \neq 1} \inf_{y} \left\{t_1 \, d_2(\max\left\{\pi, F_{\mu_1}^-(y)\right\}, \pi) + t_b \, d_2(\min\left\{\pi, F_{\mu_b}(y)\right\}, \pi)\right\},$$

which can also be shown to equal

$$\sup_{t \in \Sigma_K} \min_{b \neq 1} \inf_{y \in [x_\pi^+(\mu_1), x_\pi(\mu_b)]} \left\{t_1 \, d_2(\max\left\{\pi, F_{\mu_1}^-(y)\right\}, \pi) + t_b \, d_2(\min\left\{\pi, F_{\mu_b}(y)\right\}, \pi)\right\}. \tag{51}$$

**Lemma H.2.** *For fixed $\eta$ and $\pi$, $\text{KL}_{\text{inf}}^{\text{L}}(\eta, y)$ and $\text{KL}_{\text{inf}}^{\text{U}}(\eta, y)$ are lower-semicontinuous in $y$.*

The proof of the above lemma follows from continuity of $d_2(., \pi)$ and dual formulations for $\text{KL}_{\text{inf}}^{\text{L}}$ and $\text{KL}_{\text{inf}}^{\text{U}}$. At the points of discontinuity of $F_\eta$, $\text{KL}_{\text{inf}}^{\text{L}}$ and $\text{KL}_{\text{inf}}^{\text{U}}$ can only decrease in value, whence, lower-semicontinuous.

**Remark H.2.** Let $\eta_n = 0.25\delta_0 + 0.25\delta_1 + 0.25\delta_2 + 0.25\delta_3$, $\pi = 0.6$, $y_n = 1 - \frac{1}{n}$, and $y = 1$. With these, $\eta = \eta_n$, and $y_n \to y$. Using Lemma H.1, $\text{KL}_{\text{inf}}^{\text{L}}(\eta_n, y_n) = d_2(0.25, 0.6)$, while $\text{KL}_{\text{inf}}^{\text{L}}(\eta, y) = d_2(0.5, 0.6)$, thus showing that $\text{KL}_{\text{inf}}^{\text{L}}$ is not jointly continuous. Similar example can be constructed for $\text{KL}_{\text{inf}}^{\text{U}}$.

**Lemma H.3.** *Let $\mathcal{X}$ and $\mathcal{Y}$ be metric spaces with $d_\mathcal{X}$ and $d_\mathcal{Y}$ being the respective metics. Let $\tilde{f} : \mathcal{X} \times \mathcal{Y} \to \Re$ be such that $\forall \tilde{\epsilon} > 0, \exists \tilde{\delta}$ such that $\forall x \in \mathcal{X}$, we have*

$$\forall x' : d_\mathcal{X}(x, x') \leq \delta \implies \sup_{y \in \mathcal{Y}} \left|\tilde{f}(x, y) - \tilde{f}(x', y)\right| \leq \tilde{\epsilon}.$$

*Furthermore, for a fixed $x$, $\tilde{f}(x, y)$ is lower-semicontinuous function of $y$. Then, $\inf_{y \in \mathcal{C}(x)} \tilde{f}(x, y)$ is continuous in $x$, where $\mathcal{C} : \mathcal{X} \to 2^\mathcal{Y}$ is a compact set-valued map.*

*Proof.* Consider a sequence $x_n$ such that $d_K(x_n, x_0) \to 0$ as $n \to \infty$. Let $y_n$ be the minimizer for $\tilde{f}(x_n, y)$, i.e., $\inf_{y \in \mathcal{C}(x_n)} \tilde{f}(x_n, y) = \tilde{f}(x_n, y_n)$ and let $y_0$ be that for $\tilde{f}(x_0, y)$. Existence of $y_n$ and $y_0$ is guaranteed by lower-semicontinuity of $\tilde{f}(x, y)$ for fixed $x$, and compactness of the map $\mathcal{C}$. Then, for a fixed $n$,

$$\tilde{f}(x_0, y_0) - \tilde{f}(x_n, y_n) \leq \tilde{f}(x_0, y_n) - \tilde{f}(x_n, y_n),$$

which, for $n$ large enough so that $d_\mathcal{X}(x_n, x_0) \leq \tilde{\delta}$, is bounded by $\tilde{\epsilon}$. Similarly,

$$\tilde{f}(x_n, y_n) - \tilde{f}(x_0, y_0) \leq \tilde{f}(x_n, y_0) - \tilde{f}(x_0, y_0),$$

which is again bounded by $\tilde{\epsilon}$ for large enough $n$, concluding the proof. $\qquad \square$

For $\mu \in \mathcal{M}$ and $t \in \Sigma_K$, define $g_b(\mu, t, y) := t_1 \, \mathrm{KL}^{\mathrm{U}}_{\mathrm{inf}}(\mu_1, \pi) + t_b \, \mathrm{KL}^{\mathrm{L}}_{\mathrm{inf}}(\mu_b, \pi)$, and let

$$h(\mu, t) \;=\; \min_{b \neq 1} \; \inf_{y \in [x^+_\pi(\mu_1), x_\pi(\mu_b)]} g_b(\mu, t, y).$$

From Lemma H.2, for fixed $\mu, t$, $g_b$ is lower-semicontinuous function of $y$. Furthermore, $d_2(., \pi)$ being a continuous function on bounded support, is a uniformly continuous function. Whence, given $\epsilon > 0$, there exists $\delta > 0$ such that

$$\forall (\tilde{\mu}, \tilde{t}) : \sum_a \left( d_K(\tilde{\mu}_a, \mu_a) + d(\tilde{t}_a, t_a) \right) \leq \delta \implies \sup_y \left| g_b(\mu, t, y) - g_b(\tilde{\mu}, \tilde{t}, y) \right| \leq \epsilon,$$

where $d$ is a metric on $\Re$.

**Corollary H.3.1.** $h(\mu, t)$ *is a jointly continuous function.*

As we did in Section 4 for CVaR, we will use the maximiser, $t^*$, evaluated at the empirical distribution vector to drive our sampling rule. Observe that, unlike CVaR, we do not need to project the empirical distribution in this setting. The algorithm stops at the first time, $n$, when

$$\max_a \; \min_{b \neq a} \; \inf_{x \in [x^+_\pi(\hat{\mu}_a(n)), x_\pi(\hat{\mu}_b(n))]} N_a(n) \, \mathrm{KL}^{\mathrm{U}}_{\mathrm{inf}}(\hat{\mu}_a(n), x) + N_b(n) \, \mathrm{KL}^{\mathrm{L}}_{\mathrm{inf}}(\hat{\mu}_b(n), x) \geq \beta(n, \delta),$$

where

$$\beta(t, \delta) = 6 \log \left( 1 + \log \frac{t}{2} \right) + \log \frac{K - 1}{\delta} + 8 \log \left( 1 + \log \frac{K - 1}{\delta} \right).$$

All in all, the algorithm for finding the best VaR arm is similar to that for CVaR, with the correct definition of $\mathrm{KL}^{\mathrm{L}}_{\mathrm{inf}}$ and $\mathrm{KL}^{\mathrm{U}}_{\mathrm{inf}}$ used at all places.

**Theorem H.4** (Formal statement of Theorem 4.5). *For $\mu \in \tilde{\mathcal{M}}$, the proposed algorithm for finding the best VaR-arm is $\delta$-correct, and satisfies*

$$\limsup_{\delta \to 0} \frac{\mathbb{E}(\tau_\delta)}{\log(1/\delta)} \leq \frac{1}{V(\mu)}. \tag{52}$$

For the proof of the Theorem H.4 above, we need to discuss two things: upper-hemicontinuity of $t^*$, which is needed for the proof of (52) and deviation inequalities for the stopping statistic, which is needed for the $\delta$-correctness.

To prove $\delta$-correctness of the algorithm, we would like to show that

$$\mathbb{P} \left( \exists n \in \mathbb{N} : \max_{a \neq 1} \min_{b \neq a} \left\{ N_a(n) \, \mathrm{KL}^{\mathrm{U}}_{\mathrm{inf}}(\hat{\mu}_a, x_\pi(\mu_a)) + N_b(n) \, \mathrm{KL}^{\mathrm{L}}_{\mathrm{inf}}(\hat{\mu}_1, x_\pi(\mu_1)) \right\} \geq \beta(n, \delta) \right)$$

is at most $\delta$. Towards this, it is sufficient to show that the following event has probability at most $\delta$:

$$\left\{ \exists n \in \mathbb{N} : \max_{a \neq 1} \left\{ N_a(n) d_2(F_{\hat{\mu}_a(n)}, \pi) + N_1(n) d_2(F_{\hat{\mu}_1(n)}, \pi) \right\} \geq \beta(n, \delta) \right\}.$$

The above deviation inequality follows from the observation (see e.g. [32]) that for each arm $a$, $F_{\hat{\mu}_a(t)}(x_\pi(\mu_a))$ is an average of i.i.d. Bernoulli random variables with bias $\pi$. This means that we can employ standard uniform deviation inequalities for sums of self-normalised variables (see, [38, Section 6.1]).

Recall that the sample complexity proof for the best CVaR-arm problem required continuity (upper-hemicontinuity) of $t^*$ only in the Kolmogorov metric which, for measures $\eta_1$ and $\eta_2$ is defined as

$$d_K(\eta_1, \eta_2) = \sup_{x \in \Re} |F_{\eta_1}(x) - F_{\eta_2}(x)|.$$

Upper-hemicontinuity of $t^*(\mu)$ (in the Kolmogorov metric, as a function of $\mu$) follows from Corollary H.3.1 together with Berge's Maximum theorem. Furthermore, the set of maximizers, $t^*$, is convex.

In conclusion, we see that asymptotically optimal algorithms for the minimum VaR arm identification problem lie squarely in the convex hull of existing ideas.

One challenge with the VaR objective is that the objective inside the $\inf_{x_0}$ in the expression for $V(\mu)$ is not (quasi) convex. Our current best computational approach is to do a one-dimensional grid search over candidates $x_0$. Once we have the oracle for computing the inner optimization problem for a given $t \in \Sigma_K$, we can compute the inner $\inf_{x_0}$ problem for the best-looking arm vs all alternatives. We may then further wrap this oracle in an outer optimisation (for example by the Ellipsoid method) to find $t^*$.

# I Discussion on the mean-CVaR problem

Recall that we have $K$ arms, each associated with a distribution, which may, for example, correspond to a loss in a financial investment. When an arm is selected, an independent sample from the associated distribution is revealed. In the mean-CVaR problem, the performance-metric associated with a distribution $\eta$ is $o(\eta) := \alpha_1 m(\eta) + \alpha_2 c_\pi(\eta)$, where $\alpha_1 > 0$, $\alpha_2 > 0$, $m(\eta)$ and $c_\pi(\eta)$ denote the mean and CVaR at $\pi^{th}$-quantile for distribution $\eta$. We are interested in finding the arm with minimum value of this metric in a $\delta$-correct BAI framework. We point out that this problem is conceptually and technically similar to the CVaR-BAI problem, described in detail in the main text. Hence, we state the results directly, while omitting the proofs.

In this section, we again need to restrict arm-distributions to class $\mathcal{L} = \{\eta \in \mathcal{P}(\Re) : \mathbb{E}_\eta(|X|^{1+\epsilon}) \leq B\}$, otherwise the problem is un-learnable. This imposes restrictions on possible values of $o(\eta)$, $c_\pi(\eta)$, and $x_\pi(\eta)$, which are stated next.

**Lemma I.1.** *For $\alpha_1 > 0$, $\alpha_2 > 0$, $\pi \in (0,1)$*

$$\min_{\eta \in \mathcal{L}} \ \alpha_1 m(\eta) + \alpha_2 c_\pi(\eta) + \alpha_2 x_\pi(\eta) \frac{\pi}{1-\pi} \ = \ -B^{\frac{1}{1+\epsilon}} \left( \alpha_1 + \frac{\alpha_2}{1-\pi} \right).$$

*Proof.* First, observe that it suffices to consider distributions supported on only 2-points. Then the following is an equivalent problem:

$$\min_{x \leq y} \ \alpha_1 x \pi + (1-\pi)\alpha_1 y + \alpha_2 y + \alpha_2 x \frac{\pi}{1-\pi} \qquad \text{s.t.} \qquad \pi |x|^{1+\epsilon} + (1-\pi) |y|^{1+\epsilon} \leq B.$$

The objective function can be re-written as $\left( \alpha_1 + \frac{\alpha_2}{1-\pi} \right) (x\pi + (1-\pi)y)$, which clearly is minimized at $x = y = -B^{\frac{1}{1+\epsilon}}$, proving the desired equality. $\qquad\square$

**Lemma I.2.** *For $\alpha_1 > 0$, $\alpha_2 > 0$, $\pi \in (0,1)$*

$$\max_{\eta \in \mathcal{L}} \ \alpha_1 m(\eta) + \alpha_2 c_\pi(\eta) \ = \ B^{\frac{1}{1+\epsilon}} \alpha_1 \left( \pi + (1-\pi) \left( 1 + \frac{\alpha_2}{\alpha_1(1-\pi)} \right)^{1+\frac{1}{\epsilon}} \right)^{\frac{\epsilon}{1+\epsilon}}.$$

*Proof.* First, observe that it is sufficient to consider distributions supported only on 2 points. Thus, the problem is equivalent to

$$\max_{x \leq y} \ \alpha_1 \pi x + \alpha_1 (1-\pi) y + \alpha_2 y \quad \text{s.t.} \quad \pi |x|^{1+\epsilon} + (1-\pi) |y|^{1+\epsilon} \leq B.$$

The objective function above can be re-written as $\alpha_1 \pi x + (1-\pi)y \left( \frac{\alpha_2}{1-\pi} + \alpha_1 \right)$. Since we are optimizing a linear function over a convex set, the optimal will occur at a boundary. In particular, the moment-constraint will hold as equality, and

$$x^* = \left( \frac{B - (1-\pi)y^{1+\epsilon}}{\pi} \right)^{\frac{1}{1+\epsilon}}$$

will be satisfied. At this point, the problem reduces to

$$\max_{\left(\frac{B}{1-\pi}\right)^{\frac{1}{1+\epsilon}} \geq y \geq 0} \ \alpha_1 \pi^{\frac{\epsilon}{1+\epsilon}} \left( B - (1-\pi)y^{1+\epsilon} \right) + (1-\pi)y \left( \frac{\alpha_2}{1-\pi} + \alpha_1 \right).$$

Differentiating with respect to $y$, and setting derivative to 0, we get

$$y^* = \left(1 + \frac{\alpha_2}{\alpha_1(1-\pi)}\right)^{\frac{1}{\epsilon}} \left(\frac{B}{\theta + (1-\theta)\left(1 + \frac{\alpha_1}{\alpha_1(1-\theta)}\right)^{1+\frac{1}{\epsilon}}}\right)^{\frac{1}{1+\epsilon}}.$$

Furthermore, it can be verified that $y^* < \left(\frac{B}{1-\pi}\right)$. Hence, substituting this into the objective function gives the desired result.

$\square$

**Lemma I.3.** *For $\alpha_1 > 0$, $\alpha_2 > 0$, $\pi \in (0,1)$*

$$\min_{\eta \in \mathcal{L}} \ \alpha_1 m(\eta) + \alpha_2 c_\pi(\eta) = -B^{\frac{1}{1+\epsilon}}(\alpha_1 + \alpha_2).$$

*Proof.* First, observe that it is sufficient to consider distributions supported only on 2 points. Thus, the problem is equivalent to

$$\max_{x \leq y} \ \alpha_1 \pi x + \alpha_1(1-\pi)y + \alpha_2 y \quad \text{s.t.} \quad \pi |x|^{1+\epsilon} + (1-\pi)|y|^{1+\epsilon} \leq B.$$

The objective function above can be re-written as $\alpha_1 \pi x + (1-\pi)y\left(\frac{\alpha_2}{1-\pi} + \alpha_1\right)$. Since we are optimizing a linear function over a convex set, the optimal will occur at a boundary. In particular, the moment-constraint will hold as equality, and

$$x^* = -\left(\frac{B - (1-\pi)y^{1+\epsilon}}{\pi}\right)^{\frac{1}{1+\epsilon}}$$

will be satisfied. At this point, the problem reduces to

$$\max_{-B^{\frac{1}{1+\epsilon}} \leq y \leq 0} \ -\alpha_1 \pi^{\frac{\epsilon}{1+\epsilon}} \left(B - (1-\pi)(-y)^{1+\epsilon}\right) + (1-\pi)y\left(\frac{\alpha_2}{1-\pi} + \alpha_1\right).$$

Differentiating with respect to $y$, and setting derivative to 0, we get

$$y^* = -\left(1 + \frac{\alpha_2}{\alpha_1(1-\pi)}\right)^{\frac{1}{\epsilon}} \left(\frac{B}{\theta + (1-\theta)\left(1 + \frac{\alpha_1}{\alpha_1(1-\theta)}\right)^{1+\frac{1}{\epsilon}}}\right)^{\frac{1}{1+\epsilon}}.$$

Observe that $y^* < -B^{\frac{1}{1+\epsilon}}$. Substituting $y^* = 0$ in the objective function, we get the desired bound.

$\square$

**Lemma I.4.** *For $\alpha_1 > 0$, $\alpha_2 > 0$, $\pi \in (0,1)$*

$$\min_{\eta \in \mathcal{L}} \ \alpha_1 m(\eta) + \alpha_2 c_\pi(\eta) - \alpha_2 x_\pi(\eta) = -\alpha_1 B^{\frac{1}{1+\epsilon}}.$$

*Proof.* First, observe that the given problem can be re-written as

$$\min_{\eta \in \mathcal{L}} \ \alpha_1 m(\eta) + \alpha_2 \mathbb{E}_\eta \left((X - x_\pi(\eta))_+\right).$$

Since the second term above is non-negative, and the first term is minimized by $\eta^*$ which is a point mass at $-B^{\frac{1}{1+\epsilon}}$, with second term being 0, $\eta^*$ is optimal, proving the desired equality.

$\square$

As earlier, for $\eta \in \mathcal{P}(\Re)$ and $x \in \Re$, the following two quantities will be crucial for the algorithm and its analysis:

$$\text{KL}_{\text{inf}}^{\text{U}}(\eta, x) = \min_{\substack{\kappa \in \mathcal{L} \\ o(\kappa) \geq x}} \text{KL}(\eta, \kappa) \quad \text{and} \quad \text{KL}_{\text{inf}}^{\text{L}}(\eta, x) = \min_{\substack{\kappa \in \mathcal{L} \\ o(\kappa) \leq x}} \text{KL}(\eta, \kappa).$$

### I.1 Mean-CVaR: algorithm and results

For $x \in \Re$, $z \in \Re$, and $\eta \in \mathcal{P}(\Re)$, let $\mathcal{S}_2(z,x)$ equal

$$\left\{ \lambda_1 \geq 0, \lambda_2 \geq 0 : \min_{y \in \Re} 1 - \lambda_1(B - |y|^{1+\epsilon}) - \lambda_2\left(x - \alpha_1 y - \alpha_2 z - \frac{\alpha_2}{1-\pi}(y-z)_+\right) \geq 0 \right\},$$

$$\mathcal{Z}(x) := \left[ -\left(\frac{B}{\pi}\right)^{\frac{1}{1+\epsilon}}, \frac{x + \alpha_1 B^{\frac{1}{1+\epsilon}}}{\alpha_2} \right],$$

$$O = \left[ -B^{\frac{1}{1+\epsilon}}(\alpha_1 + \alpha_2), B^{\frac{1}{1+\epsilon}}\alpha_1 \left( \pi + (1-\pi)\left(1 + \frac{\alpha_2}{\alpha_1(1-\pi)}\right)^{1+\frac{1}{\epsilon}} \right)^{\frac{\epsilon}{1+\epsilon}} \right],$$

and let $\mathcal{S}_5$ be set of all $(\rho_1, \rho_2, \rho_4)$ such that $\rho_1 \geq 0, \rho_2 \geq 0, \rho_4 \in \Re$, and

$$\min_{y \in \Re} 1 - \rho_1(B - |y|^{1+\epsilon}) + \rho_2(x - \alpha_1 y) - \rho_4(1-\pi) - \left(\frac{\rho_2 \alpha_2 y}{1-\pi} - \rho_4\right)_+ \geq 0.$$

For $\eta \in \mathcal{P}(\Re)$ and $x \in O^o$, $\mathrm{KL}_{\mathrm{inf}}^{\mathrm{L}}(\eta, x)$ equals

$$\min_{z \in \mathcal{Z}(x)} \max_{(\lambda_1, \lambda_2) \in \mathcal{S}_2} \mathbb{E}_\eta \left( \log\left( 1 - \lambda_1(B - |X|^{1+\epsilon}) - \lambda_2\left(x - \alpha_1 X - \alpha_2 z - \frac{\alpha_2}{1-\pi}(X-z)_+\right) \right) \right),$$

and $\mathrm{KL}_{\mathrm{inf}}^{\mathrm{U}}(\eta, x)$ equals

$$\max_{\rho \in \mathcal{S}_5} \mathbb{E}_\eta \left( \log\left( 1 - \rho_1(B - |X|^{1+\epsilon}) + \rho_2(x - \alpha_1 X) - \rho_4(1-\pi) - \left(\frac{\rho_2 \alpha_2 X}{1-\pi} - \rho_4\right)_+ \right) \right).$$

As earlier, these are precisely the dual representations for the KL-projection functionals, crucial for the algorithm, and its analysis. Compactness of the dual regions, $\mathcal{S}_2$ and $\mathcal{S}_5$, can be argued as in Section D.3. Joint-continuity of these KL-projection functionals can also be established by mimicking the arguments in Section C.2, which is required for the sample complexity proof. Concentration inequalities similar to those in Proposition 4.2 can be developed for the empirical versions of these mean-CVaR KL-projection functionals, and the algorithm of Section 4, with $\mathrm{KL}_{\mathrm{inf}}^{\mathrm{U}}$ and $\mathrm{KL}_{\mathrm{inf}}^{\mathrm{L}}$ as defined in this section, and $\beta$ as in (7), gives a plug-n-play algorithm for the mean-CVaR BAI problem.

**Theorem I.5** (Formal statement of Theorem 4.3). *For $\mu \in \mathcal{M}^o$, the proposed algorithm for finding the best mean-CVaR-arm is $\delta$-correct, and satisfies*

$$\limsup_{\delta \to 0} \frac{\mathbb{E}(\tau_\delta)}{\log(1/\delta)} \leq \frac{1}{V(\mu)}.$$

## J  Discussion of $\mathrm{KL}_{\mathrm{inf}}$-based confidence intervals for CVaR

In this section we construct $\mathrm{KL}_{\mathrm{inf}}$-based confidence intervals for CVaR, and compare them to those obtained from the traditional concentration and clipping arguments. We will show how the traditional argument can be recovered from the $\mathrm{KL}_{\mathrm{inf}}$ concentration at minor overhead, but with built-in anytime validity.

Given $n$ samples from a distribution $\eta \in \mathcal{L}$, $U_n$ defined below is an upper bound on the true CVaR of $\eta$ at its $\pi^{th}$ quantile, where

$$U_n = \max\left\{ x \in \Re : n\,\mathrm{KL}_{\mathrm{inf}}^{\mathrm{U}}(\hat{\eta}_n, x) \leq C \right\}, \tag{53}$$

for an appropriately chosen threshold $C$, so that $U_n \geq c_\pi(\eta)$ with probability at least $1 - \delta$. This follows from Proposition 4.2.

This upper bound can be re-formulated as

$$U_n = \max \left\{ c_\pi(\kappa) : \ \kappa \in \mathcal{L}, \ n \, \mathrm{KL}(\hat{\eta}_n, \kappa) \leq C \right\}.$$

Using the Donsker-Varadhan variational representation for the KL divergence, $U_n$ is at most

$$\max_{\kappa \in \mathcal{L}} \ c_\pi(\kappa) \quad \text{s.t.} \quad n \mathbb{E}_{\hat{\eta}(n)} \left( g(X) \right) - n \log \mathbb{E}_\kappa \left( e^{g(X)} \right) \leq C,$$

for all measurable functions, $g$, with a finite second term above. Let $x_\pi$ denote the $\pi^{th}$-quantile for $\kappa$. Then, for a sequence of thresholds $u_n$, and $\theta > 0$, we define the function

$$g_n(X) = -\frac{\theta}{1 - \pi} X \mathbb{1} \left( x_\pi \leq X \leq u_n \right).$$

Substituting $g_n$ for $g$ in the above, and adding $\frac{n\theta}{1-\pi} \mathbb{E}_\kappa \left( X \mathbb{1} \left( x_\pi \leq X \right) \right)$ on both the sides, we get that $U_n$ is at most the maximum $c_\pi(\kappa)$ such that $\kappa \in \mathcal{L}$ and

$$\frac{\theta}{1 - \pi} \sum_{i=1}^n \left( \mathbb{E}_\kappa \left( X \mathbb{1} \left( x_\pi \leq X \right) \right) - X_i \mathbb{1} \left( x_\pi \leq X_i \leq u_n \right) \right)$$

$$- n \log \mathbb{E}_\kappa \left( e^{-\frac{\theta}{1-\pi} X \mathbb{1}(x_\pi \leq X \leq u_n)} \right) \leq C + \frac{n\theta}{1 - \pi} \mathbb{E}_\kappa \left( X \mathbb{1} \left( x_\pi \leq X \right) \right). \qquad (54)$$

Let $Y_n = X \mathbb{1} \left( x_\pi \leq X \leq u_n \right)$, $m_n = \mathbb{E}_\kappa \left( X \mathbb{1} \left( x_\pi \leq X \leq u_n \right) \right)$, and $\theta_\pi = \frac{\theta}{1-\pi}$. Then $|Y_n| \leq u_n$ and $\mathbb{E}_\kappa \left( \theta_\pi^2 Y_n^2 \right) \leq \theta_\pi^2 B u_n^{1-\epsilon}$. Using this,

$$\mathbb{E}_\kappa \left( e^{-\theta_\pi X \mathbb{1}(x_\pi \leq X \leq u_n)} \right) \leq 1 - \theta_\pi m_n + \sum_{j=2}^\infty \frac{\mathbb{E}_\kappa \left( |\theta_\pi Y_n|^j \right)}{j!} \leq 1 - \theta_\pi m_i + \frac{B}{u_n^{1+\epsilon}} \sum_{j=2}^\infty \frac{(\theta_\pi u_n)^j}{j!}. \qquad (55)$$

Thus, we have $\mathbb{E}_\kappa \left( e^{-\theta_\pi X \mathbb{1}(x_\pi \leq X \leq u_n)} \right) \leq 1 - \theta_\pi m_n + \frac{B}{u_n^{1+\epsilon}} \left( e^{\theta_\pi u_n} - \theta u_n - 1 \right)$. Using $1 + x \leq e^x$ and (55) in (54), we get that $U_n$ is at most: $\max_{\kappa \in \mathcal{L}} \ c_\pi(\kappa)$ subject to

$$\theta_\pi \sum_{i=1}^n \left( \mathbb{E}_\kappa \left( X \mathbb{1} \left( x_\pi \leq X \right) \right) - X_i \mathbb{1} \left( x_\pi \leq X_i \leq u_n \right) \right)$$

$$\leq C + n \left( \theta_\pi \mathbb{E}_\kappa \left( X \mathbb{1} \left( x_\pi \leq X \right) \right) - \theta_\pi m_n + \frac{B}{u_n^{1+\epsilon}} \left( e^{\theta_\pi u_n} - \theta u_n - 1 \right) \right).$$

Clearly, $\mathbb{E}_\kappa \left( X \mathbb{1} \left( X \geq u_n \right) \right)$ is at most $B/(u_n)^\epsilon$. The above constraint can be relaxed to

$$\frac{1}{n} \sum_{i=1}^n \left( \mathbb{E}_\kappa \left( X \mathbb{1} \left( x_\pi \leq X \right) \right) - X_i \mathbb{1} \left( x_\pi \leq X_i \leq u_n \right) \right)$$

$$\leq \frac{B}{u_n^\epsilon} + \frac{1}{\theta_\pi} \left( \frac{C}{n} + \frac{B}{u_n^{1+\epsilon}} \left( e^{\theta_\pi u_n} - \theta_\pi u_n - 1 \right) \right).$$

Choosing $\theta_\pi = \frac{C u_n^\epsilon}{nB}$, the above constraint is

$$\frac{1}{n} \sum_{i=1}^n \left( \mathbb{E}_\kappa \left( X \mathbb{1} \left( x_\pi \leq X \right) \right) - X_i \mathbb{1} \left( x_\pi \leq X_i \leq u_n \right) \right) \leq \frac{B}{u_n^\epsilon} + \frac{nB^2}{C u^{1+2\epsilon}} \left( e^{\frac{C u^{1+\epsilon}}{Bn}} - 1 \right).$$

Recall that

$$c_\pi(\kappa) = \frac{1}{1 - \pi} \mathbb{E}_\kappa \left( X \mathbb{1} \left( x_\pi \leq X \right) \right).$$

Setting

$$u_n = \left( Bn \left( \log \delta_0^{\text{-}1} \right)^{\text{-}1} \right)^{\frac{1}{1+\epsilon}} \quad \text{and} \quad \hat{c}_{\pi,n}(\delta_0) := \frac{1}{n(1-\pi)} \sum_{i=1}^{n} X_i \mathbb{1}\left( x_\pi \le X_i \le u_n \right),$$

for some parameter $\delta_0$ which will be chosen later, we get the following upper bound on $U_n$:

$$\hat{c}_{\pi,n}(\delta_0) + \frac{B^{\frac{1}{1+\epsilon}}}{1-\pi} \left( \frac{\log \delta_0^{\text{-}1}}{n} \right)^{\frac{\epsilon}{1+\epsilon}} \left( 1 + \left( e^{\frac{C}{\log \delta_0^{\text{-}1}}} - 1 \right) \frac{\log \delta_0^{\text{-}1}}{C} \right). \tag{56}$$

Observe that $\hat{c}_{\pi,n}(\delta)$ is the popular truncation-based estimator. Now, if $\delta_0 = \delta$ and $C \approx \log \delta^{\text{-}1}$, then we obtain

$$U_n \le \hat{c}_{\pi,n}(\delta) + \frac{4B^{\frac{1}{1+\epsilon}}}{1-\pi} \left( \frac{\log \delta^{\text{-}1}}{n} \right)^{\frac{\epsilon}{1+\epsilon}}, \tag{57}$$

which is a $(1-\delta)$-probability upper bound for the true CVaR, obtained using the truncated estimator, $\hat{c}_{\pi,n}(\delta)$ [39, see, (29)], assuming perfect estimation of VaR at the $\pi^{th}$ quantile. From Proposition 4.2, however, the current best $C$ permitted is $\log \delta^{\text{-}1}$ with an additional 3 log(number of samples), which gives that our upper-bound will be worse. However, our confidence intervals are any-time (as Proposition 4.2 is).

Typically, in applications to multi-armed bandit problems, we require high-probability upper bounds for a *random* number of samples allocated to an arm. Our confidence intervals are any-time and can be directly used in these applications. The same is not true for the truncation-based intervals, where a union bound would instead be needed in the analysis.

For example, in the classical regret-minimization framework of MAB with CVaR as the (unobserved) loss, a UCB algorithm based on the truncation-based estimator in (57) would choose $\delta = T^{-2}$ at time $T$, for constructing index for an arm with $n$ samples. With this choice of $\delta$, r.h.s. of (57) would correspond to index for such an arm at time $T$. On the other hand, since the bound for $\text{KL}_{\text{inf}}^{\text{U}}$ is anytime, UCB constructed using it would require $\delta \approx (T \log^2(T))^{\text{-}1}$. This would correspond to setting $C \approx \log T + 2 \log \log T + 3 \log$ (number of samples) in (53).

[3] recently show that a UCB algorithm using an arm index similar to (53) ($\text{KL}_{\text{inf}}$-UCB) is asymptotically optimal for the mean objective in heavy-tailed bandits. Their algorithm, with $\text{KL}_{\text{inf}}$ replaced with $\text{KL}_{\text{inf}}^{\text{U}}$ or $\text{KL}_{\text{inf}}^{\text{L}}$ as appropriate, will be an optimal algorithm for regret-minimization with the CVaR-objective. Since for such an algorithm, sub-optimal arms are pulled approximately $\log T$ times till time $T$, this would correspond to setting

$$C \approx \log T + 2 \log \log T + 3 \log \log T$$

for sub-optimal arms in this application. With this choice of C, (56) is an upper bound on our index at time $T$, for all values of $\delta_0$. In particular, setting $\delta_0 = T^{-2}$, we get that our index for a sub-optimal arm is dominated by

$$\hat{c}_{\pi,n}(T^{-2}) + \frac{3B^{\frac{1}{1+\epsilon}}}{1-\pi} \left( \frac{2 \log T}{n} \right)^{\frac{\epsilon}{1+\epsilon}},$$

which is smaller than the index of UCB with the truncated estimator.

Furthermore, the comparison with (56) doesn't account for error in estimating the VaR of the underlying distribution, which is also needed in the truncation-based CVaR estimator. We would also like to point out that the estimator of [39] is similar to the $\hat{c}_{\pi,n}(\delta)$ defined above, with the truncation level changing for each sample. However, using their analysis, it can be shown that both $\hat{c}_{\pi,n}(\delta)$ have the same guarantees.

## K   Batched algorithm and Sample complexity

In this section, look at the computational complexity of our asymptotically-optimal algorithm for the CVaR or mean-CVaR BAI, and propose a modification which is optimal up to constants, but is computationally less expensive.

We observe numerically that the computational cost of the KL-projection functionals increases linearly in the number of samples of the empirical distribution. In particular, the computation of

optimal weights increases linearly with number of samples. Let this cost at time $n$ be $c_1 + c_2 n$, where $c_1$ and $c_2$ are non-negative constants. Then, the over-all cost of the algorithm till time $\tau_\delta$ is $\left(c_1 + \frac{c_2}{2}\right)\tau_\delta + \frac{c_2}{2}\tau_\delta^2$, which is quadratic in the total number of samples.

Consider a modification in which we only check for stopping condition, and compute the weights at $(1 + \eta)$-geometrically spaced times, for $\eta > 0$, and use these weights to allocate the samples at all the intermediate times using any reasonable tracking rule. The $(1 + \eta)$-batched algorithm with the randomized tracking rule of [2] is formally described in the next subsection. The algorithm makes an error if at the stopping time, its estimate for the best-arm is incorrect. As earlier, the error probability can be seen to be at most 36, and $\delta$-correctness thus follows (Section K.2).

**Theorem K.1.** *The $\eta$-batched algorithm with randomized tracking is $\delta$-correct, and satisfies:*

$$\limsup_{\delta \to 0} \frac{\mathbb{E}(\tau_\delta)}{\log(1/\delta)} \leq \frac{1 + \eta}{V(\mu)}.$$

The proof of the Theorem K.1 is similar to that of sample complexity part of Theorem 4.1 and can be found in Appendix K.3 below. Thus, if $\tau_\delta$ denotes the stopping time of the original algorithm, the batched algorithm stops after at most $(1 + \eta)\tau_\delta$. Moreover, it computes optimal weights roughly at times $(1 + \eta)^i$, for $i \in \left\{0, 1, \ldots, \frac{\log \tau_\delta}{1+\eta} + 1\right\}$. Thus, the computational cost of this algorithm is at most $\left(\frac{\log \tau_\delta}{\log(1+\eta)} + 1\right)c_1 + (1 + \eta)^2 \tau_\delta \frac{c_2}{\eta}$, which is roughly linear in $\tau_\delta$, the number of samples generated.

## K.1 Algorithm

The algorithm proceeds in batches, as below. Let $t_l$ denote the time of beginning of $l^{th}$ batch, and let $b_l$ denote its size. We use the randomized sampling rule, as in [2]. The stopping and recommendation rules are as earlier (see Section 4).

- Pull each arm $K$ times. Initialize $l = 1$, $t_l = K^2 + 1$ and $b_l = \max\left\{1, \left\lceil \tilde{\eta}(t_l - K^2)\right\rceil\right\}$.
- At the beginning of each batch, $l$, compute the optimal weights $t^*(\Pi(\hat{\mu}(t_l)))$, where $\Pi$ is the map that projects the argument on to $\mathcal{L}^K$ in the Kolmogorov metric (see Section 4 for details of the projection map). Check if the stopping condition is met. If not,

  1. Compute starvation of each arm $a$, defined as $s_a = \max\left\{0, (t_l + b_l)^{1/2} - N_a(t_l)\right\}$.
  2. If $\sum_a s_a \leq b_l$, generate $s_a$ many samples from arm $a$, for all arms. Furthermore, generate $b_l - \sum_a s_a$ i.i.d. samples from the weight distribution, $t^*(\Pi(\hat{\mu}(t_l)))$. For each arm $a$, count the number of times $a$ appears in the generated samples and sample arm a that many times.
  3. Else if $b_l < \sum_a s_a$, then generate $\hat{s}_a$ samples from arm $a$, where $\hat{s} = \{\hat{s}_1, \ldots \hat{s}_K\}$ is the solution to the following load balancing problem: $\min_{\hat{s}} \max_a \{s_a - \hat{s}_a\}$ such that $\hat{s}_a \in \mathbb{N}$, $\hat{s}_a \in [0, s_a]$, and $\sum_a \hat{s}_a = b_l$.
  4. Set $t_{l+1} = t_l + b_l$, $b_{l+1} = \max\left\{1, \left\lceil \tilde{\eta}(t_{l+1} - K^2)\right\rceil\right\}$, and $l = l + 1$, and repeat.

- If the stopping condition is met, declare the empirically-best CVaR-arm as the answer.

Clearly, $K^2 + \frac{(1+\tilde{\eta})^l - 1}{\tilde{\eta}} \geq t_l \geq K^2 + (1 + \tilde{\eta})^{l-1}$ and $(1 + \tilde{\eta})^l \geq b_l \geq \tilde{\eta}(1 + \tilde{\eta})^{l-1}$, and are deterministic. Moreover, for $t > 0$, let $l(t)$ denote the batch $l$ such that $t_l \leq t \leq t_{l+1}$.

**Lemma K.2.** *The $\tilde{\eta}$-batched algorithm ensures that for all $l \geq 1$, $N_a(t_l) \geq \frac{t_l^{\frac{1}{2}}}{K} - 1$.*

*Proof.* Clearly, for $l = 1$, $t_1 = K^2 + 1$, and each arm has $K \geq \frac{t_l^{\frac{1}{2}}}{K} - 1$ samples. Let the given statement be true for all $l \leq l_0$, for some $l_0 \in \mathbb{N}$. Then, for $l = l_0 + 1$ the statement will be true if $(t_{l_0+1}^{\frac{1}{2}} - t_{l_0}^{\frac{1}{2}}) \leq \left\lceil \tilde{\eta}(t_{l_0} - K^2)\right\rceil$, where r.h.s. is the number of samples available with the algorithm in the batch $l_0 + 1$, and l.h.s. is the maximum number of samples the algorithm will need to allocate in order to ensure the inequality in the lemma. The above is equivalent to showing that $\left(t_{l_0} + \left\lceil \tilde{\eta}(t_{l_0} - K^2)\right\rceil\right)^{\frac{1}{2}} - t_{l_0}^{\frac{1}{2}} < \left\lceil \tilde{\eta}t_{l_0} - \tilde{\eta}K^2\right\rceil$. For positive $a$ and $b$, $a^{\frac{1}{2}} + b^{\frac{1}{2}} \geq (a + b)^{\frac{1}{2}}$.

Hence, $(t_{l_0} + \lceil \tilde{\eta}(t_{l_0} - K^2) \rceil)^{\frac{1}{2}} - t_{l_0}^{\frac{1}{2}} \leq \lceil \tilde{\eta} t_{l_0} - \tilde{\eta} K^2 \rceil^{\frac{1}{2}} \leq \lceil \tilde{\eta} t_{l_0} - \tilde{\eta} K^2 \rceil$, proving the desired inequality. $\qquad\square$

## K.2   $\delta$-correctness

As in Section 4, our stopping rule corresponds to thresholding the $Z$ statistic (see (7)). However, instead of checking this at each time, we do this only at the beginning of each batch. Formally, the stopping time, $\tau_\delta$, lies in $\{t_l : l \in \mathbb{N}\}$, where $t_l$ corresponds to time of beginning of $l^{th}$ batch. As earlier, error occurs when at time $\tau_\delta$, the estimated best-arm is not arm 1. Thus, the error event is contained in

$$\left\{ \exists n : \bigcup_{i \neq 1} \left\{ \inf_{x \leq y} \left\{ N_i(n) \, \mathrm{KL}_{\mathrm{inf}}^{\mathrm{U}}(\hat{\mu}_i(n), y) + N_1(n) \, \mathrm{KL}_{\mathrm{inf}}^{\mathrm{L}}(\hat{\mu}_1(n), x) \right\} \geq \beta; \, \mathcal{E}_n(i) \right\} \right\},$$

which can be bounded using Proposition 4.2, as in Section 4. We omit the details here, and refer the reader to Section 4.

## K.3   Sample complexity

We now prove that the sample complexity of the batched-algorithm matches the lower bound upto a factor of $1 + \tilde{\eta}$, asymptotically as $\delta \to 0$, i.e., it satisfies

$$\limsup_{\delta \to 0} \frac{\mathbb{E}_\mu(\tau_\delta)}{\log \frac{1}{\delta}} \leq \frac{1 + \tilde{\eta}}{V(\mu)}.$$

As in Section F.2, we use the projections in the Kolmorogov metric, i.e., $\Pi = (\Pi_1, \Pi_2, \ldots, \Pi_K)$, where

$$\Pi_i(\eta) \in \operatorname*{argmin}_{\kappa \in \mathcal{L}} d_K(\kappa, \eta), \quad \text{and} \quad d_K(\kappa, \eta) = \sup_{x \in \Re} |F_\kappa(x) - F_\eta(x)|,$$

and $F_\kappa$ and $F_\eta$ denote the CDF functions for the measures $\eta$ and $\kappa$. As earlier (Section F.2), define $\mathcal{I}_{\epsilon'} \triangleq B_\zeta(\mu_1) \times B_\zeta(\mu_2) \times \ldots \times B_\zeta(\mu_K)$, where $B_\zeta(\mu_i) = \{\kappa \in \mathcal{P}(\Re) : d_K(\kappa, \mu_i) \leq \zeta\}$, and $\zeta > 0$ is chosen to satisfy the following:

$$\mu' \in \mathcal{I}_{\epsilon'} \implies \forall t' \in t^*\left(\Pi(\mu')\right), \, \exists t \in t^*(\mu) \text{ s.t. } \|t' - t\|_\infty \leq \epsilon'.$$

Recall that $\mu \in \mathcal{M}$ is such that $-f^{-1}(B) < c_\pi(\mu_1) < \max_{j \neq 1} c_\pi(\mu_j) < f^{-1}\left(\frac{B}{1-\pi}\right)$, where $f^{-1}(c) := \max\{y : f(y) = c\}$. For $T \in \mathbb{N}$, define $l_0(T) = l(T^{\frac{1}{4}})$, $l_1(T) = l(T^{\frac{3}{4}}) + 1$, $l_2(T) = \max\{l_1(T), l(T) - 1\}$, where for $n \in \mathbb{N}$, $l(n)$ denotes $l$ such that $t_l \leq t \leq t_{l+1}$ and $t_l$ denotes time of beginning of $l^{th}$ batch. Furthermore, let

$$\mathcal{G}_T(\epsilon') = \bigcap_{l=l_0(T)}^{l_2(T)} \{\hat{\mu}(t_l) \in \mathcal{I}_{\epsilon'}\} \bigcap_{l=l_1(T)}^{l_2(T)} \left\{ \max_{a \in [K]} \left| \frac{N_a(t_l)}{t_l} - t_a^*(\mu) \right| \leq 4\epsilon' \right\}.$$

Let $\mu'$ be a vector of $K$, 1-dimensional distributions from $\mathcal{P}(\Re)$, $[K] = \{1, \ldots, K\}$, and let $t' \in \Sigma_K$. Define

$$g(\mu', t') \triangleq \max_{a \in [K]} \min_{b \neq a} \inf_{x \in \left[-f^{-1}(B), f^{-1}\left(\frac{B}{1-\pi}\right)\right]} \left( t_a' \, \mathrm{KL}_{\mathrm{inf}}^{\mathrm{U}}(\mu_1', x) + t_b' \, \mathrm{KL}_{\mathrm{inf}}^{\mathrm{L}}(\mu_b', x) \right).$$

Note that, for $\mu \in (\mathcal{P}(\Re))^K$, from Lemma C.3 and Berge's Theorem (see, [8, Theorem 2, Page 116]), $g(\mu, t)$ is a jointly lower-semicontinuous function of $(\mu, t)$. Let $\|.\|_\infty$ be the maximum norm in $\Re^K$, and

$$C_{\epsilon'}^*(\mu) \triangleq \inf_{\substack{\mu' \in \mathcal{I}_{\epsilon'} \\ t': \inf_{t \in t^*(\mu)} \|t' - t\|_\infty \leq 4\epsilon'}} g(\mu', t').$$

Recall that for $n \in \mathbb{N}$, the modified log generalized likelihood ratio statistic for $\hat{\mu}(n)$, used in the stopping rule, is given by $Z(n) = \max_a \min_{b \neq a} Z_{a,b}(n)$, where

$$Z_{a,b}(n) = n \inf_{x \in \left[-f^{-1}(B), f^{-1}\left(\frac{B}{1-\pi}\right)\right]} \left( \frac{N_a(n)}{n} \mathrm{KL}_{\inf}^{\mathrm{U}}(\hat{\mu}_a(n), x) + \frac{N_b(n)}{n} \mathrm{KL}_{\inf}^{\mathrm{L}}(\hat{\mu}_b(n), x) \right).$$

On $\mathcal{G}_T(\epsilon')$, for $T \geq K + 1$ and $l \in \mathbb{N}$ such that $l_2(T) \geq l \geq l_1(T)$,

$$
\begin{aligned}
Z(t_l) &= t_l \max_a \min_{b \neq a} \inf_{x \in \left[-f^{-1}(B), f^{-1}\left(\frac{B}{1-\pi}\right)\right]} \left( \frac{N_a(t_l)}{t_l} \mathrm{KL}_{\inf}^{\mathrm{U}}(\hat{\mu}_a(t_l), x) + \frac{N_b(t_l)}{t_l} \mathrm{KL}_{\inf}^{\mathrm{L}}(\hat{\mu}_b(n), x) \right) \\
&= t_l \, g\left( \hat{\mu}(t_l), \left\{ \frac{N_1(t_l)}{t_l}, \ldots, \frac{N_K(t_l)}{t_l} \right\} \right) \\
&\geq t_l \, C_{\epsilon'}^*(\mu).
\end{aligned}
\tag{58}
$$

Furthermore, for $T \geq K^2 + 1$, on $\mathcal{G}_T(\epsilon')$,

$$
\begin{aligned}
\min\{\tau_\delta, T\} &\leq t_{l_1(T)} + \sum_{l=l_1(T)+1}^{l_2(T)} b_l \mathbf{1}\left(t_l < \tau_\delta\right) \\
&= t_{l_1(T)} + \sum_{l=l_1(T)+1}^{l_2(T)} b_l \mathbf{1}\left(Z(t_l) < \beta(t_l, \delta)\right) \\
&\leq t_{l_1(T)} + \sum_{l=l_1(T)+1}^{l_2(T)} b_l \mathbf{1}\left(t_l < \frac{\beta(t_l, \delta)}{C_{\epsilon'}^*(\mu)}\right) \\
&\leq t_{l_1(T)} + \frac{\beta(t_{l_2(T)}, \delta)}{C_{\epsilon'}^*(\mu)} + b_{l_2(T)} \\
&\leq t_{l_1(T)} + (1+\tilde{\eta})\frac{\beta(T, \delta)}{C_{\epsilon'}^*(\mu)} + 1,
\end{aligned}
\tag{59}
$$

where for the last inequality, we use monotonicity of $\beta(\cdot, \cdot)$ in the first argument, and that $b_{l_2(T)} \leq \tilde{\eta}\frac{\beta(T,\delta)}{C_{\epsilon'}^*(\mu)} + 1$. Next, define

$$T_0(\delta) = \inf\left\{ n \in \mathbb{N} : t_{l_1(n)} + (1+\tilde{\eta})\frac{\beta(n, \delta)}{C_{\epsilon'}^*(\mu)} + 1 \leq n \right\}.$$

On $\mathcal{G}_T$, for $T \geq \max\left\{T_0(\delta), K^2 + 1\right\}$, from (59) and definition of $T_0(\delta)$,

$$\min\{\tau_\delta, T\} \leq t_{l_1(T)} + (1+\tilde{\eta})\frac{\beta(T, \delta)}{C_{\epsilon'}^*(\mu)} \leq T,$$

which gives that for such a $T$, $\tau_\delta \leq T$. Thus, for $T \geq \max\left\{T_0(\delta), K^2 + 1\right\}$, we have $\mathcal{G}_T(\epsilon') \subset \{\tau_\delta \leq T\}$ and hence, $\mathbb{P}_\mu(\tau_\delta > T) \leq \mathbb{P}_\mu(\mathcal{G}_T^c)$. Moreover, for a constant $T_{\epsilon'}$, Lemma F.5 bounds the probability of $\mathcal{G}_T^c$ for $T \geq T_{\epsilon'}$. Since $\tau_\delta \geq 0$,

$$\mathbb{E}_\mu(\tau_\delta) \leq T_0(\delta) + K^2 + 1 + T_{\epsilon'} + \sum_{T=T_0(\delta)+K^2+1+T_{\epsilon'}}^{\infty} \mathbb{P}_\mu(\mathcal{G}_T^c(\epsilon')). \tag{60}$$

For $\tilde{e} > 0$, it can be shown that

$$\limsup_{\delta \longrightarrow 0} \frac{T_0(\delta)}{\log(1/\delta)} \leq \frac{(1+\tilde{\eta})(1+\tilde{e})}{C_{\epsilon'}^*(\mu)}. \tag{61}$$

Then, from (60), (61), and Lemma K.3,

$$\limsup_{\delta \to 0} \frac{\mathbb{E}_\mu(\tau_\delta)}{\log(1/\delta)} \leq \frac{(1+\tilde{\eta})(1+\tilde{e})}{C_{\epsilon'}^*(\mu)}.$$

From lower-semicontinuity of $g(\mu', t')$ in $(\mu', t')$ for $\mu' \in (\mathcal{P}(\Re))^K$, it follows that $\liminf_{n \to \infty} C^*_{\epsilon'}(\mu) \geq V(\mu)$. First letting $\tilde{e} \to 0$ and then letting $\epsilon' \to 0$, we get the desired inequality.

**Lemma K.3.** *Let $T_{\epsilon'} = \epsilon'^{-8/3}$. Then,* $\limsup_{\delta \to 0} \dfrac{\sum_{T=T_{\epsilon'}}^{\infty} \mathbb{P}_\mu(\mathcal{G}^c_T(\epsilon'))}{\log(1/\delta)} = 0.$

*Proof.* The proof of this is similar to that in [2, Lemma 32]. However, the batch sizes in our algorithm may not be constant. We modify the proof to allow for this flexibility.

Recall that for $T \in \mathbb{N}$ and $T > K^2$, $l_0(T) = l(T^{\frac{1}{4}})$, $l_1(T) = l(T^{\frac{3}{4}}) + 1$, $l_2(T)$ equal $\max\{l_1(T), l(T) - 1)\}$, for $l \in \mathbb{N}$, $t_l$ denotes the beginning of $l^{th}$ batch, and

$$\mathcal{G}_T(\epsilon') = \bigcap_{l=l_0(T)}^{l_2(T)} \{\hat{\mu}(t_l) \in \mathcal{I}_{\epsilon'}\} \bigcap_{l=l_1(T)}^{l_2(T)} \left\{ \max_{a \in [K]} \left| \frac{N_i(t_l)}{t_l} - t^*_a(\mu) \right| \leq 4\epsilon' \right\}.$$

Let

$$\mathcal{G}^1_T(\epsilon') \triangleq \bigcap_{l=l_0(T)}^{l_2(T)} \{\hat{\mu}(t_l) \in \mathcal{I}_{\epsilon'}\}.$$

Using union bounds,

$$\mathbb{P}_\mu(\mathcal{G}^c_T(\epsilon')) \leq \sum_{l=l_0(T)}^{l_2(T)} \mathbb{P}_\mu\left(\hat{\mu}(t_l) \notin \mathcal{I}_{\epsilon'}\right) + \sum_{l=l_1(T)}^{l_2(T)} \sum_{i=1}^{K} \mathbb{P}\left( \left| \frac{N_a(t_l)}{t_l} - t^*_i(\mu) \right| \geq 4\epsilon', \mathcal{G}^1_T(\epsilon') \right). \quad (62)$$

The first term above can be bounded by

$$\sum_{l=l_0(T)}^{l_2(T)} \sum_{a=1}^{K} \mathbb{P}\left( \sup_x \left| F_{\hat{\mu}_a(t_l)}(x) - F_a(x) \right| \geq \epsilon' \right).$$

From Lemma K.2, the algorithm ensures at least $\frac{\sqrt{t_l}}{K} - 1 \geq \sqrt{t_l}/(2K)$ samples to each arm till time $t_l$. Using this, each summand in the bound above can be bounded as follows:

$$\mathbb{P}\left( \sup_x \left| F_{\hat{\mu}_a(t_l)}(x) - F_a(x) \right| \geq \epsilon' \right) \leq \mathbb{P}\left( \sup_x \left| F_{\hat{\mu}_a(t_l)}(x) - F_a(x) \right| \geq \epsilon'; N_a(t_l) \geq \frac{\sqrt{t_l}}{2K} \right).$$

R.h.s. in the above inequality can be bounded using union bound and DKW inequality by

$$\sum_{j=\sqrt{t_l}/(2K)}^{t_l} e^{-2j\epsilon'^2} \leq e^{-\epsilon'^2 \frac{\sqrt{t_l}}{K}} \left(1 - e^{-2\epsilon'^2}\right)^{-1}.$$

Thus, the first term in (62) is bounded by $KTe^{-\epsilon'^2 \frac{T^{1/8}}{K}} \left(1 - e^{-2\epsilon'^2}\right)^{-1}$.

To bound the other term in (62), for $l \in \{l_1(T), \ldots, l_2(T)\}$, let $M_{t_l}$ denote the set of times in $\{1, \ldots, t_l\}$ when the algorithm flipped coins to decide which arm to pull. Define

$$A_2 \triangleq \frac{1}{t_l} \sum_{j \in M_{t_l}} |t^*_i(\Pi(\hat{\mu}(j))) - t^*_i(\mu)|, \quad \text{and} \quad A_3 \triangleq \frac{1}{t_l} \sum_{j \notin M_{t_l}} |I_i(j) - t^*_i(\mu)|,$$

where $I_i(j)$ is the indicator that $i^{th}$ arm is pulled on $j^{th}$ time step, and $\hat{\mu}(j)$ denotes the empirical distribution vector at the beginning of the batch to which the time $j$ belongs. Using these,

$$\mathbb{P}\left( \left| \frac{N_i(t_l)}{t_l} - t^*_i(\mu) \right| \geq 4\epsilon', \mathcal{G}^1_T \right) \leq \mathbb{P}\left( \frac{1}{t_l} \left| \sum_{j \in M_{t_l}} (I_i(j) - t^*_i(\Pi(\hat{\mu}(j)))) \right| + A_2 + A_3 \geq 4\epsilon', \mathcal{G}^1_T \right).$$

Since $|I_i(j) - t^*_i(\mu)| \leq 1$, and from Lemma K.2 we have that $t_l - |M_{t_l}| \leq t_l^{1/2}$. For $T \geq T_{\epsilon'}$ and $l \geq l_1(T)$, $A_3$ above satisfies

$$A_3 \leq \frac{\sqrt{t_l}}{t_l} \leq \frac{1}{\sqrt{t_{l_1(T)}}} \leq \frac{1}{T^{3/8}} \leq \epsilon'.$$

Next,
$$A_2 = \frac{1}{t_l} \sum_{\substack{j \in M_{t_l} \\ j < t_{l_0(T)}}} |t_i^*(\Pi(\hat{\mu}(j))) - t_i^*(\mu)| + \frac{1}{t_l} \sum_{\substack{j \in M_{t_l} \\ j \geq t_{l_0(T)}}} |t_i^*(\Pi(\hat{\mu}(j))) - t_i^*(\mu)|.$$

If $t_{l_0(T)} \leq K^2$, then the first term above is 0 since in this case, $M_{t_l} \cap \{1, \ldots, t_{l_0(T)}\}$ is empty, as the algorithm does not flip any coins in this period. Otherwise, the first term is bounded by $\frac{t_{l_0(T)}}{t_{l_1(T)}}$, which is further bounded by $\frac{1}{T^{1/2}}$, which for $T \geq T_{\epsilon'}$, is bounded by $\epsilon'$.

On $\mathcal{G}_T^1(\epsilon')$, the second term in $A_2$ is atmost $\epsilon'$, since for $j \geq t_{l_0(T)}$ $\hat{\mu}(j) \in \mathcal{I}_{\epsilon'}$. Thus, $A_2 \leq 2\epsilon'$. Thus, for $T \geq T_{\epsilon'} = \frac{1}{\epsilon'^{8/3}}$, and for $l \geq l_1(T)$,

$$\mathbb{P}\left(\left|\frac{N_i(t_l)}{t_l} - t_i^*(\mu)\right| \geq 4\epsilon', \mathcal{G}_T^1\right) \leq \mathbb{P}\left(\left|\sum_{j \in M_{t_l}} (I_i(j) - t_i^*(\Pi(\hat{\mu}(j))))\right| \geq t_l \epsilon', \mathcal{G}_T^1\right).$$

Let $S_n = \sum_{j \in M_n} (I_i(j) - t_i^*(\mu))$. Clearly, $S_n$ is a sum of 0-mean random variables. Whence, it is a martingale, and satisfies $|S_{n+1} - S_n| \leq 1$. Azuma-Hoeffding inequality then gives,

$$\mathbb{P}\left(\left|\frac{N_i(t_l)}{t_l} - t_i^*(\mu)\right| \geq 4\epsilon', \mathcal{G}_T^1\right) \leq 2\exp\left(-\frac{\epsilon'^2 t_l^2}{2|M_{t_l}|}\right) \leq 2\exp\left(-\frac{\epsilon'^2 t_l}{2}\right) \leq 2\exp\left(-\frac{\epsilon'^2 T^{3/4}}{2}\right),$$

where for the last inequality, we used that $l \geq l_1(T)$. Summing this over $l$ and $i$, the second term in (62) is bounded by

$$2KT\exp\left(-\frac{\epsilon'^2 T^{3/4}}{2}\right).$$

$\square$

## L  Details on the Experiments

In this section we report the numerical studies undertaken to validate our methods. We are interested in the question whether the asymptotic sample complexity result of Theorem 4.1 is representative at reasonable confidence $\delta$. Whether this is the case or not differs greatly between pure exploration setups. [27] see state-of-the-art numerical results in Bernoulli BAI for Track-and-Stop with $\delta = 0.1$, while [22] present a Minimum Threshold problem instance where the Track-and-Stop asymptotics have not kicked in yet at $\delta = 10^{-20}$.[1] The latter work suggests the difference may very well lie in the specifics of the lower-bound optimisation problem for each task, with the good case arising when the optimal solution $t^*$ to the lower bound puts positive mass on all arms, so that convergence of estimates does not require forced exploration. Our heavy-tailed best CVaR problem (4) indeed has full support, and our experiments confirm that the approach is practical at moderate $\delta$.

To focus on the heavy-tailed regime, we select arm distributions for which higher moments do not exist. In particular, we choose Fisher-Tippett ($F(\mu, \sigma, \gamma)$), Pareto ($P(\mu, \sigma, \gamma)$), and mixtures of Fisher-Tippett arms (these heavy tailed distributions arise in extreme value theory). The standard Fisher-Tippet distribution with shape parameter $\gamma$ has CDF $F_\gamma^F(x) = e^{-(1+\gamma x)^{-1/\gamma}}$ (continuously extended to $\gamma = 0$), and this is lifted to three parameters $F_{\mu,\sigma,\gamma}^F(x) = F_\gamma^F(\frac{x-\mu}{\sigma})$ by adding a location $\mu$ and scale $\sigma$. The $m$-th moment of $F_\gamma$ exists iff $\gamma < 1/m$. Similarly, CDF for $P(\mu, \sigma, \gamma)$ is given by $F_{\mu,\sigma,\gamma}^P(x) := 1 - (1 + \gamma(\frac{x-\mu}{\sigma}))^{-1-1/\gamma}$. For $\gamma > 0$, both $F(\mu, \sigma, \gamma)$ and $P(\mu, \sigma, \gamma)$, have unbounded support on the positive axis. $F(\mu, \sigma, \gamma)$ has unbounded support on the negative axis for $\gamma < 0$. We create interesting two-sided distributions by taking (binary) mixtures of these.

In our first experiment, we look at the distribution of the stopping time of the algorithm as a function of $\delta$. In this setup, there are three arms: arm 1 is a uniform mixture of $F(-1, 0.5, 0.4)$ and $F(-3, 0.5, -0.4)$, arm 2 is $P(0, 0.2, 0.55)$ and arm 3 is $F(-0.5, 1, 0.1)$ with respective CVaRs at quantile 0.6 being $-0.1428$, $0.974$ and $1.547$. We select $\epsilon = 0.7$ and $B = 4.5$. This is a moderately hard problem of complexity $V^{-1}(\mu)^* = 49.7$. The arm-densities are shown in Figure 2b.

---

[1][21, Figure 2] show that (unmodified) Track-and-Stop is not asymptotically optimal for problems with multiple correct answers including $(\epsilon, \delta)$-BAI. They have to go out to $\delta = e^{-80}$ to see the suboptimal asymptotics.

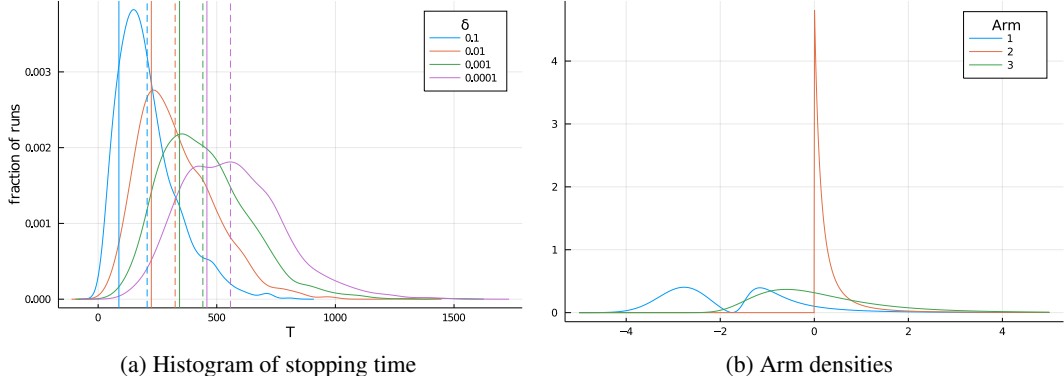

(a) Histogram of stopping time            (b) Arm densities

Figure 2: Histogram of stopping times based among 1000 runs on 3 arms with heavy-tailed distributions, with densities shown in (b), as a function of confidence $\delta$. Vertical bars indicate the lower bound (4) (solid), and a version adjusted to our stopping threshold (7) (dashed), i.e., the $n$ that solves $n = \beta(n, \delta)V(\mu)^{-1}$.

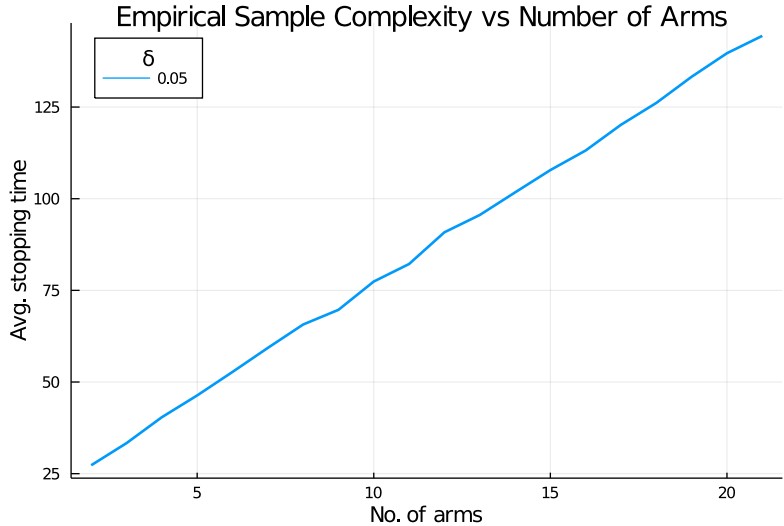

Figure 3: Stopping time (empirical sample complexity) of the algorithm at $\delta = 0.05$ as a function of number of arms. Each data point is an average of 1000 independent runs.

Figure 2a shows histograms of the sample complexity, together with the lower bound (solid vertical line) and a second reference point (dashed line) which is the $n$ that solves $n = V(\mu)^{-1}\beta(n, \delta)$, i.e. the time by which our stopping threshold activates for the optimal sampling allocation. We see that, for a range of practical $\delta$, the actual stopping time is very close to it. In particular, this means that the algorithms learns to approximate the optimal sampling strategy. We thus conclude that even at moderate $\delta$ the average sample complexity closely matches the lower bound, especially after adjusting it for the lower-order terms in the employed stopping threshold $\beta(n, \delta)$. This demonstrates that our asymptotic optimality is in fact indicative of the performance in practice.

In our second experiment, we let $\mathcal{L}$ be the collection of all distributions with $1.7^{th}$-moment bounded by $4.5$. We demonstrate in Figure 3 that the stopping time of the algorithm (empirical sample complexity), at $\delta = 0.05$, increases linearly with the number of arms, though currently theory shows a dependence of $K^4$, where $K$ is number of arms, in the lower-order terms (see the very last line of Lemma F.4). The experiment suggests that this $K^4$ dependency is an artefact, as it does not materialise in practise. For this experiment, we start with a 2-armed bandit: arm 1 being a uniform mixture of $F(-1, 0.5, .4)$ and $F(-3, 0.5, -.4)$, and arm 2 being $P(2.25, 0.1, 0.01)$. The CVaRs for these arms at $0.6^{th}$ quantile are $-0.1428$ and $2.4439$, respectively. Here, arm 2 is sub-optimal (recall

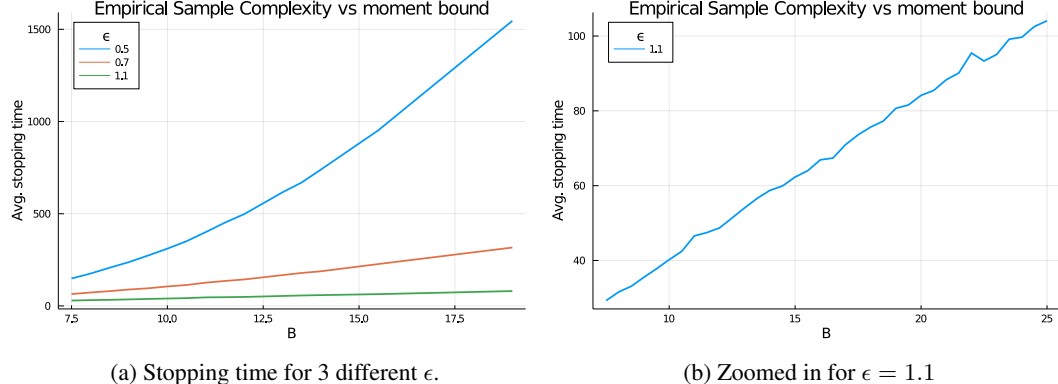

(a) Stopping time for 3 different $\epsilon$.          (b) Zoomed in for $\epsilon = 1.1$

Figure 4: Stopping time (empirical sample complexity) of the algorithm as a function of the moment bound, B. The graph shows the dependence for 3 values of $\epsilon$: 0.5, 0.7, and 1.1. We observe that for $\epsilon < 1$, the sample complexity is a convex function of $B$, and is linear for $\epsilon > 1$. Each data point is an average of 1000 independent runs.

that we are interested in the arm with minimum CVaR). We then keep adding more arms which are replicas of arm 2, thus minimizing the effect of other factors on the sample complexity.

In our final experiment, we look at the dependence of empirical sample complexity on the parameter $B$. The arms are the same as in our previous experiment, i.e., arm 1 is a uniform mixture of $F(-1, 0.5, .4)$ and $F(-3, 0.5, -.4)$, and arm 2 is $P(2.25, 0.1, 0.01)$. In this experiment, we change the comparator class $\mathcal{L}$ by changing only $B$. $\epsilon$ is set to one of $0.5, 0.7$, or $1.1$, and we start with $B = 7.5$, and increase it upto 20, in steps of 0.5. Figure 4a plots the stopping time of the algorithm as a function of $B$, for the 3 different values of $\epsilon$. It demonstrates that for $\epsilon < 1$, the dependence is convex, approaching to linear-dependence as $\epsilon \to 1$. We in fact sketch approximate lower and upper bounds for $V(\mu)$ (Section M), a quantity that characterizes the asymptotic sample complexity. These bounds show that sample complexity scales as $B^{\frac{1}{\epsilon}}$ for $\epsilon < 1$, and scales linearly for $\epsilon > 1$. This is clearly visible from the graph in Figure 4a, where the blue curve, which corresponds to $\epsilon = 0.5$, can be checked to be quadratically increasing, and the curve in Figure 4b demonstrates a linear dependence for $\epsilon = 1.1$.

In each case we perform 1000 independent replications. We use the stylised threshold $\beta(n, \delta) = \log \frac{1 + \log(n)}{\delta}$. This threshold is not currently allowed by theory. Yet we find that it is still conservative, as we do not observe a single mistake. Finally, instead of computing $\boldsymbol{t}^*(\hat{\boldsymbol{\eta}}_n)$ at each round, we make use of a technique recently introduced by [22] to reduce computation: namely after each round we perform a single step of an iterative saddle point solver for $\boldsymbol{t}^*$. We do not make use of their optimistic gradients, instead relying on classical $\sqrt{n}$-forced exploration. We use C-tracking from [27].

Finally, the computation of the stopping statistic (GLRT) and also the gradient involves an optimisation over $x_0$ as in the optimization problem in Proposition 3.5. We use bisection search to find the minimum in $x_0$. Even though this is not licensed by theory, we consistently observe in practice that after a few rounds all these minimisation problems are in fact quasiconvex in $x_0$. We use the ellipsoid method for the inner minimisation problem. As the number of terms grows by one each round, the overall run-time is $O(Kn)$ in round $n$.

We conclude that even at moderate $\delta$ the average sample complexity closely matches the lower bound (with adjusted stopping threshold $\beta(n, \delta)$). This demonstrates that our asymptotic optimality is in fact indicative of the performance in practice. The stopping time of the algorithm increases linearly with the number of arms. Moreover, for $\epsilon < 1$, the stopping time is a convex function of the class parameter, $B$, indicating that it is important to correctly estimate this parameter for smaller $\epsilon$.

# M   Interpretable Lower Bound Approximation

In this section we consider an approximate version of the lower bound problem. Even though it is heuristic, it is worthwhile as it gives an interpretable result. We take as our starting point (57), which we may invert to give us

$$
\mathrm{KL}_{\mathrm{inf}}^{\mathrm{U}}(\eta, x) \approx \left(\frac{4}{1-\pi}\right)^{1+1/\epsilon} B^{-1/\epsilon}(x - c_\pi(\eta))_+^{1+1/\epsilon}
$$

$$
\mathrm{KL}_{\mathrm{inf}}^{\mathrm{L}}(\eta, x) \approx \left(\frac{4}{1-\pi}\right)^{1+1/\epsilon} B^{-1/\epsilon}(c_\pi(\eta) - x)_+^{1+1/\epsilon}
$$

Let $\mu_1$ be the best CVaR arm, in that $c_\pi(\mu_1) < c_\pi(\mu_j)$ for all $j > 1$. The lower bound problem (see Lemma 3.1) then requires solving the approximate problem (denoted by a tilde)

$$
\tilde{V}(\mu) := \sup_{t \in \Sigma_K} \min_{j \neq 1} \inf_x \left(\frac{4}{1-\pi}\right)^{1+1/\epsilon} B^{-1/\epsilon} \left\{ t_1 (x - c_\pi(\mu_1))_+^{1+1/\epsilon} + t_j (c_\pi(\mu_j) - x)_+^{1+1/\epsilon} \right\}.
$$

$$(63)$$

Plugging in the optimiser $x = \frac{t_1^\epsilon c_\pi(\mu_1) + t_j^\epsilon c_\pi(\mu_j)}{t_1^\epsilon + t_j^\epsilon}$, which is the midpoint under the renormalised $\epsilon$-powered weights, results in

$$
\tilde{V}(\mu) = \left(\frac{4}{1-\pi}\right)^{1+1/\epsilon} B^{-1/\epsilon} \sup_{t \in \Sigma_K} \min_{j \neq 1} \frac{\Delta_j^{1+1/\epsilon}}{\left(t_1^{-\epsilon} + t_j^{-\epsilon}\right)^{1/\epsilon}},
$$

where we abbreviated $\Delta_j = c_\pi(\mu_j) - c_\pi(\mu_1)$ for $j \neq 1$ and $\Delta_1 := \min_{j \neq 1} \Delta_j$. From this point we can already see that the characteristic time, $1/\tilde{V}(\mu)$, scales with $B^{1/\epsilon}$, which is clearly visible e.g. the blue line in in Figure 4a, corresponding with $\epsilon = 1/2$, and which matches a quadratic (quadrupling when $B$ doubles).

At this point we can follow [27, Appendix A.4] and obtain an interpretable sandwich on $\tilde{V}(\mu)$ with a multiplicative factor $2^{1/\epsilon}$.

**Lemma M.1.**

$$
\left(\frac{1-\pi}{4}\right)^{1+1/\epsilon} B^{1/\epsilon} \sum_j \frac{1}{\Delta_j^{1+1/\epsilon}} \leq \tilde{V}(\mu)^{-1} \leq 2^{1/\epsilon} \left(\frac{1-\pi}{4}\right)^{1+1/\epsilon} B^{1/\epsilon} \sum_j \frac{1}{\Delta_j^{1+1/\epsilon}}.
$$

*Proof.* Let $C = \left(\frac{1-\pi}{4}\right)^{1+1/\epsilon}$. First, by plugging in the sub-optimal choice for $t$ given by

$$
t_j = \frac{\Delta_j^{-1-1/\epsilon}}{\sum_j \Delta_j^{-1-1/\epsilon}},
$$

where we interpret $\Delta_1 = \min_{j \neq 1} \Delta_j$. We then find

$$
\tilde{V}(\mu)^{-1} \leq CB^{1/\epsilon} \left(\sum_j \Delta_j^{-1-1/\epsilon}\right) \max_{j \neq 1} \left(\left(\frac{\Delta_1}{\Delta_j}\right)^{1+\epsilon} + 1\right)^{1/\epsilon} \leq \sum_j \frac{2^{1/\epsilon} CB^{1/\epsilon}}{\Delta_j^{1+1/\epsilon}}.
$$

We may also obtain a lower bound on the characteristic time of the same order by considering the sub-optimal choice $x = c_\pi(\mu_1)$ in (63) instead. We obtain

$$
\tilde{V}(\mu^*) \leq \sup_{t \in \Sigma_K} \min_{j \neq 1} t_j C^{-1} B^{-1/\epsilon} \Delta_j^{1+1/\epsilon} = \frac{1}{\sum_j \frac{CB^{1/\epsilon}}{\Delta_j^{1+1/\epsilon}}}.
$$

Taking the reciprocal gives the result. $\qquad\square$