# OpenReview forum: "Optimal Best-Arm Identification Methods for Tail-Risk Measures"
_NeurIPS.cc/2021/Conference — NeurIPS 2021 Poster_

### Official Review · Reviewer_rZt1 · 2021-07-07

**Rating:** 7
**Confidence:** 3

**Summary:**

The paper considers best CVaR arm identification in a fixed confidence setting. They can also deal with conic combinations of mean and CVaR, as well as VaR based criteria.

Main contributions include a lower bound which involves optimization over reals, and an algorithm which is asymptotically optimal in terms of expected sample complexity.

**Limitations And Societal Impact:**

No major societal impact

**Main Review:**

This paper considers the CVaR optimisation problem in a multi-armed bandit setting. Similar works exist in the context of mean [2] and for fixed budget best CVaR arm[39]. However, the CVaR optimisation problem over the distribution space L^K involves more intricate analysis. This paper carries out an involved analysis for obtaining the explicit KL characterisation of the lower bound. The authors also propose an algorithm that meets the lower bound asymptotically as the confidence δ goes to zero. Numerical results indicate that the average sample complexity matches the lower bound well at moderate values of δ.

Overall, the paper studies a problem which has been studied a lot in the recent years, but proposes a sophisticated analysis technique which is worth recording and could be reusable for other settings.

Could similar techniques be used for other convex risk measures? For example, utility based shortfall?

**Time Spent Reviewing:**

6

---

> ### Author Response · Authors · 2021-08-10
> **Thank you for the positive comments and for suggesting an interesting extension.**
>
> It is indeed an intriguing question. We think that a similar technique should be usable for a much general class of objectives. We see a lot of structure in the problem (e.g. the Kusuoka representation). However, in the current analysis, we explicitly use the variational representations of CVaR (see, (2) and (3) in Section 3). It is not clear how these representations generalize to more general risk measures. Further work is needed to solve this problem for more general risk measures.

---

> > ### Comment · Reviewer_rZt1 · 2021-08-19
> > **Satisfactory responses**
> >
> > Thank you for your response and comments. I am satisfied, and retain my recommendation.

---

### Official Review · Reviewer_sUiJ · 2021-07-09

**Rating:** 7
**Confidence:** 3

**Summary:**

The authors study the problem of multi armed bandits with the goal of identifying the arm with smallest (i) CVaR (ii) VaR or (iii) conic combination of CVaR and mean. This problem has applications in finance and clinical trials and captures a notion of risk sensitivity towards undesirable outcomes. The authors show that in the parametric case where the arm reward distributions follow a canonical SPEF, the problem reduces to best arm identification.

The authors study the VaR problem without any assumptions on the distributions, and study the CVaR problem assuming that distributions satisfy bounded $(1+\epsilon)$ moments. The proposed algorithm projects the empirical distribution to the considered family of distributions and considers an exploratory transform that guides the arm sampling distribution. At each time the empirical reward distribution suggests an arm with empirical minimum CVaR - this is the null hypothesis which is tested against all alternatives using GLRT. The algorithm upon satisfying the termination condition outputs the arms with the empirical minimum CVaR.

The authors show that the proposed algorithm asymptotically achieves the optimal sample complexity as the error threshold $\delta \to 0$.
The results are supported by numerical simulations demonstrating good performance on instances with Bernoulli arms. The performance of the algorithm hinges on the asymptotic convergence of the track-and-stop exploration protocol which is often fast even when the error parameter delta is moderate.

**Ethical Concerns:**

None to the best of my knowledge

**Limitations And Societal Impact:**

Yes, I believe the authors have sufficiently addressed such concerns

**Main Review:**

Overall the paper is well written and discusses prior work in sufficient detail, to the best of my knowledge. The considered notion of CVaR and VaR minimization have been studied in prior work, but the optimal rates have not been established for any regime of error tolerance, $\delta$ until this paper. I think this paper presents an important stepping stone towards exactly characterizing the minimax rates in best arm identification problems beyond the standard notion of mean. The algorithm presented in the paper is new and its proof uses novel analytical tools, which are of independent value.
- The only suggestion I can provide in terms of the writing of the paper is: it might help to define the CVaR and VaR problems informally in the introduction instead of referring the reader to Section 2. The contents of the introduction would be more interpretable that way.
- In financial settings, I can imagine more than best arm identification (which may take exponentially long in K if the gap is small), regret minimization with stochastic / adversarially chosen rewards is also of interest. Does the analysis here provide any intuition as to what to expect for the regret minimization setting as $T\to\infty$?
- Are there instances where the best arm and the arm with minimum CVaR are very different from each other? This would provide more justification for proposing algorithms specific for CVaR minimization and mean-CVaR minimization.
- Is there any intuition about the tightness of the algorithm in the non-asymptotic regime? It might help to include a discussion in the body of the paper as to the challenges of extending to the case when delta does not $\to 0$.


**Time Spent Reviewing:**

7

---

> ### Author Response · Authors · 2021-08-10
> **Thank you for raising some interesting questions. Responses are given below.**
>
> 1. We will incorporate the changes suggested by the reviewer in point 1.
>
>
> 2. As discussed in Appendix I, we think that the concentration inequalities for KLinfL and KLinfU developed in this paper can be used to construct the UCB/LCB index. A recent paper (see, reference [59] in our appendix) shows that a similar algorithm for the mean setting is optimal. We expect that the UCB algorithm with KLinfU/L based index should give an optimal algorithm for the risk-averse setting as well. Furthermore, like in the mean case, the lower bound for the risk-averse regret-minimization problem would also be in terms of these projection functionals. However, the details still need to be worked out.
>
>
> 3. Instances where the best-mean-arm and minimum-CVaR-arm are very different can be constructed using the following observation.
> Fix $\pi = 0.8$, i.e., the confidence at which CVaR is computed is fixed at $0.8$, and let $\delta_{x}$ denote a point mass at $x$. Consider a distribution with 2-point support, say $$\mu_1 = 0.8 \delta_{0} + 0.2 \delta_{1}.$$ Clearly, the mean of $\mu_1$ equals $0.2$, and the CVaR of $\mu_1$ at $\pi$ equals $1$. Now, we can construct distributions with mean smaller/larger than $0.2$ using 2-point distributions by only moving the lower point mass up/down, while keeping it below $1$. However, this keeps the CVaR fixed at $1$. More concretely, fix $\pi$ at $0.8$ and consider the following 3-armed MAB instance:
> $$ \text{Arm }1: 0.8 \delta_{0} + 0.2 \delta_{1}. \quad \text{Mean }= 0.2,\text{ CVaR }= 1.$$
> $$ \text{Arm }2: 0.8 \delta_{0} + 0.2 \delta_{0.5}.\quad\text{ Mean }= 0.1,\text{ CVaR }= 0.5.$$
> $$ \text{Arm }3: 0.8 \delta_{-0.5} + 0.2 \delta_{2}.\quad \text{ Mean }= 0,\text{ CVaR }= 2.$$
> Observe that these arms do not belong to a canonical SPEF. Clearly, mean of Arm 1 > mean of Arm 2 > mean of Arm 3. However, CVaR of Arm 2 < CVaR of Arm 1 < CVaR of Arm 3. Thus, the best-mean and the best-CVaR arms are very different. Although we have demonstrated this for 2-point distributions, it is easy to construct similar examples for general distributions that may occur in practice.
>
>
> 4. In our experiments, for example, see Figure 2, we show numerically that the algorithm's stopping time is very close to the lower bound computed using the adjusted threshold, at moderate $\delta$, indicating that the algorithm matches the lower bound even for moderate $\delta$. Note that 'adjusted threshold' means that we solve with equality the sample complexity lower bound (4), with the lower-bound threshold of $\log(1/\delta)$ replaced by the stopping-rule threshold $\beta(n, \delta)$ from line 1576. As can be seen in Figures 1 and 2, this accurately predicts the expected sample complexity in the experiments.

---

> > ### Comment · Reviewer_sUiJ · 2021-08-20
> > **Response to the authors**
> >
> > Thank you for the clarifications.
> > I would recommend that the authors include a short remark with the particular example in point 3 above differentiating the mean and CVaR objectives. It would be helpful for a reader to understand qualitatively when these objectives differ to contrast the claim made for the case of canonical SPEFs.
> > I maintain my score of the paper as an accept.

---

### Official Review · Reviewer_Gtxo · 2021-07-16

**Rating:** 7
**Confidence:** 2

**Summary:**

This paper addresses the multi-armed bandit best arm identification problem which aims to identify the arm with the minimum tail risk measured by VaR, CVaR, or a weighted sum of CVaR and the mean. It proposes an optimal $\delta$-correct algorithm that works with a mild restriction on the arm distributions for CVaR and works with any arm distributions for VaR. In particular, the paper develops a lower bound for the problem and shows that the proposed optimal $\delta$-correct algorithm has the sample complexity that matches the lower bound asymptotically (as $\delta \rightarrow 0$). Additionally, the paper also suggests a solution to control the trade-off between the time computational complexity and the sample complexity, which is useful in practical applications.

**Main Review:**

The paper makes an important contribution to the multi-armed bandit best arm identification problem by considering 3 risk measures: VaR, CVaR, and a weighted sum of CVaR and the mean (mean-CVaR), all of which are well-motivated in the introduction. The proposed optimal $\delta$-correct algorithm is capable of handling a large class of arm distributions (with only mild assumption when the risk measure is CVaR and mean-CVaR). Moreover, its sample complexity exactly matches the lower bound as $\delta \rightarrow 0$. I think it is a good paper, so I tend towards acceptance. There are only several minor questions as follows:
1. What are the advantages/disadvantages between existing solutions in the ($\epsilon$,$\delta$)-PAC setting and the proposed optimal $\delta$-correct algorithm in this paper, e.g., how large $\epsilon$ should be such that existing PAC solutions are more favourable than the proposed algorithm in terms of sample complexity/computational complexity?
2. When the arm distributions are parametric but not from canonical SPEF, what are the implications when we apply the proposed algorithm (which is designed for non-parametric arm distributions)? For example, the knowledge of the parametric arm distribution may not be fully utilized.
3. I think that paper will be easier to read for people without much background if references and some brief explanations are included. For example, including a brief explanation of the intuition behind $KL_{\inf}^{U}$ and $KL_{\inf}^{L}$ in equation (1); references in lines 189-190 and 413-414; more details in lines 278-280; the derivation of equation (4) (or mentioning the specific chapter in the book [40]); the derivation of the GLRT in line 294.
4. Why are there only heavy-tailed arm distributions in the experiments? Maybe the experiments with light-tailed arm distributions should also be demonstrated.
5. The experiments in figure 3 in Appendix K contain arm distributions whose CVaR values are quite far apart (-0.1428 vs. 2.4439), so I suspect it is easy for the algorithm to identify the best arm with a small sample complexity in this case. Thus, the linear relationship between the empirical sample complexity and the number of arms will be more persuasive if we consider similar arm distributions as those in Figure 2 where the sample complexity is roughly 200 for $\delta = 0.1$ and only $3$ arms (compared with figure 3 where the sample complexity is only around 150 for $\delta = 0.05$ and $20$ arms).


**Time Spent Reviewing:**

24

---

> ### Author Response · Authors · 2021-08-10
> **Some of the points raised are intriguing and are subject for exciting future research.**
>
> 1. To the best of our knowledge, there is no existing literature on the $(\epsilon,\delta)$-PAC mean-CVaR/VaR BAI problem in the generality we work in. The $(\epsilon,\delta)$-PAC version of the CVaR identification problem for $\epsilon>0$ is an interesting extension. The reference [20] in our paper develops the (surprisingly non-trivial) extension of the Track-and-Stop family of methods to problems with more than one correct answer (which is the case for some instances when $\epsilon>0$). This extension is orthogonal to our core contribution, which is getting the details right for the (already subtle) case of $\epsilon=0$. Whether and when earlier, non-(asymptotically)-instance-optimal methods currently have the advantage is an empirical question that we did not attempt to study here. We believe that there is no intrinsic hurdle here, and that the advantage of earlier methods will shrink over time as we come to refine our understanding of the design of asymptotically optimal methods.
>
> $$ $$
>
> 2. and 4. These are intriguing questions that also occupy us. We would love to characterize the effect of the assumption about the ambient class of distributions (here we consider moment constraints and SPEFs), as well as the effect of the specific bandit instance at hand. The optimization-based expression for the characteristic time ($V(\mu)$ in (5), Lemma 3.1) is unfortunately currently opaque. The reviewer asks in particular what happens if we work in our moment constrained class and execute the algorithm at a bandit with arms from a (sub)-class. We currently have no good way of quantifying this beyond the worst-case picture of our Appendix L. Our intuition though is that if the input is say, Gaussian, there may still be distributions rather close by in the moment-constrained class with rather different CVaRs, implying that the sample complexity will not be especially small.
>
> $$ $$
>
> 5. As we point out in the discussion around Figure 3, the current analysis is consistent with the sample complexity being max{the lower bound, $K^4$}. Our Figure 3 is intended to show that the $K^4$ term does not materialize in practice, thus giving confidence that the algorithm can be used in non-asymptotic regimes. It is not clear to us how making the identification problem more complex (thus increasing the first maximand) would help in this regard. However, we tried this experiment with the suggested difficult bandit instance and saw a similar plot with a linear dependence on the number of arms. If we dig deeper into the possible failure mode hinted at by the analysis, we see that it points out that it may take $K^4$ rounds for the algorithm to sufficiently deviate from uniform sampling. There is no reason to believe specifically in a fourth power. Recent finite-time analyses for track-and-stop (see e.g. reference [22] in our paper) bring the additional cost down to $K^2$ for the best arm problem, and even conjecture that $K\log(K)$ is enough.
>
> $$ $$
>
> 3. We will include the changes suggested by the reviewer in point 3.

---

> > ### Comment · Reviewer_Gtxo · 2021-08-19
> > **Response to Authors**
> >
> > Thank you for your reply. I'm satisfied with the answers, so my score remains the same.

---

### Official Review · Reviewer_w1AF · 2021-07-18

**Rating:** 7
**Confidence:** 3

**Summary:**

The paper studies the problem of identifying the arm with the minimum CVaR, VaR, or a conic combination of the mean and CVaR. It provides an \delta-correct algorithm that operates on possibly heavy-tailed arm distributions and matches the asymptotic lower bound on the expected number of samples needed as delta approaches 0. The paper also provides a result of separate interests, i.e., an anytime-valid confidence interval for CVaR estimation, which is tighter than truncation-based intervals (under certain conditions). Finally, the empirical studies show that the asymptotic sample complexity result of the proposed algorithm is indicative of its performance in practice.

**Limitations And Societal Impact:**

Yes, the authors have adequately discussed the limitations and societal impact of their work.

**Main Review:**

The authors have provided a clear exposition of the motivation and the theoretical results of the work. In particular, the KL_inf-based anytime-valid confidence intervals for CVaR are (to the best of my knowledge) very novel. In addition, the empirical results show that the asymptotic optimality of the proposed algorithm is indicative of the performance of the algorithm in practice.

- In general, the paper will benefit from more detailed discussions on interpreting some of the defined quantities (especially the ones defined in L235-239 for the dual formulation and the ones defined in L252-256). In their current forms, the readers are hard to gain intuitions on them.
- Could the authors comment on differences between this paper and "Statistically Robust, Risk-Averse Best Arm Identification in Multi-Armed Bandits" (https://arxiv.org/pdf/2008.13629.pdf)?
- Empirically, what is the performance of the algorithm for identifying the best arm under VaR?



**Time Spent Reviewing:**

6 hours

---

> ### Author Response · Authors · 2021-08-10
> **Thank you for the positive comments.**
>
> 1. These are useful observations. We now include a discussion detailing the quantities in lines 235-239 and 252-256. These introduce notation needed in Theorem 3.4 and Proposition 3.5 and are related to the dual representations of KLinfU/L and the overall lower bound optimization problem. For example, the quantities in lines 235-237 correspond to the scaling of the input measure, used to get the optimal primal measure in Theorem 3.4.  The sets defined in lines 238-239 correspond to the dual variables that ensure these scalings to be non-negative. These are necessary to ensure that the corresponding scaled primal measure is a probability measure. One can similarly interpret the quantities in lines 252-256 as solutions of the joint lower bound problem. We now add these to the main text.
>
> $$ $$
>
> 2. There are 3 key differences between the current paper and that on statistically robust, risk-averse BAI. We work in the fixed-confidence setting, while in their work, the authors consider the fixed-budget setting of the BAI problem. The two settings require a different set of techniques to come up with optimal algorithms and their analysis. Secondly, in our work, knowledge of B and $\epsilon$ is crucial, while the authors do not assume the knowledge of these parameters in their setting. Finally, we propose asymptotically optimal algorithms, while the algorithms proposed in the suggested paper are only 'almost' optimal.
>
> $$ $$
>
> 3. In the current work, we chose to focus experiments on the CVaR problem, which is both theoretically more challenging, and a practically more important measure of risk than VaR. We do not perform experiments for the best-VaR-arm identification. However, as testing a quantile reduces to testing a Bernoulli random variable, our intuition is that the algorithm's performance should not be very different from that for Bernoulli-best-arm identification (for which there are experiments demonstrating the success of track-and-stop in [27])

---

> > ### Comment · Reviewer_w1AF · 2021-08-18
> > **Response to rebuttal**
> >
> > Thanks for the detailed reply. I'm satisfied with the responses to the comments and remain my score.

---

### Official Review · Reviewer_1ikr · 2021-07-25

**Rating:** 7
**Confidence:** 4

**Summary:**

The paper studies the problem of identifying the arm with smallesst CVaR and VaR. The lower bound for the specific structured best-arm identification problem is studied. The paper also proposed an algorithm that achieves asymptotic optimality in sample size.

**Limitations And Societal Impact:**

The authors have adequately addressed the limitations and potential negative societal impact of their work.

**Main Review:**

The paper provides asymptotically matching upper and lower bounds for the best-arm identification methods for CVaR and VaR etc. I appreciate the efforts the authors have made for organizing all the content. However, the presentation and clarity of the paper are the major obstacles for understanding the whole paper, along with some technical incorrectness and ambiguity. I'm listing all my concerns as below:

1. Although it might be trivial, it's indeed unclear to me how standard arguments in best arm identification lead to Equation (4) and (8). First, it is better to indicate where the equations come from in the standard references. For example, (4) seems to come from a direct result of Theorem 33.5 in [1]. However, even with the results there, it is unclear why the results of best arm identification can be directly applied to a different problem of identifying the arm with smallest CVaR and VaR. It would be better if the authors can make the proof of (4) and (8) explicit.

2. Furthermore, Equation (4) and (8) make me even confused since they're suggesting that the problem of identifying the arm with smallest CVaR is exactly the same as the original best-arm identification problem in both upper and lower bounds. And the follow-up analysis (Lemma 3.1) is mostly simplification that does not depend on the structure of CVaR. If (4) and (8) turns our to be a simple application of the known result, it would be better if the authors can be clear on their own technical contributions.

3. With the above point said, I believe one of the most important contribution of this paper is a statistically and (maybe) computationally efficient algorithm. However, the proof in Appendix F appears to be wrong. In particular, the authors only show that $G_z$ has its $(1+\epsilon)$ moment bounded. However, the authors fail to show that among all the distributions that has moment bound, the way $G_z$ moves the mass is the optimal way such that $d_K(F, G_z)$ is minimized. Indeed, this is definitely wrong: the authors suggest that one shall move the mass on the left and right tail equally to point 0. However, in a highly asymmetric distribution, it would incur a much smaller distance if we only move one side of the mass that has a really long tail. The authors also didn't discuss whether the step of stopping rule is computationally efficient or not.

4. There are many ambiguities in the paper that is not rigorously defined, which makes it hard to understand the paper. My suggestions for improving the presentation of the paper are listed as below:

(a). In Section 4 when you describe the algorithm, it is better to write it in a formal algorithmic environment and be explicit on all the procedures. It is unclear to me how 'any reasonable tracking rule' are applied, how 'empirically-minimal CVaR arm' are selected. These might appear trivial but are not rigorously defined. One shall replace the languages there with equations.

(b). Similar issues also happen in other places, e.g. the notation $\mathcal{\epsilon}$ in line 301, the notation $\mathcal{L}$ in line 285 (although it is defined as the set of moment bounded distributions in introduction, later it is also used to refer to any set of distributions in line 150 and 199 etc.), and the notion of $\delta$-correctness in the paper.



[1] Lattimore, Tor, and Csaba Szepesvári. Bandit algorithms. Cambridge University Press, 2020.


**Update based on author responses:

The authors resolved the questions I had and the proof of Lemma F.1 looks good to me after the clarification. I have adjusted the score accordingly.

**Time Spent Reviewing:**

4

---

> ### Author Response · Authors · 2021-08-10
> **The main criticism raised is serious but in fact invalid. We invite the reviewer to reevaluate the paper.**
>
> The reviewer's main criticism is in point 3 where the reviewer believes that the proof of Lemma F.1 is wrong. Please note that the proof is correct. In addition, the reviewer suggests minor notational and presentation changes. We incorporate these and thank the reviewer for the suggestions.
>
> $$ $$
>
> Point 3:
>
> As for Lemma F.1, in lines 1189-1190 we choose z to be the minimum distance of F from elements in L and use this z to come up with a distribution in L which is at this minimum distance from F (i.e., $G_z$).
>
> To see this, from the definition of $G_z$, it is easy to see that its distance from F is at most z. We next demonstrate that $G_z$ is in L (lines 1191-1194). Since in lines 1189-1190 z was chosen to be the minimum distance, the constructed $G_z$ cannot be closer than the closest distribution in L, showing that it is indeed an optimal solution.
>
> Please note that we do not suggest moving an equal mass from both tails always. We move equal mass only if it is possible to do so. In particular, in the asymmetric setting where it may not be possible, Lemma F.1 does not suggest moving an equal mass from both the tails.
>
> Recall from Section 4 that we project in the Kolmogorov metric (also defined in lines 1286-1288), which is the supremum in the point-wise difference of CDFs. Intuitively, if we pull some mass from one of the tails to 0, say from right, we already incur some supremum-distance (namely at 0 or $0^-$). Pulling an equal mass from the left tail to 0 is free. Moreover, doing this is preferable, as it helps towards satisfying the moment constraint. So the optimum $G_z$ is pulling an equal mass from the 2 tails, as long as this is possible. We now include a picture explaining this in the appendix, and also recall the definition of the projection function, $d_K$ in Appendix F.
>
> The overall computational cost of the proposed algorithm has been briefly discussed in Section 4. A detailed discussion on this can be found in Appendix J, where we observe a quadratic dependence on the total number of samples, and propose a batched/doubling algorithm that improves the running time to almost linear in the total number of samples. We also discuss the alternatives leading to improvement in the computational cost of the main algorithm in Section 3, where we introduce the dual formulations for the lower bound optimization problem.
>
> $$ $$
>
> Point 1:
>
> (4) is indeed a standard lower bound for any $\delta$-correct algorithm. See, e.g., references [21], [22], [34] in the paper, where the authors use the same argument for deriving lower bounds in settings beyond the best-mean-arm problem. The problem-specific quantity in (4) is $V(\mu)$, specifically, the alternative set, $A^c_1$, over which the minimization is done. As per your suggestion, we now refer to Theorem 33.5 in [40].
>
> A proof of (8) is in Appendix E, where we give detailed proof of all the parts of Theorem 4.1. To avoid any confusion, we have added a pointer to that appendix around (8).
>
> $$ $$
>
> Point 2:
>
> (4) and (8) are not the same as their analogues for the best-mean-arm setting. The difference is in the definition of the alternative set, $A^c_1$, and hence, in the two KL-projection functionals, KLinfU and KLinfL. As mentioned in the introduction, these KL-functionals have CVaR constraints, which makes one of these a non-convex optimization problem, and requires more nuanced analysis techniques to handle. On the other hand, the corresponding functionals in the mean-setting have mean constraints in the definition and are symmetric and convex. We now add a line highlighting these differences again, near (4) and Lemma 3.1.
>
> $$ $$
>
> Point 4:
>
> 4.a) In our proof of Theorem 4.1, and in the numerical experiments, we use C-Tracking from [27]. We have now made this clear in the main algorithm in Section 4 and have described the C-tracking rule in the appendix. As per your suggestion, we have now clearly stated the recommendation rule to mean "the arm with the minimum CVaR of the corresponding empirical distribution" and included the exact mathematical form of it in the main text.
>
> 4.b) Though we have used L to mean the moment-bound set for most of the exposition, and mentioned explicitly wherever it refers to all the probability measures, we now change the notation in the paper to use L specifically for the moment-bounded set defined in the introduction, and a use different notation for other sets.

---

### Decision · Program_Chairs · 2021-09-27

**Decision:**

Accept (Poster)

**Comment:**

This paper studies the multi-armed bandit problem with the goal of identifying the arm with the smallest CVaR/VaR/conic combination of CVaR and the mean. It is a significant question with applications in various domains. This paper proposes an algorithm that projects the empirical distribution to the considered family of distributions and considers an exploratory transform that guides the arm sampling distribution. It shows that the algorithm asymptotically achieves the optimal sample complexity. I agree with the reviewers that this paper has made interesting technical contributions and I am happy to recommend acceptance.